# Linear and inverted U-shaped dose-response functions describe estrogen effects on hippocampal activity in young women

Janine Bayer [1], Jan Gläscher [1], Jürgen Finsterbusch[1], Laura H. Schulte[1] & Tobias Sommer[1]

In animals, 17-beta-estradiol (E2) enhances hippocampal plasticity in a dose-dependent, monotonically increasing manner, but this relationship can also exhibit an inverted U-shaped function. To investigate E2's dose-response function in the human hippocampus, we pharmacologically increased E2 levels in 125 naturally cycling women (who were in their low-hormone menstruation phase) to physiological (equivalent to menstrual cycle peak) and supraphysiological (equivalent to levels during early pregnancy) concentrations in a placebo-controlled design. Twenty-four hours after first E2 intake, we measured brain activity during encoding of neutral and negative pictures and then tested recognition memory 24 h after encoding. Here we report that E2 exhibits both a monotonically increasing relationship with hippocampal activity as well as an inverted U-shaped relationship, depending on the hippocampal region. Hippocampal activity exhibiting a U-shaped relationship inflects at supraphysiological E2 levels, suggesting that while E2 within physiological ranges stimulates hippocampal activity, supraphysiological ranges show opposite effects.

[1] Department of Systems Neuroscience, University Medical Center Hamburg-Eppendorf, Martinistr. 52, 20246 Hamburg, Germany. Correspondence and requests for materials should be addressed to J.B. (email: j.bayer@uke.de)

A large body of animal studies has characterized the manifold effects of 17-beta-estradiol (E2) on brain circuits[1–3]. These effects have been most intensely investigated in the hippocampus, where E2 has been found to modulate neural plasticity[1,4,5]. Although far less studied than its effects on the hippocampus, E2 also impacts neuronal processes in the adjacent amygdala[6,7]. E2 affects neural plasticity through synaptogenesis and glutamatergic neurotransmission leading to, for example, increases in neuronal spine density or long-term potentiation (LTP) magnitude[1,4,5]. Interestingly, based on studies that have induced more than one level of E2 it was proposed that the dose-response relationship between E2 and cellular measures of neural plasticity might not always follow the traditional sigmoidal but often an inverted U-shaped function[8–10]. That is, one would expect increasing benefits until medium levels of E2 but then a decrease and none or even negative effects at high levels[4].

The inconsistency in reported responses might be resolved at least for the hippocampus with the observation that cellular measures of neural plasticity monotonically increase with E2 levels mostly at physiological E2 concentrations[8–13]. Only when supraphysiological E2 levels (i.e., levels above estrous cycle peaks) are reached does the dose-response relationship exhibit an inverted U-shape[8–11]. However, a few studies have also reported monotonic improvements at clearly supraphysiological E2 levels, and some effects only occur at supraphysiological levels[9,14–16]. Only one study investigated the effect of different E2 dose on cellular measures of neural plasticity in the amygdala, where the neuroprotective effect of E2 on the amygdala followed a sigmoidal function at supraphysiological levels[6]. The question of whether the response to E2 turns into an inverted U even at physiological levels or only at supraphysiological levels is of relevance to determine whether E2 would normally reach concentrations disadvantageous to functions mediated by the hippocampus and the amygdala during the regular menstrual cycle.

In humans, hormonal effects on hippocampal and amygdala activity, as well as macroscopic volumes, have been detected using magnetic resonance imaging (MRI)[2,17–21]. However, most of these studies have focused on the consequences of natural fluctuations in hormone levels across the menstrual cycle, or induced by pharmacotherapies, e.g., hormone replacement therapy (HRT), in order to better understand these specific conditions[2]. For a variety of reasons, these studies can only be compared to a limited degree to the placebo-controlled, randomized E2 treatment regimens in young, healthy animals. In addition, no study has investigated the effects of different E2 doses in young women. Given the large body of evidence from animal studies on the effects of E2 on brain function, E2's role in modulating neural processes in humans is surprisingly understudied.

Therefore, the current study aimed to more directly extend the striking findings on the central E2 effects from animal research to humans by systematically increasing E2 levels in young, healthy women in a randomized, double-blind, placebo-controlled design. E2 valerate in doses of 0, 2, 4, 6 or 12 mg were orally administered on two consecutive days to 125 naturally cycling women during the low-hormone menstruation phase. This E2 treatment regimen induced a wide range of E2 levels, from physiological (within 2- and 4-mg groups; equivalent to cycle peak) to supraphysiological (within 6- and 12-mg groups; equivalent to early pregnancy) on the second day. This wide range enabled us to test our hypothesis that the dose-response function between E2 and neural activity assessed by functional MRI (fMRI; as a proxy for changes in neural plasticity) increases monotonically within physiological ranges and turns into an inverted U-shape only at supraphysiological levels.

As most evidence exists for E2 effects on hippocampal neurons[1,2], we employed a recognition memory paradigm known to elicit hippocampal activation[22]. To concurrently investigate the effects of E2 on the amygdala, volunteers encoded both neutral and emotionally arousing scenes while in the MR scanner. In addition to the hippocampus and amygdala, this paradigm activates a larger network of cortical and subcortical areas which allowed us to also explore E2 effects in other brain areas that express estrogen receptors but have not yet been addressed by animal studies using whole-brain fMRI[22–24]. Additionally, regional cerebral blood flow (rCBF) was measured at rest via arterial spin labeling (ASL) to assess task-independent effects of E2, which could increase the baseline perfusion, thereby leading to smaller relative task-related blood-oxygen level-dependent (BOLD) effects[25].

Twenty-four hours after encoding, a recognition test was administered outside of the scanner with confidence ratings followed by remember/know judgements for 'old' responses (Fig. 1). The memory test was followed by an arousal rating for all studied pictures. Salivary concentrations of E2, progesterone, and cortisol were measured on all three testing days; serum E2 levels were measured in a subsample of 90 women. Linear and quadratic regression analyses, using the increase of E2 concentrations in saliva from Day 1 (baseline) to Day 2 (E2 peak) as a predictor, were used to determine any linear or quadratic (potentially inverted U-shaped) relationships between E2 levels and neural activity in the hippocampus and the amygdala. Importantly, this strictly data-driven approach does not make any a priori assumptions about the inflection point of a potential U-shaped relationship. Additional robust regression analyses were performed to make sure that results were not biased by outliers. Linear versus quadratic model fits were compared using the Bayesian Information Criterion (BIC).

We found that posterior hippocampal activity not only monotonically increases with E2 levels, but activity can also decrease after an initial increase, depending on subregion. Notably, this decrease (inflection of the inverted U) occurs only at supraphysiological levels. We conclude that E2 stimulates hippocampal activity within physiological ranges, but this effect can be attenuated at supraphysiological E2 levels.

## Results

**Volunteer demographics.** Experimental groups did not differ significantly with respect to weight, body-mass index, age, education or lifetime pregnancy (all $p$'s > .1; Supplementary Table 1). Importantly, there was no difference in self-reported side-effects or guesses of whether having received placebo or E2 (all $p$'s > .2).

Confirming the effectiveness of the pharmacological manipulation, the increases in salivary and serum E2 levels from baseline (Day 1) to expected peak (Day 2) differed between experimental groups (both $p$'s < .001; Fig. 2a). E2 levels in the 2- and 4-mg groups roughly matched E2 levels during menstrual cycle peaks in late follicular and luteal phase[26]. E2 levels in the 6- and 12-mg groups were comparable to E2 levels in early pregnancy[27]. Both linear and quadratic relationships were found between the dose of E2 valerate given per kg bodyweight and salivary Day 1 to Day 2 E2 increases (linear: $t(123) = 21.49$, $p < .001$, $\beta_{std} = 1.006$; quadratic: $t(123) = 8.82$, $p < .001$, $\beta_{std} = 1.023$; Fig. 2b). The inter-individual variance within groups was high, leading to an overlap of the individual E2 increases between groups (Fig. 2b; Supplementary Table 2). Additional analyses showed that this variability cannot be accounted for by differences in age (see Supplementary Note 1). Further statistical analyses were conducted using Day 1 to Day 2 increase in salivary

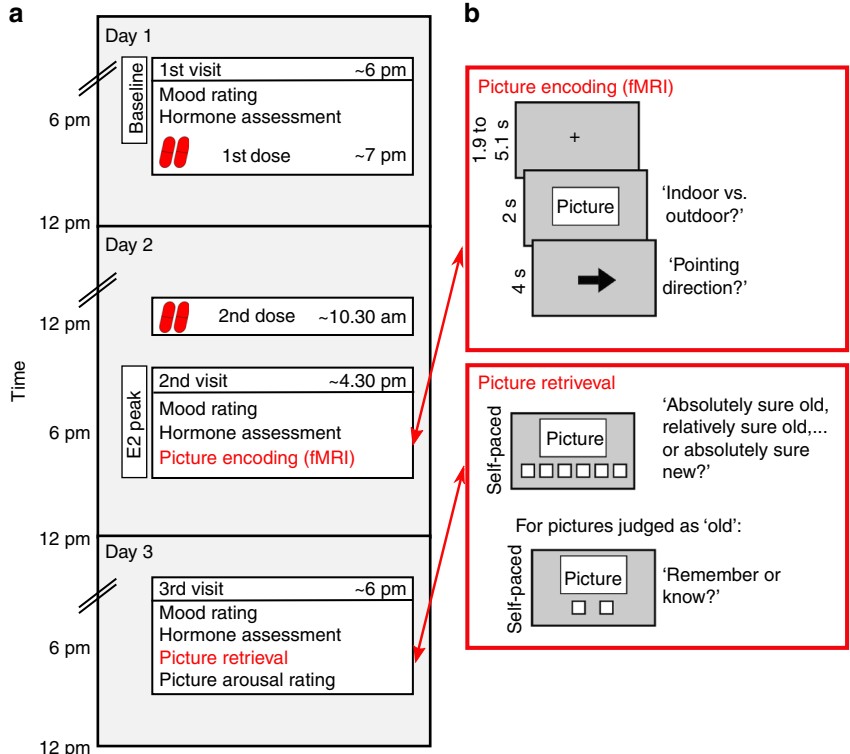

**Fig. 1** Study design. **a** Volunteers participated in the study on three consecutive days. On the evening of Day 1, baseline hormone concentrations and mood were assessed, followed by the first dose of either 17-beta-estradiol (E2; 2, 4, 6 or 12 mg) or placebo, depending on assigned group (double-blind). On the morning of Day 2, volunteers took the second dose on their own. In the afternoon of Day 2, when E2 levels were expected to peak, volunteers encoded emotional pictures inside the scanner (**b**, upper red box). On the evening of Day 3, volunteers performed a recognition test (**b**, lower red box) and rated the encoded pictures for arousal. See Supplementary Methods for more details on the timeline of the study

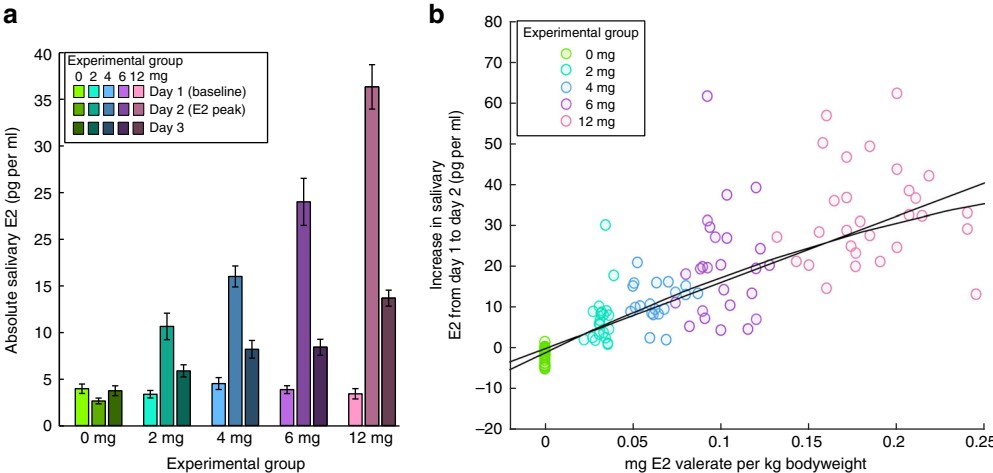

**Fig. 2** 17-beta-estradiol (E2) levels in saliva of the experimental groups ($N = 125$). **a** Absolute levels. Error bars represent standard errors of the mean. Confirming the effectiveness of the pharmacological manipulation, the increases in salivary and serum E2 levels from baseline (Day 1) to the expected peak (Day 2) differed as intended between experimental groups (saliva: $F(4, 47.98) = 102.84$, $p < .001$; serum: $F(4,31.24) = 87.54$, $p < .001$; see Supplementary Note 2 for tests of variance homogeneity). **b** Robust linear regression analyses between dose of E2 valerate given per kg bodyweight and salivary E2 increase ($p < .001$). A low Bayesian Information Criterion difference score ($\Delta$BIC; linear—quadratic model) of −1.36 suggests that the linear and quadratic model fit equally well. Colors represent experimental groups

E2 as a predictor (E2 on Day 2–E2 on Day 1). The Day 1 to Day 2 salivary E2 increase showed a high correlation with the Day 1 to Day 2 increase in serum E2 ($r(90) = .825$, $p < .001$), confirmed by linear robust regression analysis ($t(90) = 28.61$, $p < .001$, $\beta_{std} = 1.040$).

**Questionnaires and hormone values**. Please note that all analyses on hormone levels, questionnaires and behavioral performance were corrected for multiple comparisons using modification of the alpha criterion according to the Bonferroni method if necessary (denoted by $p_{corr}$).

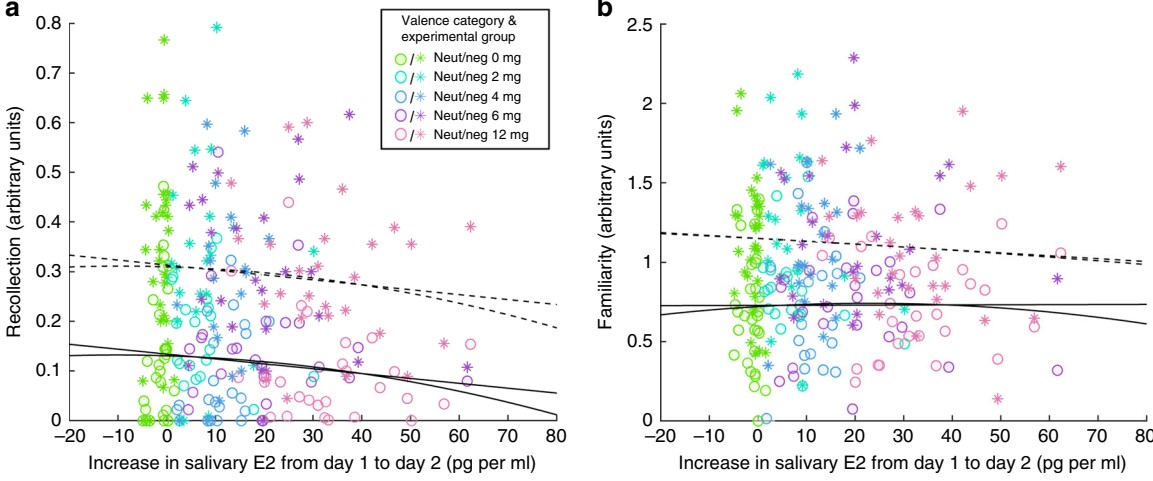

**Fig. 3** Regression between memory and E2 increases in salivary 17-beta-estradiol (E2; $N = 123$). Neither hippocampus-dependent memory (recollection; **a**) nor hippocampus-independent memory (familiarity; **b**) shows significant relationships to salivary Day 1 to Day 2 E2 increase. Colors represent experimental groups (all $p_{corr}$'s > .2)

Changes in physical complaint ratings from Day 1 to Day 2 did not show a linear or quadratic relationship with increases in E2 levels (all $p$'s > .2). According to principal component analyses on data from the mood questionnaire, none of the factors correlated significantly with E2 increases with the strongest non-significant relationship for the principal component which correlated with alertness, strength, energy, clear-headedness ($t(121) = 2.31$, $p_{corr} = .092$, $\beta_{std} = 0.188$; see Supplementary Table 3). We will refer to this principal component as 'subjective vigilance'. To test whether changes in subjective vigilance are reflected in an objective measure of alertness, individual reaction times (RT) from the distractor task (i.e., arrow task) and the encoding task were used as a proxy for intrinsic alertness. Neither reaction time (rt) nor accuracy in the arrow task (rt: $t(121) = -0.68$, $p_{corr} > .999$, $\beta_{std} = -0.011$; accuracy: $t(121) = -0.20$, $p_{corr} > .999$, $\beta_{std} = -0.004$) or the encoding task (indoor/outdoor decision) (rt: $t(121) = -0.57$, $p_{corr} > .999$, $\beta_{std} = -0.047$; accuracy: $t(121) = 0-.37$, $p_{corr} > .999$, $\beta_{std} = 0-.023$) were significantly related to the Day 1 to Day 2 increase in salivary E2.

Changes in salivary progesterone or cortisol did not yield a linear or quadratic relationship with Day 1 to Day 2 in E2 increases (all $p_{corr}$'s > .2; Supplementary Figure 1). Therefore, changes in progesterone or cortisol did not bias neural or behavioural effects of E2.

**Behavioral results**. Hit and correct rejection rates were above chance level (see Supplementary Note 3). In order to assess whether confidence ratings reflected recognition memory, paired $t$-tests were calculated on the confidence ratings ('absolutely sure old/new', 'relatively sure old/new', or 'unsure old/new'). Confidence was indeed significantly higher for correct than for incorrect responses (targets: $t(122) = 15.57$, $p < .001$; lures: $t(122) = 6.83$, $p < .001$). The employed a memory paradigm allowed us to disentangle hippocampus-independent ('familiarity' based) from hippocampus-dependent ('recollection' based) recognition memory[28], where 'recollection' refers to the retrieval of an item together with contextual information about the encoding episode in contrast to the acontextual sense of 'familiarity'. Recollection- and familiarity-based recognition memory were estimated from the confidence ratings using receiver-operating characteristic (ROC) curves according to the Dual Process model[28] (Supplementary Figure 2). Neither recollection, thought to be hippo-campus-dependent, nor familiarity was associated with E2 in a

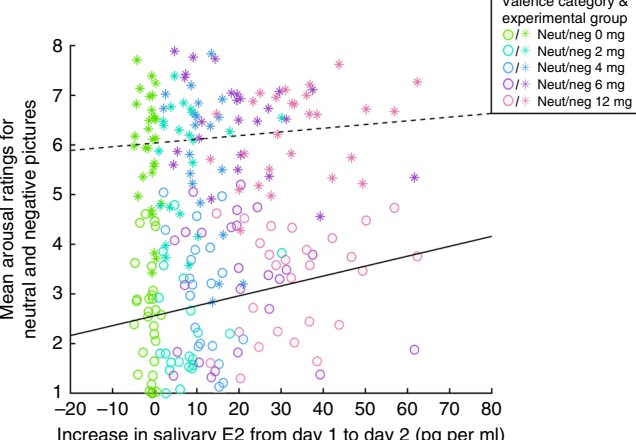

**Fig. 4** Linear regression between E2 increases and mean arousal ratings. Negative pictures are represented by asterisks, neutral pictures by circles. Colors represent experimental groups ($N = 123$). A significant positive linear relationship was observed between salivary E2 increase and arousal ratings for neutral pictures (solid line; $p_{corr} = .015$) but not negative pictures (dashed line; $p_{corr} = .524$)

linear or a quadratic manner (all $p_{corr}$'s > .2; Fig. 3; Supplementary Table 4). A similar pattern emerged when recollection and familiarity were computed on the remember/know judgements (all $p_{corr}$'s > .2). Neither d-prime, the EEM (=d-prime for negative pictures—d-prime for neutral pictures), nor response criteria showed a linear or quadratic relationship to the increase in salivary E2 (all $p_{corr}$'s > .3). See Supplementary Tables 4, 5, 6 and 7 for descriptive and inferential statistics of additional measures of recognition memory (e.g. hit rate, area under the curve, meta d-prime), confidence ratings and rt. None of these measures was significantly related to salivary Day 1 to Day 2 increase in E2 (all $p_{corr}$'s > .1).

Mean arousal ratings for neutral ($M = 2.86$, SD = 1.19) and negative images ($M = 6.05$, SD = 1.12) differed significantly ($t(122) = 31.45$, $p < .001$). Individual mean arousal ratings for negative pictures did not show either a linear or quadratic relationship with salivary E2 increases (all $p_{corr}$'s > .5). In contrast, arousal ratings for neutral pictures showed a positive linear but not a quadratic relationship with the Day 1 to Day 2 E2 increase

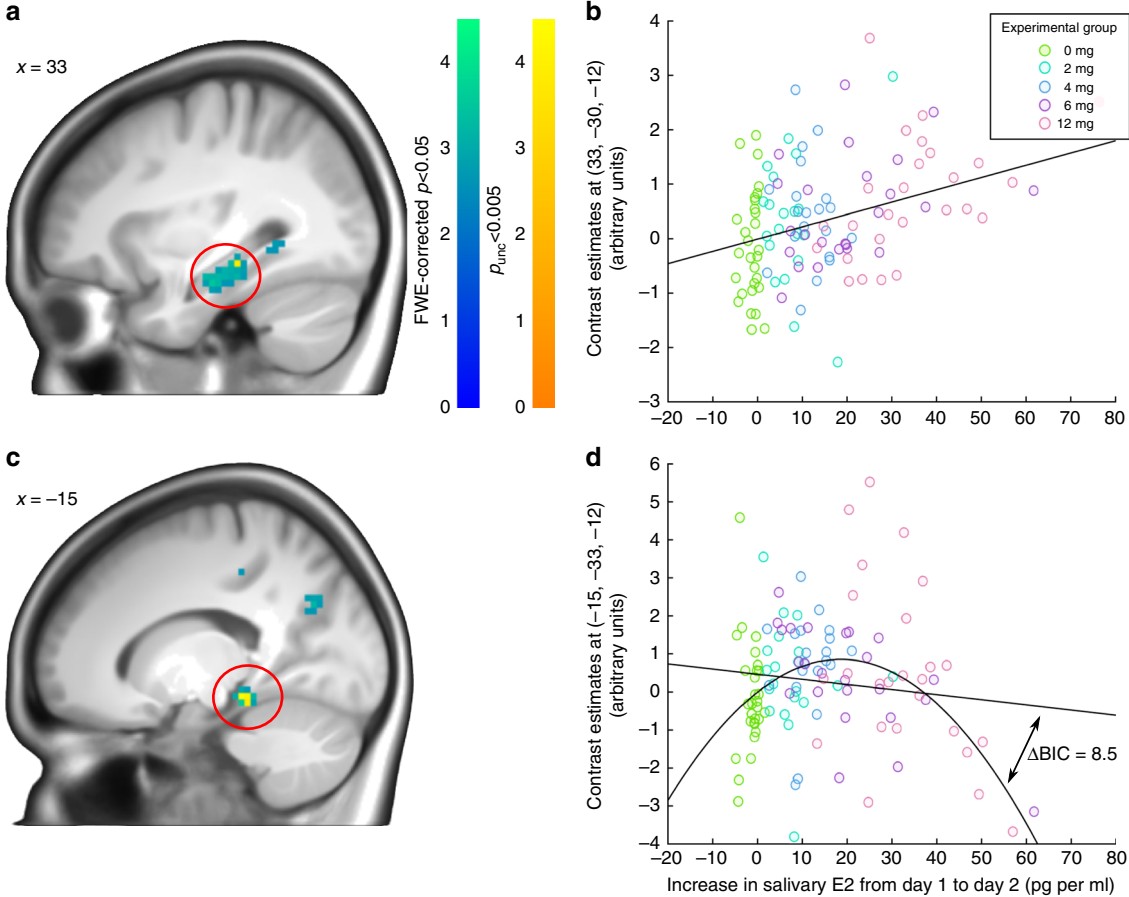

**Fig. 5** Relationship between 17-beta-estradiol (E2) increase and the remember >know contrast. **a** Statistical t-map of the positive linear relationship between salivary Day 1 to Day 2 increase in E2 and the remember > know contrast ($N = 118$). Statistical maps are thresholded at FWE-corrected $p < .05$ (warm color scale) and $p_{unc} < .005$ (cold color scale) for visualization purposes. A cluster in the right hippocampus showed a positive linear relationship with salivary E2 increase (FWE-corrected $p = .043$; $t$-test). **b** Robust regression analysis between contrast estimates (y-axis) and salivary E2 increase (x-axis) confirmed the positive linear relationship. Colors represent experimental groups. **c** Statistical t-map of the negative quadratic relationship between salivary Day 1 to Day 2 E2 increases and the remember > know contrast. A cluster in the left hippocampus showed an inverted U-shaped relationship with salivary E2 (FWE-corrected $p = .006$; $t$-test). **d** Robust regression analysis between contrast estimates (y-axis) and salivary E2 increase (x-axis) confirmed the inverted U-shaped relationship. A Bayesian Information Criterion difference ($\Delta$BIC) of 8.5 between linear and quadratic models strongly evidences a better model fit of the quadratic compared to the linear model

(linear: $t(121) = 2.72$, $p_{corr} = .015$, $\beta_{std} = 0.257$; quadratic: $t(121) = 1.51$, $p = .133$, $\beta_{std} = 0.366$; Fig. 4). The Day 1 to Day 3 increase in salivary E2 showed a similar pattern to the Day 1 to Day 2 increase with respect to arousal ratings, with a positive linear relationship being only significant for neutral pictures (neutral: $t(121) = 2.28$, $p_{corr} = .049$, $\beta_{std} = 0.219$; negative: $t(121) = -0.05$, $p_{corr} > .99$, $\beta_{std} = 0.267$). Although we have no explicit information on mood changes after picture encoding/arousal ratings, further analyses indicate that the increase in arousal ratings for neutral stimuli could be a side effect of an E2-related increase in negative mood induction over time (Supplementary Note 4). See Supplementary Table 8 for descriptive and inferential statistics for arousal ratings as a function of valence categories and responses in the recognition memory task.

**Neuroimaging results**. To rule out the possibility that differences in rCBF might have biased the relationship observed between E2 increase and memory task-related BOLD effects, we conducted whole brain regression analyses which revealed no positive linear or inverted U-shaped relationships between salivary E2 increase and rCBF. (lowest family-wise error (FWE)-corrected $p = .158$ found in the right amygdala, with peak at (16, −4, −20) in

Montreal Neurological Institute (MNI) space; $t$-test; $N = 117$; Supplementary Figure 3). In addition, more sensitive volume of interest analyses were performed from signal extracted from a 5-mm sphere around all peaks showing significant relationships with salivary E2 increase in the fMRI analyses (see below in this section). None of these analyses showed significant associations between E2-level effects on rCBF and the E2-level effects on BOLD activity.

To address our main research question of E2's effects on hippocampal activity, we compared activity during encoding of pictures that were subsequently recognized with recollection (remember-response) or familiarity (know-response). Confirming the sensitivity of the memory paradigm to detect subtle differences in hippocampal activity, the main effect associated with the remember > know contrast (i.e., irrespective of E2 increase) showed robust effects in the left and right hippocampus (left: (−30, −30, −18); $Z = 7.39$, FWE-corrected $p < .001$; right: (24, −9, −18); $Z = 5.58$, FWE-corrected $p < .001$; $t$-tests; $N = 118$; Supplementary Figure 4) and a wide network of brain areas (Supplementary Table 9). A linear regression analysis on the whole brain using salivary E2 increase as a predictor and the remember > know contrast estimates as the dependent variable, was then performed to identify areas showing linear relationships

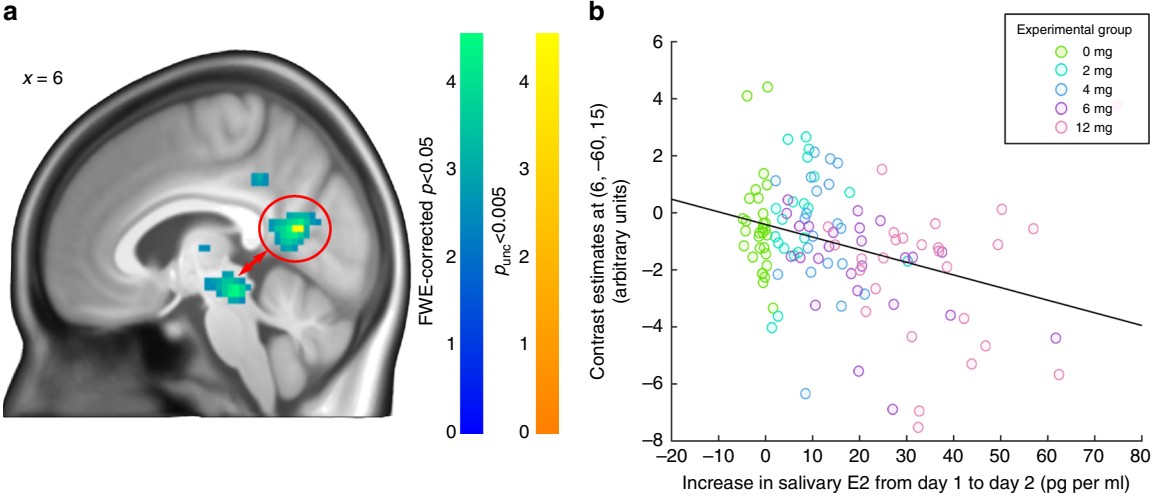

**Fig. 6** Relationship between E2 increase and the negative > neutral contrast. **a** Statistical t-map of the negative linear relationship between salivary Day 1 to Day 2 increase in 17-beta-estradiol (E2) and the negative > neutral contrast (N = 121). The statistical map is thresholded at FWE-corrected p < .05 (warm color scale) and $p_{unc}$ < .005 (cold color scale) for visualization purposes. Activity in a cluster in the right precuneus [circled in red; FWE-corrected p = .034; t-test] showed a negative linear relationship with salivary E2 increase. A negative linear relationship in the brainstem was not significant ([FWE-corrected p = .056; t-test]). A psycho-physiological interaction (PPI) analysis (seed: precuneus cluster, region of interest: brainstem cluster) revealed a negative linear relationship between E2 levels and precuneus-brainstem connectivity. **b** Robust regression analysis between contrast estimates (y-axis), extracted from the peak of the precuneus cluster, and salivary E2 levels (x-axis) confirmed the negative linear relationship. Colors represent experimental groups

between increases in E2 from Day 1 to Day 2 and hippocampal activity. Activity in a cluster in the right posterior hippocampus was positively linearly associated with E2 ((33, −30, −12); Z = 3.69, FWE-corrected p = .043; t-test; N = 118; Fig. 5 a, b). A quadratic regression analysis using the squared salivary E2 increase as a predictor and again the remember > know contrast estimates as the dependent variable, revealed an inverted U-shaped relationship between hippocampal activity and E2 increases in a more medial cluster within the left posterior hippocampus ((−15, −33, −12); Z = 4.22, FWE-corrected p = .006; t-test; N = 118; Fig. 5 c, d). Additional analyses showed that these relationships between salivary E2 increase and brain activity were specific to hippocampal regions (see Supplementary Note 5). Finally, exploratory psycho-physiological interaction (PPI)[29] analyses did not show any effect of salivary E2 increase on the functional coupling of either of the two posterior hippocampal clusters (peaks at ((33, −30, −12) and (−15, −33, −12), respectively) with other brain regions for remember compared to know trials (see Supplementary Note 6). Brain activity linked to general encoding success (subsequent hits > misses; see Supplementary Table 10 for main effects) neither showed a linear nor quadratic relationship in the hippocampus or the amygdala (lowest FWE-corrected p of .222 at (27, −12, −18); t-tests; N = 121).

To address the second research question of estrogenic effects on amygdala activity related to emotional processing, brain activity during the encoding of negative pictures was compared with neutral pictures, without considering encoding success. Confirming the sensitivity of this contrast to detect subtle differences in amygdala activity, robust effects were observed in the bilateral amygdala (left: (−24, −3, −21); Z = 11.00, FWE-corrected p < .001; right: (24, −6, −15); Z = 9.80, FWE-corrected p < .001; t-tests; N = 121; Supplementary Figure 4) and a wide network of brain areas. Again, linear and quadratic regression models were fitted to whole brain activity in order to identify areas that might show linear or inverted U-shaped relationships between E2 and neural activity related to affective processing. The models were estimated in SPM using the negative > neutral whole brain contrast images as the dependent variable and the increase

in E2 from Day 1 to Day 2 as the independent variable. No relationship, either linear or quadratic, was found between E2 increases and emotional processing activity in the amygdala (lowest FWE-corrected p of .935 at (−30, −6, −15); t-tests; N = 121). Although not predicted, there was a negative linear relationship between E2 levels and differences in brain activity for negative compared to neutral pictures in the right precuneus ((6, −60, 15), Z = 4.70, FWE-corrected p = .037; t-test; N = 121), which survived FWE correction for multiple comparisons for the whole scan volume. There was no significant linear relationship in the right brainstem surviving FWE correction ((6, −30, −12), Z = 4.60, FWE-corrected p = .056; t-test; N = 121; Fig. 6).

To explore the effects of salivary E2 increase on the functional coupling between precuneus and brainstem for the processing differences between negative and neutral pictures, a PPI analysis was conducted. For this purpose, we created mask images from the clusters in the precuneus (peak: (6, −60, 15)) and the brainstem (peak: (6, −30, −12)) which showed a negative relationship with salivary E2 increase thresholded at $p_{unc}$ < .001. The connectivity analysis (seed: precuneus cluster; region of interest: brainstem cluster) revealed a negative linear relationship between E2 increases and precuneus-brainstem connectivity ((6, −21, −9), Z = 2.94, FWE-corrected p = .038; t-test; N = 118). In other words, the greater the E2 increase the smaller the brain activity difference between encoding negative and neutral pictures as well as the less coupled the activity between the precuneus and the brainstem.

Finally, to test for an interaction of emotional processing and hippocampus-dependent memory formation, we calculated the emotional enhancement of memory (EEM) through computing the interaction between encoding success and picture valence, i.e., valence (negative > neutral) × encoding success (subsequent hits > misses). Confirming that the EEM effect evokes activity within the regions of interest (ROIs), main effects were observed bilaterally in the hippocampus (left: (−21, −9, −18), Z = 4.09, $p_{corr}$ = .009; right: (18, −12, −9), Z = 3.88, FWE-corrected p = .021; t-tests; N = 121) and the amygdala (left: (−21, −3, −21), Z = 6.86, FWE-corrected p < .001; right: (24, −3, −21), Z = 6.15, FWE-corrected p < .001; t-tests; N = 121;

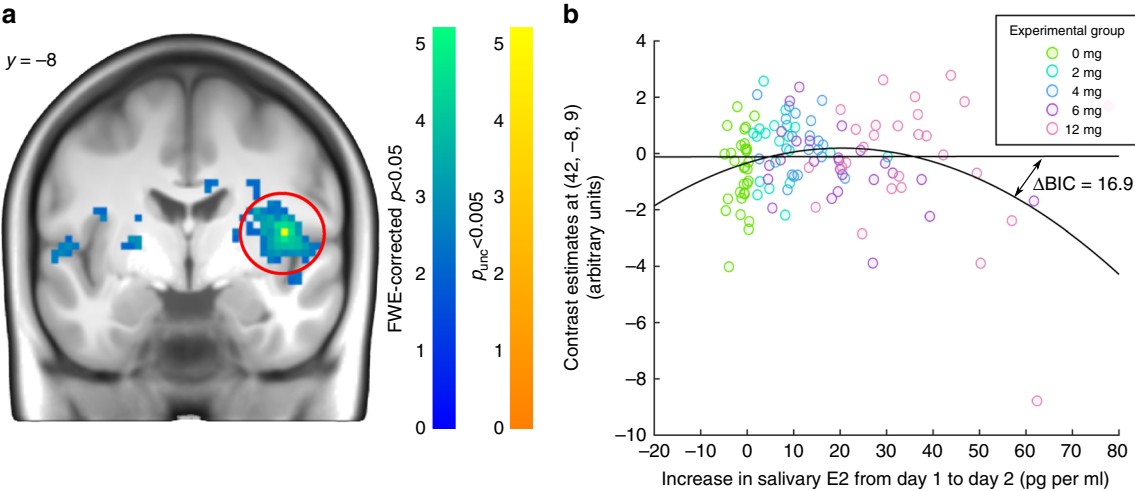

**Fig. 7** Relationship between E2 increase and the emotional enhancement of memory contrast. **a** Statistical t-map of the negative quadratic relationship between salivary Day 1 to Day 2 increase in 17-beta-estradiol (E2) and the emotional enhancement of memory (EEM; hit > miss × negative > neutral) contrast in the right insula ($N = 121$; FWE-corrected $p = .015$; t-test). The statistical map is thresholded at FWE-corrected $p < .05$ (warm color scale) and $p_{unc} < .005$ (cold color scale) for visualization purposes. **b** Robust regression analysis between contrast estimates (y-axis) and salivary E2 increase (x-axis) confirmed the inverted U-shaped relationship. Difference in Bayesian Information Criterion (ΔBIC) between the linear and quadratic model was 16.9, strongly evidencing a better model fit of the quadratic compared to the linear model. Colors represent experimental groups

Supplementary Figure 4). However, neither a significant linear nor quadratic relationships between E2 increases and hippocampal or amygdala activity related to the EEM effect was observed in independent whole brain SPM regression analyses (lowest FWE-corrected $p$ of .487 at (27, −3, −9); t-tests; $N = 121$). A cluster in the right insula showed an inverted U-shaped relationship with E2 surviving FWE correction for the whole scan volume ((42, −8, 9), $Z = 4.93$, FWE-corrected $p = .015$; t-tests; $N = 121$; Fig. 7).

Robust regression analyses using betas extracted from peak voxels confirmed that the observed relationships were not driven by outliers (Supplementary Table 11, also for cluster extents). Results of the fMRI analyses using a more lenient statistical threshold of $p < .001$ uncorrected for multiple comparisons are reported for exploratory reasons in Supplementary Table 12.

## Discussion

We identified different patterns in which hippocampal activity responded to E2 dose: we found an inverted U-shape in the medial posterior hippocampus but a monotonically increasing dose-response function across the full range of E2 levels in the lateral posterior hippocampus. While E2 did not modulate amygdala activity even at high levels, we observed an unpredicted monotonically decreasing relationship for activity associated with affective processing in the precuneus and for the functional coupling between precuneus and brainstem. Finally, activity in the insula exhibited an inverted U-shaped relationship with E2. Critically, the strictly data-driven analyses showed that the inflection point of the quadratic function in both the hippocampus and the insula were clearly above the physiological levels of E2 reached during the menstrual cycle[26]. Our results therefore support the hypothesis that the effect of E2 on hippocampal neural activity follows an inverted U-shaped dose response curve only at supraphysiological levels but increases in E2 are beneficial within the levels experienced regularly during the menstrual cycle.

The BOLD response measured in fMRI mainly reflects glutamatergic synaptic activity, with greater glutamate release at individual synapses and higher synapse density leading to greater BOLD signal[30–32]. The inverted U-shaped BOLD response in the

medial posterior hippocampus and insula might be explained by E2's effects on cellular processes, which appear intricate because different estrogen receptors have unique characteristics. E2 can regulate the transcription of many genes by activating intracellular estrogen receptors α and β (ERα and ERβ) within a few hours, but also triggers rapid effects through membrane-bound ERα, ERβ and G-protein coupled ER 1[1,4,5,13]. ERα activation stimulates whereas ERβ decreases synaptogenesis, where a decrease reduces overall glutamate release and hence the BOLD response. Low levels of E2 will mostly activate the high-affinity receptor ERα (activating synaptogenesis and thus accounting for a linearly increasing BOLD response), whereas higher E2 levels activate the more abundantly expressed low-affinity receptor ERβ[33–36] (decreasing synaptogenesis, thus, causing a return of plasticity to baseline, i.e., downward inflection BOLD in the response). The amount of synaptogenesis may additionally depend on the relative expression of ERα/ERβ, where E2 upregulates ERα expression[4]. It has been shown that only supraphysiological E2 levels downregulates ERα expression[4,37], consistent with our findings of U-shaped BOLD responses only at supraphysiological levels. Besides these genomic effects, E2 also activates membrane-bound ERβ of inhibitory hippocampal interneurons[9], reducing glutamate release and thereby also the BOLD response. This account would thus explain how the inverted U-shaped BOLD response can be the result of E2's rapid and/or genomic effects on hippocampal plasticity. Furthermore, given that our study induced a continuous range of E2 levels, this would also offer a coherent account which unifies our current systems-level results with those from cellular studies.

Given these intricate patterns, at first sight it might seem surprising that we also observe a monotonically increasing BOLD response, not only an inverted U-shaped response, and that these two types of response curves were observed in the hippocampus, albeit in distinct clusters. This is however consistent with the animals studies mentioned in the Introduction, which have observed monotonic increases in cellular measures of hippocampal plasticity also at supraphysiological levels of E2[9,14–16]. Moreover, two animal studies have described dissociated dose-response functions for different hippocampal subregions before[8,9]. Such region-specific dose-response functions could be a

result of the reported differences in ERα/ERβ ratios[8,34,38–41]. A higher ERα/ERβ ratio, as reported for instance in the CA3 subregion of the hippocampus compared to CA1[39], could result in a shift of the inflection point of the dose-response curve to even higher, supraphysiological E2 levels that were above the range induced in the present study. This would explain why the E2-BOLD signal relationship continues to even increase at supraphysiological E2 levels in the more medial posterior hippocampus. Further plausible explanations for the observed responses could lie in the differences in how E2 regulates ERα transcription in CA1 and CA3[39], or subfield-specific differences in nuclear E2 uptake[42]. However, the spatial resolution of our fMRI scans is too coarse to relate the different dose-response curves to specific anatomically defined hippocampal subfields and only permits a functional dissociation.

Many animal studies have observed that E2 affects not only for cellular measures of hippocampal neuronal plasticity but also hippocampus-dependent memory performance[1–3]. Given that we used a paradigm which we know elicits menstrual-cycle-dependent variations in hippocampus-dependent memory[19], at least for negative stimuli, the null effect of E2 on memory performance in the current study seems to be particularly surprising. However, despite the many studies providing evidence for a positive relationship between hippocampus-dependent memory and E2 levels during learning within a physiological range in female animals[1,3], even these effects appear to depend highly on dose, treatment duration and, in particular, the specific task used[43–47]. We also observed highly task-specific behavioral effects of E2 in two of our previous studies in women, where hormonal effects were restricted to hippocampus-dependent memory for negatively valenced stimuli or a specific form of hippocampus-dependent associative verbal memory[18,19]. Similarly, although only studied in young men, a genetic polymorphism related to higher E2 levels was associated with larger posterior hippocampi but not with better hippocampus-dependent memory performance in three different tasks[46,48]. One might speculate that subtle differences in the neural circuits involved in specific hippocampus-dependent tasks, in combination with the influence of a (sub)region's specific ERα/ERβ distribution might lead to a complex interaction between task performance and dose. It therefore seems plausible that a different, more sensitive task with the current E2 regimen could elicit observable behavioral changes.

E2 appears to modulate affective processing, as is suggested by the positive correlation between E2 dose and subjective arousal ratings for neutral pictures. The absence of a significant correlation with negative pictures is likely related to ceiling effects, since negative pictures elicited high arousal ratings even at baseline E2 levels[49]. The increase of arousal for neutral stimuli with increased E2 levels might be caused by a more pronounced induction of negative mood in response to the negative pictures with higher E2 levels, consistent with our supplementary analyses and with animal data showing an increase of depression-like behavior and anxiety at higher E2 levels[50,51].

Amygdala activity associated with emotional processing was surprisingly not modulated by E2 despite the robust task-elicited activity in the amygdala. Given the well-controlled increase in this highly estrogen-sensitive sample across a wide range of E2 levels, the absence of such an effect could suggest that the previously reported variation of amygdala activity across the menstrual cycle should be attributed instead to progesterone[52]. The discrepancy between the current study and the established effects of E2 on amygdala functions in animals[6,50,51,53] is likely caused by the different tasks used that rely on different amygdala nuclei. For instance, fear extinction learning, which is widely used in animal research, is modulated by E2 via the central amygdala[54] while the effect of emotional arousal on episodic memory formation

implicates the basolateral amygdala[55]. Therefore, our data are consistent with the neuroanatomically specific effects of E2 on different nuclei of the amygdala[6].

Neuronal activity linked to emotional processing was monotonically lowered by E2 in the posterior precuneus and E2 decreased the connectivity between precuneus and a cluster in the brainstem. Animal research has not focused on the cellular actions of E2 in these two areas, although they express ERs[40,56], so we had no predictions on how E2 would modulate these areas during affective processing. However, such a modulation is coherent with previous studies in humans. Both areas have been implicated in affective processing, among other processes, and are anatomically as well as functionally connected[57,58]. Activity in these regions is known to be modulated by the menstrual cycle or HRT[59,60]. The monotonically decreasing dose-response function might suggest a lower ERα/ERβ-ratio than in the hippocampus. The observation that subjectively reported emotional arousal increased is consistent with the precuneus having a down-regulating role during affective processing[61].

Finally, the interaction between hippocampus-dependent memory and amygdala-dependent processing of emotional arousal, i.e., EEM effect, did not show a monotonic or inverted U-shaped relationship with E2 on behavior nor for activity in the hippocampus or amygdala. Interestingly, an inverted U-shaped relationship was observed with activity in the posterior insula, a region that has been implicated in mediating the effect of emotional arousal on memory formation[78]. In fact, both ER subtypes have been detected there and HRT modulates activity and cerebral blood flow in the insula[62–64]. Again, based on our current results one might speculate that the menstrual-cycle-dependent variations in EEM-related activity in the hippocampus and the amygdala observed in our previous study might be accounted for by fluctuations in progesterone[19].

Of note, we did not observe an overlap between effects of E2 on any BOLD activity elicited by the emotional memory task and the subtle differences in brain perfusion during rest as measured with ASL. This implies that E2 effects on brain activity, as measured with fMRI, in the current study are not confounded by general task-unrelated effects of E2 on brain perfusion. Moreover, the effects of E2 on memory and affective processing-related BOLD activity were specific to a restricted set of brain regions, i.e., the hippocampus, the precuneus and insula, although the task evoked activity in wide networks of areas. This further reinforces our conclusions that E2 had no general effect on the neurovascular coupling underlying the BOLD effect but that the observed differences in the BOLD signal indeed reflect E2 effects on synaptic activity.

Despite the well-controlled E2 treatment regimen and carefully chosen experimental design, the current study faces some limitations. As with most animal studies on which we based our hypotheses, we could not measure hippocampal but only serum and saliva E2 levels. On the other hand, hippocampal and ovarian E2 syntheses are synchronized by the gonadotropin-releasing hormone so that peripheral and central levels of E2 are highly correlated[65,66]. However, it is unclear whether these synchronizing feedback mechanisms also function with exogenously increased E2 levels. Therefore, our conclusion that the response function of E2 turns into an inverted U-shape only at supraphysiological levels can only be applied to peripheral E2 levels until this interaction is clarified. Second, although being one of the major strengths of the current study, investigating isolated E2 effects in intact women has the drawback that it is currently unknown how these findings translate to phases of the menstrual cycle when fluctuations in E2 and P4 co-occur and to findings with ovariectomized animals. Nonetheless, similar neural activity on the cellular level has been observed in intact and

ovariectomized rats treated with physiological E2 doses[9]. Third, animal studies sometimes involve higher supraphysiological E2 doses which one would not give to human volunteers because of potential health risks (e.g., thrombosis). As such, we cannot make any statements on the relationship between salivary E2 levels and hippocampal activation for such high supraphysiological doses. Moreover, the BOLD effect approximately reflects the net amount of local glutamatergic synaptic activity, which can be the sum of distinct, even opposing, cellular processes affected by E2 (e.g. E2 modulates inhibitory interneurons). Finally, current results should not be overgeneralized as evidence against an effect of E2 on memory in humans, as the null behavioral effect could, for instance, be specific to the paradigm employed, which differs from paradigms used in animal research.

In conclusion, under tight experimental control in young, healthy women, we showed that the pharmacological increase of a wide range of physiological and supraphysiological E2 levels modulates neuralactivity differently in distinct hippocampal clusters. In particular, we observed an inverted U-shape response in a medial posterior hippocampal cluster, that is, the effect of E2 was beneficial with respect to neural activity within physiological E2 levels but returned to baseline only at supraphysiological levels. In a more medial posterior hippocampus cluster, however, activity monotonically increased across the full range of E2 levels. The fact that we did not observe any effect of E2 on hippocampus-dependent memory performance in this rigorously controlled study suggests that although effects on neural activity during encoding were visible 24 h after first E2 intake, this time span was too short to also manifest in behavioral changes after a consolidation period of 24 h in the current paradigm. Of interest is also that E2 did not modulate amygdala activity during affective processing and memory formation, implicating that previous reports of such effects in menstrual cycle studies should be attributed to progesterone instead. Finally, although the task activated a network of brain areas that have not been focused on by animal research yet, activity in only a small number of these areas were found to be modulated by E2.

## Methods

**Volunteers**. One hundred and twenty-five healthy, naturally cycling female volunteers aged 18 to 35 years ($M = 26$, $SD = 4$) participated in the current study (see Supplementary Methods for sample size calculation). All subjects were right-handed. They reported to be free of psychiatric illnesses, not to be users of illicit drugs or on central nervous system medication, and did not smoke on a regular basis. They had no contraindications for taking E2 (e.g., obese, at risk for cardiovascular problems) or for MR examinations (e.g., pregnant). Only women who had not taken any oral contraceptives or were pregnant in the 6 months prior to the study were included. Menstrual cycle length ($M = 30$, $SD = 5$), based on the reported dates of last menstruation, was used to determine adequate time points for testing.

Volunteers were randomly assigned to receive either placebo (PL; mannitol and highly dispersed silicon dioxide) or E2 valerate in doses of 2, 4, 6 or 12 mg (from Progynova 21 UTA, Schering, Germany). All pills were in the form of two identical capsules taken orally. E2 valerate is the synthetic ester of natural E2, with an average $t_{max}$ of approximately 3–6 h and a half-life of 14 h[67,68]. Two days of E2 intake were chosen in order to maintain E2 at elevated levels for a time period of several hours.

All volunteers visited the institute on three consecutive days. The first day was matched as closely as possible to menstruation onset ($M = 1$ day, $SD = 4$ days after menstruation onset). The first dose of E2 was administered double-blind by the experimenter in the evening of day 1 ($M = 15.45$ h before second dose on Day 2, $SD = 1.37$ h). Volunteers took the second dose on their own the next morning of Day 2 ($M = 5.71$ h before testing on Day 2, $SD = 0.90$ h). On all testing days, volunteers rated their mood on a standardized questionnaire and gave saliva samples (details below in this section). Blood samples were drawn only on Day 1 and Day 2 from a subgroup of the study sample ($n = 92$), because not all women were willing to give blood, or blood could not be drawn within the first three attempts. Memory encoding took place after 2 pm on Day 2 inside the scanner when E2 levels peaked[67]. The memory retrieval test was conducted after 3:30 pm on Day 3 (see Fig. 1 for a schematic overview of the experimental design). All experimenters were female to avoid gender stereotype effects. Two volunteers had

to be excluded from analysis due to technical problems (see Supplementary Table 13 for a full list of volunteers included in each analysis).

Volunteers received financial compensation of €120 for their time. All volunteers gave written informed consent according to the Declaration of Helsinki. Ethics approval was obtained from the Ethics Committee of the Hamburg Medical Association (PV3612).

**Emotional memory task**. A total of 360 color pictures depicting emotionally neutral and negative content were used for the study. Pictures were drawn from the International Affective Picture Set (IAPS)[69] and from the Internet. We ensured an equal number of images showing animals and people in the two emotional valence categories. Based on arousal ratings from a previous study[19], we created 90 negative and 90 neutral picture pairs matched for level of arousal. The pictures of each pair were pseudo-randomly assigned as a target or a lure for each volunteer.

All volunteers encoded 180 of the 360 pictures ($480 \times 720$ pixels; $11° \times 16°$) stimuli equally split over five runs inside the MR scanner (i.e., 36 pictures per block). All stimuli were shown only once during encoding. Stimulus order were pseudo-randomized with the restriction that no more than three pictures of the same valence category were presented in a row. Volunteers were informed about the memory test prior to encoding.

Each trial began with a fixation cross lasting for a jittered duration of 1.9 to 5.1 s, followed by presentation of the picture to be encoded for 2 s. During picture presentation, volunteers had to indicate whether the picture depicted an indoor or outdoor scene. Then volunteers performed a distraction task to prevent rehearsal and emotional carry-over effects by prolonging ISIs while still maintaining volunteers's attention throughout the task. In this task, an arrow pointing either to the left or the right was presented for 800 ms in the middle of the screen. Volunteers were asked to indicate the direction of the arrow as fast as possible by a button press. After an inter-stimulus interval of 200 ms, the next arrow was presented. Total trial length ranged from 7.9 to 11.1 s.

During retrieval, volunteers saw the 180 pictures seen during encoding randomly intermixed with the 180 unseen pictures ($480 \times 720$ pixels; $11° \times 16°$). While viewing each picture individually on the screen, volunteers were first asked to indicate on a 6-point confidence scale, ranging from 'very sure new' to 'very sure old,' whether the image was presented during the encoding task and how sure they were in their response (Fig. 1b). For pictures judged as 'old', volunteers were then asked to indicate whether they remembered contextual details from encoding ('remember') or just 'knew' that the image was seen before. Boxes corresponding to the six levels of the confidence scale and the remember/know options were presented below the picture until a response was given. These confidence ratings were used to estimate the contribution of recollection and familiarity to recognition memory. The Dual Process model of recognition memory was fitted to the ROC curves, which were derived from the hit and false alarm rates across confidence levels (Supplementary Figure 2) using maximum likelihood estimation[28,70]. This was done separately for neutral and negative items and performed in Matlab R2014b (Mathworks, Inc, Natick, MA, USA). For exploratory reasons, confidence ratings were additionally used to estimate meta d-prime and meta criterion (see Supplementary Table 4). The remember/know procedure (see Supplementary Methods for instructions) was used to back sort each encoding trial as to whether they were subsequently retrieved with contextual details (hippocampus-dependent 'recollection') or only with a sense of familiarity (hippocampus-independent 'know'). Estimates for recollection and familiarity were also based on remember/know responses using the formulas provided by Martin and colleagues[71] for completeness. For behavioral and fMRI analyses based on remember/know responses, three additional volunteers had to be excluded because of missing data (two excluded from 6- and 12-mg groups for not having responded 'know' for any neutral picture; one excluded from the placebo group for no 'remember' responses; see Supplementary Table 13 for a full list of volunteers included in each analysis).

After retrieval, volunteers rated all 180 target stimuli on subjective emotional arousal. They were instructed to rate the arousal they experienced when viewing the pictures for the first time (i.e., during encoding on Day 2). For this purpose, pictures were shown in the same order and with the same duration as during encoding (2 s). After each picture was presented for 2 s in the middle of the screen, volunteers were presented a 9-point pictorial Self-Assessment-Manikin (SAM) scale on which they had to rate emotional arousal of the pictures ranging from maximally calming to maximally arousing [93]. Arousal ratings were self-spaced.

**Hormone concentrations**. On each test day, three saliva samples were collected 20–30 min apart and subsequently pooled to achieve reliable measures (~3 ml in total). Serum (~1 ml) was separated from plasma by allowing the blood sample to rest for at least 30 min, to allow sample to clot spontaneously, before centrifuging at 2000 RPM for 10 min. Saliva and serum samples were stored at −18 °C until analysis by IBL (Hamburg, GER) using highly sensitive luminescence assays for salivary E2, P4 and cortisol (sensitivity E2: 0.3 pg per ml, P4: 2.6 pg per ml, Cortisol: 0.06 ng per ml) and ELISA for serum E2 levels (sensitivity: 10.6 pg per ml).

**Mood ratings**. In order to examine potential effects of E2 on mood, a questionnaire consisting of 16 pairs of opposing feelings was used[72]. Because E2 is also known to affect anxiety, we included one question for anxiety[51]. Every word pair

was presented one word at a time on the screen, and volunteers were asked to rate how they felt at the moment by moving a cursor on a continuous scale.

To reduce dimensions, a principal component analysis (PCA; varimax rotation) was conducted using SPSS Version 18 for Windows (SPSS, Inc., Chicago, IL, USA). The Kaiser-Meyer-Olkin measure of sampling adequacy (KMO = .804) and the Bartlett test of sphericity ($\chi^2(136, N = 123) = 812.656$, $p < .001$) indicated the appropriateness of performing PCA[73]. The analysis yielded four principal components. Please see Supplementary Table 3 for initial dimensions and component loadings.

**Side effects and drug guessing**. To control for potential confounds due to variations in menstruation-related physical discomforts or side effects of E2 pill intake, volunteers had to report the presence of physical complaints ('headache', 'abdominal pain', 'breast tenderness', 'nausea', 'vertigo', 'backpain') using a 6-point Likert scale on each testing day. On Day 2 and Day 3, volunteers were additionally asked whether they experienced any change attributable to the pill intake. To also rule out the possibility that drug guessing might have biased study results, volunteers were asked whether they believed that they received E2 or placebo after testing on Day 3. A total physical complaint score was calculated for each day.

**Statistical analyses of behavioral data**. Testing for differences between groups was conducted by means of univariate ANOVAs for continuous variables (e.g., age or E2 levels) and non-parametric Chi-Squared Tests for categorical variables (e.g., drug guessing and side effects). With the exception for memory test scores and arousal ratings, for which we only had one time point, all statistical analyses were conducted on difference scores (Day 2 minus Day 1). Linear and inverted U-shaped responses to E2 increases from Day 1 to Day 2 ($E2_{Day2-Day1}$) were tested using a hierarchical robust regression approach run in MATLAB 2014b (Mathworks, Inc, Natick, MA, USA; fitlm function, robust fitting option). This approach is robust to potential outliers and violations of the normality assumption. Two regression models were implemented for every variable of interest: the first including a linear term only ($y = a + b^* E2_{Day2-Day1}$), the second including the linear plus a quadratic term ($y = a + b_1^* E2_{Day2-Day1} + b_2^* E2_{Day2-Day1}^2$). To rule out the possibility that changes in other measures might bias hypothesis testing, variables of no interest (e.g., changes in mood) were used as predictors in separate regression models. Statistical results were considered significant at a threshold of $p < .05$.

**Neuroimaging data**. Event-related functional MRI was performed on a 3 Tesla scanner (Siemens Trio) with an echo planar imaging T2*-weighted sequence in 38 contiguous axial slices (3-mm thickness with 1-mm gap; TR 2.21 s; TE 30 ms; flip angle 80°; field of view 216 × 216; matrix 72 × 72). For spatial normalization, a high-resolution T1-weighted structural MR image was acquired by using a 3D-MPRAGE sequence (TR 2300 ms, TE 2.89 ms, flip angle 9°, 1-mm slices, field of view 256 × 192; 240 slices).

Event-related fMRI data were preprocessed and analyzed using Statistical Parametric Mapping (SPM12; Wellcome Department of Imaging Neuroscience, London, UK) run in Matlab R2014b. To prevent biases due to spin saturation, the first five functional images were discarded. All functional images were slice-time corrected. Next, to correct for susceptibility-by-movement artifacts, all functional images were realigned and unwarped (as implemented in SPM12). Individual structural T1 images were then coregistered to functional images. Coregistered T1 images were segmented into gray and white matter, which were subsequently used within the 'diffeomorphic anatomic registration through an exponentiated lie algebra algorithm' (DARTEL) toolbox to create structural templates and individual flow fields. The latter was used for normalizing structural and functional images to MNI space. Finally, images were smoothed with a full-width half maximum Gaussian kernel of 8 mm in all spatial directions. We excluded two volunteers from fMRI data analyses due to technical failure and/or excessive head movement (see Supplementary Table 13 for a full list of volunteers included in each analysis).

Event-related BOLD responses were analyzed in a general linear model as implemented in SPM using a mass univariate approach. Subject-level models included six separate regressors, convolved with the canonical hemodynamic response function, for the factors valence (neutral, negative) and memory (hit remember, hit know, miss). To investigate the effects of E2 on memory and affective processing, the following four contrasts, defined a priori, were calculated for each volunteer: 1. A remember/know contrast (remember > know) to specifically examine hippocampus-dependent memory; 2. General encoding success (subsequent hit > miss); 3. The effects of emotional processing (negative > neutral); 4. EEM (valence (negative > neutral) × general encoding success (hit > miss)).

Each individual's contrast images were used for second-level group analyses with subject as a random factor. To test whether the planned contrasts were associated with activity in the areas of primary interest, i.e., the hippocampus and amygdala, and other wide network of brain areas, the main effects for remember/know, general encoding success, emotional processing, and EEM contrasts were estimated. To identify brain regions showing linear or quadratic effects with E2 levels, additional linear only ($y = a + b^* E2_{Day2-Day1}$) and quadratic regression ($y = a + b_1^* E2_{Day2-Day1} + b_2^* E2_{Day2-Day1}^2$) models were implemented for each

contrast of interest (remember/know, general encoding success, emotional processing, and EEM) on the group level. As we were interested in the inverted U-shaped dose-response function, only the negative contrast of the quadratic term was estimated. Results of all analyses, i.e., the main effects as well as the models testing for E2 relationships, were considered significant at $p < .05$, family-wise error corrected for multiple comparisons within anatomical ROIs defined a priori, namely the hippocampus and the amygdala. In addition, results outside of these ROIs were considered significant at $p < .05$, family-wise error corrected for multiple comparisons on the entire scan volume. The hippocampus and amygdala masks were created using the Anatomy toolbox[74,75].

To verify that resulting relationships are not biased by outliers or violations of the normality assumption, beta values were extracted from peaks in clusters surviving multiple comparison correction and used for the robust regression analyses explained above in this section. As long as an appropriate multiple comparison correction is applied, calculating more elaborate statistical models, in particular in confirmatory analyses, using extracted beta values is statistically sound[76].

Please see Supplementary Methods for the rationale behind using ASL and analysis details.

**Data availability**. Data that support the findings of this study have been deposited online under the following links https://www.neurovault.org/collections/3541/ and https://figshare.com/s/cb29874025f15ae21249 (https://doi.org/10.6084/m9.figshare.5932810).

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

## Acknowledgements

We thank H. Kuhl, A. O. Mück, and J. Höchel for pharmacokinetic advice. We thank G. Rune for helpful discussions and G. Joue for her careful proofreading and language editing. The project was supported by a grant of the German Research Foundation (DFG SO 952/6-1). J.G. was funded by SFB TRR 169 'Crossmodal Learning' and the BMBF grant 01GQ1003.

## Author contributions

J.B. and T.S. designed the experiments with support of J.G., J.F., and L.S. J.B. and L.S. performed the experiments. J.B., J.F., and T.S. analyzed the data. J.B. and T.S. wrote the paper. All authors discussed the results and commented on the manuscript at all stages.

## Additional information

**Competing interests:** The authors declare no competing interests.

