## [Peer Review File(PDF 2267 kb) · Nature Communications]

Reviewers' comments:

Reviewer #1 (Remarks to the Author):

REVIEW

Manuscript Number: NCOMMS-17-00194

Title: Linear and inverted u-shaped dose-response functions of estrogen on memory-related hippocampal activity in healthy young women

- This is an ambitious and well run study examining E2 effects on memory in healthy women. The methodology is solid, the memory task has been used before, and they provide some interesting data. However, enthusiasm for the paper is diminished due to the fact that the paper seems largely empty of theory and hypotheses, and they report no correlation between hippocampal-dependent memory and E2 levels across a number of different analyses. Furthermore, the role of this hippocampus in this task is questionable and certainly is not translational to the nonhuman literature. Globally, it is unclear how these results contribute to our understanding of E2 and hippocampus function.

- Specific Comments.

- o The Introduction is thorough, but it does not offer any theory about what the authors expect to see. In fact, this lack of theory is a recurring theme throughout the paper. To say that phenomenon have been shown to have a linear function, a quadratic function, and that things sometimes show an inverted U function offers little information about E2 or the hippocampus. For example, if after giving E2, there is an increase in something, that might be due to a linear function or due to being in the middle of a U function. If there is a decrease, that might be due to a linear function or on at one of the ends of the U function. If there is nothing, it would not be reported. So, what have we learned about the role of E2 and brain function? Can specific predictions be made from previous studies? If rodent studies report an inverted U function, then do we see an inverted U function in humans at similar E2 levels? That is, if it was low E2 for the rat, it would be low E2 for the human, etc. - that would be a solid translation. Otherwise, if the results do not offer predictions about the role of E2 in brain function, it is unclear what contribution these results make to the literature.

- o The Methods are well done and ambitious. The task choice is interesting, because due to the introduction, it seems that a task more translational to rodent research would be used, such as a spatial navigation task. Nonetheless, this task offers some insight into amygdala function.

- o For the Results section, the PCA analysis somewhat obscures easy interpretation of the data. As I understand it, as E2 increases, so does factor 1 (dimensions: alertness, strength, energy, clear-headedness). And, the authors examine the effects of vigilance (is that what they are calling factor 1?) on RTs during the active baseline task to show no effect. It would make more sense to compare the RT and accuracy for the primary task, that is, is the picture an indoor or outdoor scene. Do the groups differ on accuracy and RT for this? If so, this should be used as a covariate to examine the effects of E2 over and above these

attentional differences. At the very least, this analysis should be reported.

- o The precuneus link should be developed in the Intro. As it stands, it seems highly speculative, and just because a brain area might be involved with pain is not reason enough to suggest that it is linked to emotional processing.

- o It would be more revealing and accurate to remove the 0mg dose from the analyses. It is a placebo dose, and it is common to separate this dose from most dose function curves. So, the 0mg dose could be plotted, but the trend analyses would not include it because there is no E2 in that dose.

- o Overall, it seems that the scanning and fMRI analyses were confined mostly to hippocampus/amygdala regions, so it does not seem overly surprising that some sort of relation is evident in these areas and memory, particularly when using linear, quadratic, and advanced contrasts.

- o The brainstem connection seems highly speculative.

SUMMARY:

- o The study is ambitious in terms of number of participants and doses. However, the lack of theory and the multitude of analyses that seem to encompass every possible result in the areas examined seem to fall short of making a contribution to the literature on E2 and brain function. At conclusion of the paper, it is not clear that there is any deeper or meaningful understanding of E2 and hippocampus, except, that "was no significant relationship between E2 and hippocampus-dependent memory performance".

Reviewer #2 (Remarks to the Author):

The paper by Bayer et al is an interesting well-designed study examining exogenous manipulations of estradiol valerate at different doses to investigate the relationship (quadratic, linear) between estradiol level, memory and activation of the hippocampus and amygdala. It's been very important in the literature to show that different doses of E2 can have profoundly different effects and this reviewer feels this paper will make an important contribution to the literature as they examined multiple exogenous doses of E2 in a controlled manner. This paper help clarify some issues in the literature on the effects of estradiol on memory and activation in limbic regions. The statistics and conclusions are appropriate except were noted in comments below. I have a few suggestions that I feel will improve the manuscript.

Please provide a table with a breakdown of age, education and parity in the manuscript (or supplement). These data, excluding parity (see below) are analyzed but are not given in the manuscript itself. Is there a difference in parity between groups? Parity can alter menstrual cycle and estradiol levels, and while it likely may not have a role it would be important to verify that there are no significant differences between groups.

The inter-individual variance within groups was high but also interesting. Even if the groups didn't differ by age – is there an age effect in the ability of E2 to increase E2? I am wondering if the variability comes from age-related changes in menstrual cycle fluctuations

in E2/FSH?

In the first sentence of Discussion and throughout the Introduction the authors refer to Linear versus quadratic...wouldn't this also depend on where on the curve you were? So for example if your dose or range of E2 isn't large enough you may expect to see a linear relationship but if your doses were larger you could move into the quadratic part of the curve. I just find this line of reasoning a little odd as it may just be due to the limited range of E2 choices per paper?

E2 also biases strategy use as Korol and Galea's groups have shown that higher E2 is associated with a greater likelihood of using a hippocampal based strategy (even if they may not be as good at it).

On page 3, lines 3-5 neurosteroids are referenced as 7-9 (but did you mean 9-11?) as 7 and 8 refer to distribution of ER subtypes. I think you mean 10 and 11..but clearly according to Kato's and Caruso work plasma levels follow brain levels of steroids and likely greater influence is gonadally-derived (or peripherally-derived estradiol rather than de novo synthesis which is not always seen in adult female rodents (see Barker, Caruso). Because you are not directly testing this I suggest removing this part or toning down your discussion. There is a lot of attention on neurosteroids when in reality there are a number of very distinct changes when ovaries are removed (in women and in rodents) on cognition.

Was there a correlation between serum and saliva levels of E2 (even though one is free and one is bound, one would still expect some correlations?)

Minor:

P values should be in figure captions as well as in text.

Reference 20 is missing an author

Page 15, line 19 – reference 64 is given but this same relationship is seen with cell proliferation and E2 in the hippocampus (Barha, Tanapat)

P16, lines 19-24 – the same dose response is also seen in relationship to working memory in the radial arm maze or in the non-spatial delayed alternation T maze (Holmes, Wide, Galea, 2002; Wide et al., 2004) – it could be changes in task, but it also could be type of estradiol used as well or as you point out but may want to do so again – dose! Certainly when those studies have examined a variety of doses the data tend to show the curvilinear effect. It also matters on the timing of E2 to memory task (if measuring effects on consolidation or acquisition for example).

P16, line 24 - Please briefly expand on the idea that P or menopausal status may contribute to the variability of findings. You could also refer to study by Baxter and others in primates on the negative impact of P with E. In your study your levels of P are consistent across groups. Relative to pregnancy or mid-luteal what are these levels of P like?

Reviewer #3 (Remarks to the Author):

The study investigated E2 dose-response functions in humans (0, 2, 4, 6 or 12 mg E2-valerate) in a randomized, double-blind fashion in 125 naturally cycling women on two consecutive days during the low-hormone menstruation phase. fMRI was acquired during the encoding of neutral and negatively arousing scenes when E2 levels peaked and potential vasomodulatory effects of E2 were examined using ASL. Memory performance was assessed 24 hours later by a recognition test, in a task designed to disentangle hippocampus-dependent from -independent memory processes. Authors report a positive linear as well as an inverted u-shaped relationship between E2-increase and task-related activity in distinct hippocampal subregions, but no significant relationship between E2 and hippocampus-dependent behavioral performance. E2 modulated emotion-related activity in three further areas that express estrogen receptors and are involved in emotional processing, namely the insula following an inverted u-shape, as well as the precuneus, the brainstem and their connectivity following a negative linear function. The authors have undertaken a complex study to address some important questions. I have some questions and comments for the authors prior to being able to recommend publication.

General Comments

- First sentence of the Abstract as well as the very first sentence of the Introduction are rather awkward. Authors should consider re-writing/simplifying.
- In general, the manuscript, although fairly well written, contains awkward language and should be proof-read. [example: tasks were administered? not accomplished... (page 5, line 16); "Beneath statistical tests..." (page 5, line 23)]
- Given the complexity of the study – it would be good to be very specific throughout the manuscript when it comes to measures used to assess various function, etc.

Introduction

- Authors give some specific examples of u-shaped dose-response curve findings in the literature, but not linear. Introduction could benefit from both.
- Authors mention potential effects of E2 on blood flow in carotid arteries – although it turns out that authors do not observe any effects, it should be discussed what the anticipated effects and related concerns were for those not familiar with the literature.
- I may have missed - why were serum E2 levels assessed in only a subset of women? And did the distribution vary across groups?

Results

- Measures of vigilance and alertness (page 6, lines 6+7) - what are those measures exactly? More specific language could be used throughout.
- What does the linear relationship between salivary E2 increase and neural pictures mean (page 9, Figure 3)?
- I may have missed this, but 2 subjects were lost, so how many total per each group?

Discussion

- Authors refer to "task-related effects of E2" several times on page 15. What is a "task-related effects of E2", should be defined much earlier in the paper and I apologize if I missed this. The manuscript in general uses a lot of jargon, which makes it harder for the reader to get a complete picture, especially given the multiple aspects of this complex study – the reader may not be an expert on all.
- Hippocampus-dependent and –independent memory processes should be defined more specifically and early in the manuscript.
- I find the explanation of findings of both linear and inverted u-shape curve between E2-increase and memory-related activity somewhat unsatisfactory. At a later point in the manuscript, authors suggest the lack of spatial resolution elsewhere in the brain with sequences employed when addressing secondary aims – and the resolution required to delineate hippocampal subfields is rather challenging.
- I am not sure I agree with statement that non-human studies/tasks differ on motivation because often the animals are rewarded or punished. Same can be accomplished in humans by adding or subtracting points while performing the task, and many tasks do so. Humans are also generally motivated to do well, especially in front of others (i.e., when tested). Although there may be species-specific differences in how to accomplish those, I would not discount their presence in human studies.
- One big difference between human and non-human studies is that in most non-human studies, animals are OVXed, The action of estrogen in such a system may be fundamentally different from that in an intact system. More over, women in this study were all young and relatively healthy.
- There is generally a lack of acknowledgements of the study's limitations. Moreover, likely small Ns per group and numerous comparisons should be more of a concern.
- The discussion is disappointing and should be placed within the context of non-human findings. It is unclear which questions exactly the results of the current study settle.

Reviewer #4 (Remarks to the Author):

Rationale

The authors sufficiently describe in their introduction the rationale of the study. The overall aim of the study is to examine the effect of estrogen on cognitive function. Although this is not a new research question, there are still unresolved question, which the authors aim to address by manipulating the level of estrogen in a controlled manner.

However, the main underlying hypothesis is that estrogen is the main driving factor and, hence, that the follicular phase is the phase of the menstrual cycle is the mostly deviating cycle phase. While there are many studies supporting this view, there are also studies demonstrating that the luteal phase is deviating as well, i.e. the phase where estrogen and progesterone are high and that this interaction might influence cognitive functioning. The authors briefly mention this interaction in line 6 on page 4 but could have discussed that in more detail (see also comment on progesterone levels in Supp.Figure 1).

Experimental design

The author aim to manipulate the level of estrogen within naturally occurring levels between 0 and 12mg. Therefore, the authors administered estrogen (E2-valerate) on two consecutive days to naturally cycling women in their menstrual phase, where estrogen levels are supposed to be low. Participants were blind to the administrated dose. This procedure appears to be sufficient for studying the effect of estrogen alone.

- Have the authors considered using differences scores in E2 levels rather than total E2 concentrations?
- Was the study controlled for stereotype effects, i.e. were equally often male as female research leaders present?
- Is it recorded whether participants have been pregnant before or have been using contraceptives in the last couple of month before they participated?
- Page 5, line 20: Can the authors comment on why serum levels were assessed only in a subsample? Just for quality check of the saliva samples?
- Page 6, Figure 1, Page 21, line 3/4 & page 23, line 3/4: Although displayed and mentioned in figure 1, the authors may want to emphasize more on the temporal structure of the three days with respect to sampling of hormonal concentration, administration of the drug and time of the day for the fMRI examination, as these are essential parameter for a study like this one.
- Page 25, line 20: Have the authors included realignment parameters as well? Have they used cut-offs for movements?
- Page 26: Was an additional cluster threshold applied?

Results/Discussion

- Supplementary Figure1: Why has panel A only two bars per group? Further, only the placebo group showed a (significant?) drop in progesterone concentration from Day1 to Day2. Could the authors please confirm that the administration of estrogen didn't interact with progesterone decrease? They only comment (page 8, line 7) that they didn't find a linear / quadratic relationship, but that doesn't have to occur - it might be more the absence of the drop of progesterone concentration during the menstrual phase.

Reviewer's Conclusion

This is a well-designed and well-conducted study. However, the results are difficult to understand since most of the discovered effects are not directly reflected in behavioral measures and vice-versa. Consequently, this study contributes more to the heterogeneity of the current literature rather than giving a clear picture.

Further, the discussion is mostly related to the description of the results with only a few attempts to explain the results. In particular, given that the discovery of a u-shaped

response appears to be important to the authors, a more extended explanation why they found a u-shaped response could have been expected. They only mention (page 3, line 13) that it is a matter of debate but this debate should have been discussed in light of the present results.

Reviewer #5 (Remarks to the Author):

Bayer et al. report the results from a task-related fMRI study in which ROI based questions about hippocampal and amygdala activity during the encoding of neural and negative scene based visual stimuli – described as an “emotional-memory task” – were assessed following the administration of either 0, 2, 4, 6, or 12mg of 17-beta-estradiol (E2-valerate) (N=125 young adult women). The wide range of E2 doses was administered within the context a randomized, double-blind, placebo-controlled between-subjects design. Recognition memory was assessed 24 hrs after initial encoding, outside of the scanner, by asking participants to provide remember/know discrimination based responses and a confidence rating. The main claim is that linear and inverted u-shaped dose-response functions describe the link between E2 levels and memory-related hippocampal activity, whereas there was a null effect of E2 levels on emotion-related amygdala activity. The results are novel and stem from a comprehensive experimental protocol in which multiple dependent measures were acquired in order to support the key claims.

However, there are at least four areas of significant concern that currently undermine the main claims and conclusions: (1) lacunae in the reporting of methodological information and statistical analyses preclude full evaluation of several of the key claims (outlined in detail below); (2) there is repeated reference to subregional hippocampal activity, yet none of the reported neuroimaging analyses examined hippocampal activity at a subfield unit of analysis; (3) the manuscript does not currently adequately contextualise the aims, results and conclusions within a broader context of studies. Indeed, in the abstract, the conclusions and broader implications of the results are not addressed, nor are directional hypotheses stated in the introduction, particularly in relation to what one might specifically predict in terms of the link between E2-dose mediated modulation of hippocampal activity and human recognition memory performance; and (4) the null association between hippocampal activity and memory-guided behaviour leaves the behavioural significance of the E2-mediated modulation of hippocampal activity an unresolved question.

Overall, the manuscript would benefit from considerable additional detail regarding the reasoning behind several of the analyses and more thorough consideration of the potential caveats associated with the results.

Methods and statistical analyses

Procedural details provided in the current version of the manuscript associated with the emotional-memory encoding task and offline recognition test do not currently contain sufficient detail to enable reproduction of the study:

a. How was the emotional valence of the 180 lure items matched against the target 180 items? Were targets and lures counterbalanced across participants and group? What was the number/portions of emotional and neutral items in the target and lure sets of items?

b. It is not possible to evaluate whether the effects of encoding can be assessed discretely from novelty based effects – i.e., in the current description, it is unclear whether each of the 180 target pictorial stimuli were presented once across the 5 blocks of trials at encoding, because neither the number of trials per encoding block and/or the number of times each stimulus presented are stated.

c. The authors state, “In this task, participants indicated every second the direction of a randomly left or right pointing arrow presented in the middle of the screen.”- this needs to be re-written, since it is not possible to decipher the task from this description and evaluate whether this was an appropriate baseline task. How were the presumed differences in the number and timing of manual responses between the baseline and encoding task modelled in the GLM analyses?

d. The event structure of the recognition test cannot be inferred from the current description – (1) as written, it seems that a (second-order) confidence estimate was acquired, possibly conflating first-order and second-order estimates. If so, why?, “6 participants were required to indicate on a 6-point confidence scale ranging from 'very sure new' to 'very sure old' whether the image was presented during encoding and how sure they were about this decision.” Did the recognition test, in fact, ask participants' confidence separately from their type-1 ratings? It is unclear why the Elfman et al.. (2014) paper is cited on Line 17, Page 5 in support of the design of the recognition test?; (2) how long was each picture presented on each trial of the recognition test? What were the verbatim instructions given to participants for each event on the recognition test?; and, (3) were the response options on the recognition test remember/know/new? State the instructions associated with the presumed remember/know/new response options?

e. State the method of correction applied to the behavioural statistics, “Statistical results were considered significant at a threshold of $p < .05$ corrected for multiple comparisons if adequate.” – presumably, the alpha criterion was modified according to the method of correction for multiple comparisons so that it was fixed or adjusted, such as the Holm-Bonferroni method.

f. As currently written, the rationale and method for obtaining the estimates of recollection and familiarity from ROC plots and the remember/know responses is difficult to understand without prior knowledge. This is undesirable for a widely-read journal such as Nature Communications. Also, add a new figure to depict the area under the ROC curve associated with type-1 and type-2 performance.

g. Are the d' values, estimates of recollection and familiarity, and criterion reported in Supplementary Table 1 significantly different as a function of experimental group? Report the relevant inferential statistics. Was there a significant difference in mean confidence

associated with all 'old' and all 'new' retrieval cues, as a function of group and stimulus type (neutral, negative); i.e., report whether the confidence ratings were diagnostic of recognition memory. It might also be worthwhile reporting analyses that incorporate participants' biases and first-order response such as meta-d' (Maniscalco & Lau 2012) or type-II ROC curve (Flemming et al. 2010).

h. Supplementary Table 2 – update to report beta and r-squared values associated with each of the regression analyses.

Similarly, there is information missing from the neuroimaging methods that makes it hard to adjudicate whether some of the choices in the analyses and reporting of the results are appropriate:

a. Specifically state the voxel spatial resolution of the EPI and T1-weighted sequences in mm³ – it is currently not possible to evaluate the choices made in terms of spatial smoothing and critically the merit of consistent reference to the capacity to evaluate, “hippocampal subregional activity” and later qualification in the discussion section about limits on these types of analyses due to the acquired spatial resolution.

b. State the approach used to assess and exclude the impact of movement on the fMRI data.

c. State the MNI coordinates of the precuneus seed region in the PPI analysis. What was the size of the sphere in this analysis? Separately, what were the sizes of the spheres used to derive beta values extracted from peaks inside clusters surviving multiple comparison correction and used for the hierarchical robust regressions?

d. The authors should test their hypotheses against ROIs drawn from clusters that are not derived from the peak inside the clusters surviving multiple comparisons – more recent approaches contend the latter is a form of double-dipping; i.e, the authors should evaluate whether the results hold when ROIs are based on coordinates defined independently from the primary task based contrast.

e. The authors state, “ASL data of six participants were missing because of technical failure and excessive head movement” – state which experimental group these participants were assigned to; it is not possible to evaluate the impact of these missing data without this information.

f. Report was the final n / group for each set of analyses throughout the results section. In Figure 4, why are fMRI results based on n=118? – partial clues in the methods do not reveal the whole picture, “3 further participants had to be excluded because of empty cells (i.e. no correct 'know' responses for neutral pictures, no 'remember' responses)” and later, “Functional MRI data of two participants were missing due to technical failure and excessive head movement.”

In Figures 5 and 6, why are the fMRI results based on n=121?

g. Figures 4-6 – the figures are thresholded at $p < 0.005$ – is this threshold for a p-value corrected for multiple comparisons? Either way, the thresholding is at variance with the $p > 0.05$ FWE corrected for the statistical analyses. A t-map showing the results based on $p > 0.05$ FWE corrected should be included in updates to figures 4 and 5.

h. Provide information on how post-error trials and reaction time outliers were handled in the analyses. What proportion of trials were affected by errors?

i. In the fMRI analyses, what was the duration of the response window examined for each back-sorted remember>know trial at encoding? Likewise for the other two main sets of event-related fMRI analyses.

j. Were trial-by-trial manual response latencies at encoding included as regressors of interest in the fMRI GLM analyses?

Several important additional analyses are needed in order to improve how the main claims have been tested and thus provide more convincing evidence to strengthen the conclusions. In particular, report the following additional analyses:

a. What is the link between salivary and sera E2 levels? Include an analysis to test this link; the results are important for understanding the implications of basing the main results and claims on the increase in D1-D2 absolute salivary E2 levels, especially given that sera E2 levels were subsampled in 90/123 participants. State why the results of salivary levels of E2 from 2 of the 125 are missing.

b. Given the wide variances in absolute salivary E2 levels within each group, report inter-quartile ranges for the absolute salivary E2 levels associated with each of the 5 groups. Also report the corresponding effect size for the D1-D2 absolute salivary E2 levels, as a function of group.

c. The results should not be discussed in terms of trend level significance; e.g., Lines 1-4, Section 2.1, "With respect to factors extracted from the mood questionnaire, only a positive linear relationship of factor 1 (dimensions: alertness, strength, energy, clear-headedness) survived Bonferroni correction with trend-level significance [$t(121)=2.31$, $pcor=.092$]." On Page 9, "The analysis of rCBF revealed a trend-level significant positive relationship with salivary E2 increase in the right amygdala [$x=16$, $y=-4$, $z=-20$; $Z=2.85$, $pcor=.067$; Supplementary Figure 2]." Similarly, on Pages 10-11, "with trend-level significance in the right brainstem [$x=6$, $y=-30$, $z=-12$, $Z=4.60$, $pcor=.056$]." If the authors wish to pursue this approach, then, for equivalence sake and as an example, one of the key clusters in the right posterior hippocampus and the significant positive linear association with increases in E2 levels [$x=33$, $y=-30$, $z=-12$; $Z=3.69$, $pcor=.034$; Figure 4AB] should be discussed as trending towards non-significance.

d. (i) State the hit and false alarm rate on the recognition test and plot the ROCs; (ii) What were the median number and range of hit and correct rejection trials, as a function of group; and (iii) Report the mean/median response latencies associated with the presumed remember/know/new responses on the recognition test and evaluate whether these differed as a function of group.

e. What were the mean arousal ratings for neutral and negative stimuli, as a function of group. Were the differences significant between the groups? Further, what are the mean arousal ratings for neutral and negative stimuli associated with the presumed, remember, know and new responses, as a function of group? These data have relevance for the decision to examine E2 effects on amygdala activity by collapsing across what I presume is retrieval success; cf. Page 10, Lines 16-17.

f. What are the precedents for the linear relationship between arousal ratings for neutral stimuli and D1-D2 and D1-D3 increases in E2? – this warrants consideration.

g. Is it being suggested here that manual RT should be considered as a proxy for alertness? If so, the following sentence needs to be unpacked and reasoning explained, “exploratory analyses using reaction times from the active baseline task of the fMRI paradigm (see below) showed that the subjective effects in vigilance are not reflected in objective differences in a measure of alertness [$t(116)=-1.46$, $p=.146$].

h. What were the other brain regions associated with the remember>know whole-brain contrast? Indeed, in the discussion, the authors aver, “Moreover, the task-related effects of E2 were highly selective and restricted to well defined brain regions although the paradigm evoked activity in a wide networks of areas.” Report these data with the appropriate corresponding statistics.

i. The regression analysis does not provide sufficient evidence to adjudicate whether or not E2 selectively modulates hippocampal activity associated with the remember>know contrast. Hippocampal activity needs to be compared against activity for the same contrast from at least one other neural region that is hypothesised to be affected by E2, such as the prefrontal cortex (e.g., Ottowitz et al., 2008) and a control region associated with recollection-based recognition memory such as lateral parietal cortex. Currently, it is not possible to assess the selectivity of the main effect associated with one of the key conclusions. Indeed, in the analysis of the link between E2 levels and amygdala activity, the authors already tested/reported results from at least two other regions (right precuneus, right brainstem).

j. Additionally, a PPI analysis is reported for the contrast between neutral and negative stimuli, yet a corresponding analysis is not reported to explore changes in functional connectivity associated with the remember>know contrast. Although not stated a priori, the rationale for the PPI analysis is reasonable, therefore, it is unclear why the authors did not conduct and report a PPI analysis for the remember>know contrast for a hippocampal seed region to examine functional connectivity with, for example, the PFC – another key target of

E2.

k. The neural regions associated with encoding success – hits>misses (Page 10, Lines, 14-15) – should (a) be reported in full and (b) the region(s) from this contrast tested in the reported regression should be stated along with the appropriate statistics.

Discussion

As mentioned, it is repeatedly stated that, “A large body of animal studies intensely characterized the beneficial effects of E2 on hippocampal neuroplasticity and hippocampus-dependent memory”, yet the analyses do not reveal a significant effect of E2 levels on hippocampal “neuroplasticity”. It thus remains unconvincing how these data necessarily “bridge the gap” in the way the authors suggest because: (1) there is no direct assessment of how hippocampal activity measured using event-related fMRI is a proxy for neuroplasticity observed in animal studies – the key contrast is based on a difference between remember>know rather than a learning-related neuroplastic change in activity over, for example, time at encoding. Indeed, a more standard approach based on encoding success (hits>misses) reveals no link with E2 levels; and, (2) there is no connection between hippocampal activity and hippocampal-dependent behavioural performance; (3) what are the empirical precedents for postulating that the modulation in hippocampal activity is mediated by between-group differences in synaptic density that have occurred over a day? (Page 15); and, (4) the current way in which the subfield-task-computational interactions are discussed as part of the conclusions almost implies that such analyses were conducted; this is misleading and should be revised to avoid the implication that the analyses were conducted at the level of hippocampal subfields, since it is entirely speculative in relation to the current study; namely, Lines Page 19-20.

Another consideration overlooked when discussing the interpretation of the main result is that the link between E2, hippocampal activity and behaviour was only examined from the perspective of an offline recognition memory test – the hippocampus has been extensively implicated outside of recognition memory, so it is conceivable that the E2 mediated-modulation of encoding-related hippocampal activity has an impact on cognitive behaviour outside of “memory paradigms”. This possibility should at least be introduced and discussed. Furthermore, a more direct test of the significance of E2-mediated changes on hippocampus-dependent memory would have been to acquire fMRI data during the recognition test.

Minor Points

1. Please re-write: (1) “24 h later a recognition test was accomplished outside of the scanner using confidence ratings followed by remember/know judgements for old responses (58).” - a recognition test was “administered” and explain event structure with more step-wise detail; (2) Relatedly, in the abstract, the timeline is unclear, “Memory performance was assessed 24 hours later by a recognition test”...memory performed was assessed 24 hours after initial encoding?; and,

(3) Please clarify... ", designed to disentangle hippocampus-dependent from -independent memory processes." - as written, it is unclear how a recognition test per se disentangles hippocampal-dependent and independent "retrieval" processes. This needs to be re-written to address this point.

2. Should read, "were significantly," in the following sentence: "Neither recollection, which is thought to be hippocampus-dependent, nor familiarity was significantly associated to E2 in a linear or a quadratic manner [all $p_{cor} > .02$; Supplementary Table 1]."

3. Should read, "a psycho-physiological interaction (PPI; 62) analysis was conducted, in the following sentence: "To examine exploratory whether E2 administration also affected the connectivity between precuneus and brainstem during emotional processing, a psycho-physiological interaction (PPI; 62) analysis was calculated.

4. The following sentence needs to be re-written because it is unclear what was being examined in this subsidiary analysis, "However, exploratory analyses using reaction times from the active baseline task of the fMRI paradigm (see below) showed that the subjective effects in vigilance are not reflected in objective differences in a measure of alertness [$t(116) = -1.46$, $p = .146$]." See above.

5. Grammar, Page 11, Lines 5-6: "To examine exploratory whether E2 administration also affected the connectivity between precuneus and brainstem during emotional processing."

6. Pages 20-21, what is the unit related to the mean value reported on lines 21-22; 1-2, respectively?

7. What was the amount of financial compensation given to the participants?

Introductory Response to all Reviewers

We first want to take the opportunity to express our sincere gratitude for the extraordinary effort and time the referees put into reviewing our manuscript. In response to the reviews, we substantially rewrote the introduction and discussion as well as modified the method section and added the requested analyses. We believe these revisions resulted in a significantly improved manuscript.

In particular, we agree that framing our results better within animal literature and more explicitly explaining and motivating our research hypothesis, would allow us to clarify our contribution to the understanding of E2's effects on the human brain. The additional analyses proposed by the reviewers has also greatly improved the transparency and strengthened the conclusions of our study. We apologize for the confusion caused by incomplete reporting and unclear language. In response to the reviewers' comments about the writing, a native English speaker (Gina Joue) helped us with improving the grammar and language of the manuscript.

Please see below for detailed responses to specific comments. Passages from the main manuscript and the Supplementary Material are in italics. Changes to the manuscript are marked in red.

Response to Reviewer #1

Comment 1: The Introduction is thorough, but it does not offer any theory about what the authors expect to see. In fact, this lack of theory is a recurring theme throughout the paper. To say that phenomenon have been shown to have a linear function, a quadratic function, and that things sometimes show an inverted U function offers little information about E2 or the hippocampus. For example, if after giving E2, there is an increase in something that might be due to a linear function or due to being in the middle of a U function. If there is a decrease, that might be due to a linear function or on at one of the ends of the U function. If there is nothing, it would not be reported. So, what have we learned about the role of E2 and brain function? Can specific predictions be made from previous studies? If rodent studies report an inverted U function, then do we see an inverted U function in humans at similar E2 levels? That is, if it was low E2 for the rat, it would be low E2 for the human, etc. - that would be a solid translation. Otherwise, if the results do not offer predictions about the role of E2 in brain function, it is unclear what contribution these results make to the literature.

Response: We sincerely thank the reviewer for directing our attention to the overly superficial and hence massively oversimplified way we presented the theoretical and empirical background motivating our study. In particular, we agree that comparing dose-response curves across studies is meaningless when it is not related to the induced range of estrogen levels and interpreted.

In response to the reviewer's comment we have substantially rewritten the introduction and discussion. In particular, we expound on a concrete hypothesis of the dose-range that led to different response curves in the revised introduction.

We have copied the most important parts of the revised introduction and discussion that directly address the comment here in the rebuttal. We refer the reviewer to the revised manuscript for a more complete view of all the changes in context.

Parts of the introduction modified in response to the reviewer's comment:

“Interestingly, based on studies that have induced more than one level of E2 it was proposed that the dose-response relationship between E2 and neural plasticity as well as behavior might not follow the traditional sigmoidal but rather an inverted U-shaped function¹⁻³. That is, one would expect increasing benefits until medium levels of E2 but then a decrease and none or even negative effects at high levels. However, some studies have also observed the traditional sigmoidal or even linearly increasing dose-response relationships⁵⁻⁷.

Taking the heterogeneity of these studies into account, the inconsistency in reported response functions might be resolved with the observation that neural plasticity monotonically improves with increasing E2 levels mostly at physiological E2 concentrations^{6,8-13}. Critically, only when supraphysiological E2 levels (i.e. levels above estrous cycle peaks) are reached does the dose-response relationship exhibit an inverted U-shape^{6,8,9,12,13}. However, a few studies have reported monotonic improvements also at clearly supraphysiological E2 levels, and some effects only occur at supraphysiological levels⁵⁻⁷. Whereas the monotonic increase and saturation is compatible with the traditional sigmoid dose-response function of pharmacological studies, the inverted U-shaped curve needs to be explained by a

more intricate pattern of underlying processes. It has been proposed that the different characteristics of the estrogen receptors and their interaction accounts for this more complex response curve ^{3,14}.

With respect to learning and memory, the diversity of regimens and tasks makes it more difficult to see a clear pattern: inverted U-shaped curves might already characterize high physiological E2 levels, and null effects and even monotonic decreases have been reported ^{9,15-18}. *The question whether the response to E2 turns into an inverted U even at physiological levels or only at supraphysiological levels is of relevance because only in the first case would E2 regularly reach concentrations disadvantageous to memory and learning during the normal menstrual cycle.*

(...)

Importantly, this E2 treatment regimen induced a wide range of physiological (within 2- and 4-mg groups) but also supraphysiological (within 6- and 12-mg groups) E2 levels. We chose this wide range of E2 levels to test our hypothesis that the dose-response function between E2 and neural activity increases monotonically within physiological ranges and turns into an inverted U-shape only at supraphysiological levels.”

In the revised discussion, we relate our findings much more to the animal literature and the mechanisms that have been proposed to underlie the inverted U-shape:

“Critically, the inflection point of the quadratic function in the hippocampus and the insula were clearly above the physiological levels of E2 reached during the menstrual cycle. It should be emphasized that the procedure of fitting a quadratic in addition to a linear function to the continuously distributed levels of E2 increase was strictly data-driven and did not require any a priori assumptions about the inflection point. Our results therefore support the hypothesis that inverted U-shape dose-dependent responses for E2 and neural activity are only observed at supraphysiological levels. In other words, within the range of E2 levels experienced during the menstrual cycle, higher levels of E2 were increasingly beneficial in the areas that exhibit an inverted U-shaped response function. E2 effects on neural activity returned to baseline in these areas only at levels that occur during early to mid-pregnancy. In other brain areas, monotonically increasing but also decreasing response functions were observed across the full range of E2 levels. (...)

Various mechanisms have been suggested to underlie the inverted U-shaped curves of rapid and genomic effects of E2. The inverted U-shaped response in rapid effects might reflect the optimal level of receptor activation ¹⁴. Alternatively, different doses might lead to differences in coupling to distinct pathways via changes in receptor conformation, or the activated kinases might be inhibited at higher doses ^{3,14}. Moreover, the activation of ER β in hippocampal interneurons at higher E2 levels can decrease neural activity ⁹. Genomic effects might show an inverted U-shaped response due to differences between ER α and ER β . In particular, whereas ER α activation stimulates synaptogenesis, ER β activation and an increase in ER β levels result in a decrease in synaptogenesis ⁵⁰⁻⁵³. As the affinity of ER α for E2 is higher, low levels of E2 will mostly activate ER α , which is transcriptionally more active. In

contrast, at higher levels, more of ER β are activated. The amount of synaptogenesis may additionally depend on the relative expression of ER α /ER β . ER α is overall less expressed in the hippocampus and its transcription is upregulated by E2, so that its expression increases with E2 levels. However, with higher E2 levels, activated ER β downregulates ER α -dependent transcription³. Interestingly, consistent with our finding of inverted U-shaped functions only at supraphysiological levels, it has been suggested that E2 levels within the physiological range promotes ER α expression in the hippocampus, whereas at supraphysiological levels, ER α expression decreases⁵⁴.

Strikingly, we not only found an inverted U-shaped relationship in the lateral posterior hippocampus but also a monotonically increasing response curve in a more medial posterior hippocampal area. An increase in hippocampal plasticity at supraphysiological levels of E2 has similarly been observed in animals^{9,12,13,55}. Although at first sight it seems surprising that we observed both types of response curves in distinct hippocampal clusters, previous studies have also described dissociable dose-response functions for different hippocampal subfields (CA1 vs. CA3) and depending on whether hippocampal subregions or whole hippocampus is considered (whole vs. rostral hippocampus)^{5,9}. It has been suggested, that such region-specific dose-response functions could be a result of different ER α /ER β ratios^{5,51}, which vary depending on hippocampal subregion⁵⁶⁻⁵⁹. Further explanations for the observed responses could lie in the differences in how E2 regulates ER α transcription in CA1 and CA3 (as suggested by the transcription patterns describe by Mendoza-Garcés and colleagues⁵⁷), or subfield-specific differences in nuclear E2 uptake⁶⁰. In fact, the inhibition of aromatase activity affects CA1 and CA3 differently⁶¹. However, the spatial resolution of our fMRI scans is too coarse to relate the different dose-response curves to specific anatomically defined hippocampal subfields and only permits a functional dissociation.

These mentioned differences in hippocampal subregions would also explain why the E2-BOLD signal relationship continues to increase at all at supraphysiological E2 levels in the more medial posterior hippocampus. A higher ER α /ER β ratio, as reported for instance in CA3 compared to CA1⁵⁹, could result in a shift of the inflection point of the dose-response curve at higher, supraphysiological E2 levels that were above the range induced in the present study. An alternative explanation for the different regional dose effects could be that the observed changes of the BOLD signal with increasing E2 levels reflect distinct underlying neuronal actions of E2 in both hippocampal clusters, such as genomic regulation of synaptogenesis vs. the rapid effects on glutamatergic synaptic activity, each of which might manifest as different response functions.”

Comment 2: The Methods are well done and ambitious. The task choice is interesting, because due to the introduction, it seems that a task more translational to rodent research would be used, such as a spatial navigation task. Nonetheless, this task offers some insight into amygdala function.

Response: The Reviewer raises a very pertinent point here which we considered at great length before finally opting for the emotional memory task. We decided not to attempt to

adapt a spatial animal task for a variety of reasons.

First, and most importantly, the chosen task allows us to not only investigate activity related to hippocampus-dependent memory formation, but also amygdala-dependent emotional processing, including emotional memory formation which is based on the interaction of both areas. The effects of estrogen on amygdala and amygdala-related affective processing have gained increasing interest in recent years (Toffoletto et al., 2014). Moreover, the task elicits activity in larger networks of cortical and subcortical areas that express estrogen receptors but have not yet been addressed by animal studies (Spaniol et al., 2009; Spalek et al 2015, Kritzer et al., 2002). By acquiring whole brain fMRI volumes, this task allowed us to therefore test hypotheses about estrogen effects in other brain regions in parallel, which is a great advantage particularly given such a large-scale project. Second, we have already used a similar emotional memory task before and observed an influence of menstrual-cycle dependent hormone fluctuations on hippocampal activity (Bayer et al., 2014). Third, we have reassurance from animal studies using non-spatial memory tasks that effects of estrogen are not necessarily exclusive to spatial memory formation (Luine, 2015).

We acknowledge that our introduction as well as our discussion might have lead the reader to expect a more direct translational approach. Therefore, we revised the introduction as follows to more clearly explain the motivations for the task we chose:

“Most animal studies have investigated E2 effects on hippocampal neurons and hippocampus-dependent learning and memory. It also has been reported to influence the shift from hippocampus-independent to -dependent learning strategies^{34,35}. Therefore, we employed a memory paradigm that allowed us to disentangle hippocampus-independent from hippocampus-dependent recognition memory^{36,37}. ‘Recollecting’ an item, i.e. together with contextual information about the encoding episode, relies critically on the hippocampus, whereas the acontextual sense of ‘familiarity’ with an item relies on parahippocampal structures and is independent of the hippocampus³⁶. To concurrently investigate the effects of E2 on the amygdala and affective processing, both neutral and emotionally arousing stimuli were used. Taken together, we employed a memory paradigm that is known to not only recruit hippocampal and amygdala activity but also to dissociate hippocampal from extra-hippocampal contributions to memory performance. Moreover, as the task activates a larger network of cortical and subcortical areas, it allowed us to use whole-brain functional magnetic resonance image (fMRI) analyses to also explore E2 effects in brain areas that express estrogen receptors but have not yet been addressed by animal studies^{38–40}.

and in the discussion:

“In order to increase sensitivity to detecting E2 effects on memory, we employed a paradigm which disentangles hippocampus-dependent from hippocampus-independent memory processes. We were previously able to detect changes in hippocampus-dependent memory for negative stimuli throughout the menstrual cycle using a similar paradigm²⁷.

Luine, V. Recognition memory tasks in neuroendocrine research. *Behavioural Brain Research* **285**, 158–164 (2015).

Spalek, K. et al. Sex-Dependent Dissociation between Emotional Appraisal and Memory: A Large-Scale Behavioral and fMRI Study. *J. Neurosci.* **35**, 920–935 (2015).

Spaniol, J. et al. Event-related fMRI studies of episodic encoding and retrieval: Meta-analyses using activation likelihood estimation. *Neuropsychologia* **47**, 1765–1779 (2009).

Toffoletto, S., Lanzenberger, R., Gingnell, M., Sundström-Poromaa, I. & Comasco, E. Emotional and cognitive functional imaging of estrogen and progesterone effects in the female human brain: a systematic review. *Psychoneuroendocrinology* **50**, 28–52 (2014).

Comment 3: For the Results section, the PCA analysis somewhat obscures easy interpretation of the data. As I understand it, as E2 increases, so does factor 1 (dimensions: alertness, strength, energy, clear-headedness). And, the authors examine the effects of vigilance (is that what they are calling factor 1?) on RTs during the active baseline task to show no effect. It would make more sense to compare the RT and accuracy for the primary task, that is, is the picture an indoor or outdoor scene. Do the groups differ on accuracy and RT for this? If so, this should be used as a covariate to examine the effects of E2 over and above these attentional differences. At the very least, this analysis should be reported.

Response: The reviewer is correct in this understanding. The first component of the PCA (originally labeled as ‘Factor 1’) represents a measure of subjective vigilance derived from the mood questionnaire. We rephrased this sentence in the revised manuscript to remove the ambiguity:

“Only the first component (dimensions: alertness, strength, energy, clear-headedness; see Supplementary Table 13), determined by a principal component analyses on data from the mood questionnaire, showed a non-significant trend after Bonferroni correction [$t(121)=2.31$, $p_{corr}=.092$, $\beta_{std}=.188$]. We will refer to this principal component as ‘subjective vigilance.’ To test whether changes in subjective vigilance are reflected in an objective measure of alertness, individual reaction times from the distractor task (i.e. arrow task) and the encoding task were used as a proxy for intrinsic alertness.”

We also followed the recommendation of the reviewer and included robust regression analysis for reaction time and accuracy during the encoding task (indoor/outdoor decision):

“Neither reaction times (rt) nor accuracy in the arrow task [rt: $t(121)=-.68$, $p_{corr}>.999$, $\beta_{std}=-.011$; accuracy: $t(121)=-0.20$, $p_{corr}>.999$, $\beta_{std}=-.004$] or the encoding task (indoor/outdoor decision) [rt: $t(121)=-.57$, $p_{corr}>.999$, $\beta_{std}=-.047$; accuracy: $t(121)=-.37$, $p_{corr}>.999$, $\beta_{std}=-.023$] were significantly related to the Day 1 to Day 2 increase in salivary E2.”

Comment 4: The precuneus link should be developed in the Intro. As it stands, it seems highly speculative, and just because a brain area might be involved with pain is not reason enough to suggest that it is linked to emotional processing.

Comment 7: The brainstem connection seems highly speculative.

Response to Comments 4 & 7: We agree with the reviewer that the link between precuneus or the brainstem with emotional processing is tenuous, and we had attempted to mediate how we presented the results. We hoped to have clearly conveyed that we did not

hypothesize their involvement in emotional processing when we set out to do the study, as neither has been a focus in animal research. For this reason, we did not mention them in the introduction. As estrogen receptors are expressed in many brain areas which have not yet been focused on in animal studies, we used a whole brain fMRI sequence and a task that elicits activity in large networks of areas in order to detect potential dose-dependent E2 effects in the entire brain. Importantly, to avoid false positive findings we adjusted our statistical test significance thresholds with a very conservative correction for multiple comparisons across the entire scan volume. We hope to also have mitigated any interpretation that we are implying that they have direct or exclusive involvement in emotional processing, but we can only report that we see E2 modulate activity in these areas when we focus on brain activity changes *related* to affective processing.

In response to the reviewer's comment, we have modified the manuscript (for details, see below). However, based on our reasoning outlined above and the potential implications for future studies, we were hesitant about completely removing these results which strongly implicate the precuneus (significantly) and the brainstem (almost significantly). However, if we fail to convince the reviewer, we would certainly follow his/her advice.

In response to the reviewer's comment, we now clearly state in the revised introduction why this task and the whole brain fMRI sequence was chosen:

"Therefore, we employed a memory paradigm that allowed us to disentangle hippocampus-independent from hippocampus-dependent recognition memory^{36,37}. 'Recollecting' an item, i.e. together with contextual information about the encoding episode, relies critically on the hippocampus, whereas the acontextual sense of 'familiarity' with an item relies on parahippocampal structures and is independent of the hippocampus³⁶. To concurrently investigate the effects of E2 on the amygdala and affective processing, both neutral and emotionally arousing stimuli were used. Taken together, we employed a memory paradigm that is known to not only recruit hippocampal and amygdala activity but also to dissociate hippocampal from extra-hippocampal contributions to memory performance. Moreover, as the task activates a larger network of cortical and subcortical areas, it allowed us to use whole-brain functional magnetic resonance image (fMRI) analyses to also explore E2 effects in brain areas that express estrogen receptors but have not yet been addressed by animal studies³⁸⁻⁴⁰."

Additionally, we explicitly state in the results and discussion sections that these findings were unexpected and emphasize that a particularly conservative correction for multiple comparisons across the entire scan volume was used in order to avoid false positive results. We also substantially shortened the discussion of these findings:

"Although not predicted, we observed monotonically decreasing responses associated with affective processing in the precuneus and a non-significant tendency for such a response in the brainstem, where the functional coupling between these two areas seems to also be inversely affected by E2 levels. (...)

We also found that greater E2 levels monotonically lowered neuronal responses linked to emotional processing in the posterior precuneus, similarly tended (non-significantly) to decrease activity in a brainstem cluster, putatively covering the serotonergic raphe nuclei and/or the dopaminergic VTA, and decreased the connectivity between both areas. Although we did not specifically predict E2 to

modulate these areas during affective processing because animal research on E2 has not focused on these areas yet, it is coherent with previous studies in humans. Not only have both areas been implicated in affective processing, among other processes^{72,73}, these two areas express ERs, and activity in these regions is known to be modulated by the menstrual cycle and HRT^{38,74-77}.”

Comment 5: It would be more revealing and accurate to remove the 0mg dose from the analyses. It is a placebo dose, and it is common to separate this dose from most dose function curves. So, the 0mg dose could be plotted, but the trend analyses would not include it because there is no E2 in that dose.

Response: The reviewer raises a valid point which we thoroughly discussed in our group and with the pharmacologist of our institute, Dr. Jan Haaker. However, we came to the conclusion that there might be a critical difference in the meaning of placebo groups in studies administering substances that naturally occur in the body such as hormones compared with studies administering substances which naturally do not occur in the body such as most pharmaceuticals. In particular, as our participants were women with intact ovaries, the placebo group also has meaningful E2 levels and, importantly, fluctuations in endogenous E2 levels have been shown to influence neural activity. As we were interested in testing how differences in E2 level affects the brain, the placebo group was not just a negative control but a meaningful reference group; therefore, the natural changes in E2 levels in these women should not be neglected but included in the analysis. Therefore, all of our analyses are based on the change in E2 level by subtracting E2 baseline concentration from E2 concentration on day 2, resulting in non-zero dose differences also for the placebo group. We leave it to the reviewer to decide whether we should include this reasoning in the manuscript in order to make it explicit to the reader.

Comment 6: Overall, it seems that the scanning and fMRI analyses were confined mostly to hippocampus/amygdala regions, so it does not seem overly surprising that some sort of relation is evident in these areas and memory, particularly when using linear, quadratic, and advanced contrasts.

Response: The reviewer is correct in saying that we selected a task that targeted the hippocampus and amygdala. As such, it is not surprising that the main effects derived from this task (i.e. remember>know, hit>miss, emotional processing, EEM effect) were associated with robust activity in these areas. However, it is not trivial that we observed linear and quadratic relationships between salivary E2 increase and activity related to hippocampus-dependent memory. Moreover, there was no relationship between salivary E2 increase and amygdala activity associated to any of the four contrasts. It is also of note that the whole-brain fMRI sequence made it possible to observe significant relationships in areas that were not predicted by animal research, i.e. the precuneus, the brainstem and the insula.

Our motivation was to test our primary hypothesis that E2 modulates the hippocampus and the amygdala, as proposed by animal research. However, because of the widespread expression of E2 receptors, as mentioned above, we deliberately chose a task known to reliably elicit activity in larger cortical and subcortical networks (Spaniol 2009; Spalek et al., 2015). Although we did not have strong a-priori hypotheses for E2 effects in these other areas (see response to reviewer's comments 4 and 7), the chosen fMRI sequence and

analyses would have been sensitive to detecting E2-related changes in activity in other regions – as we did in the precuneus and brainstem. (In fact, it could be even easier to detect E2 effects on cortical areas because the signal-to-noise ratio of the BOLD signal in fMRI is 4-5 times higher in cortical regions compared to the medial temporal lobe, which is much farther from the head coil of the MRI scanner.)

Neither the task, the scanning parameters, nor fMRI analyses were confined to the hippocampus and amygdala. Only our a-priori hypotheses were limited to these areas. We believe that it is of particular interest that although the task elicits activity in larger network of areas, the effects of estrogen are restricted to just a limited number of foci, which we have highlighted in the discussion (see paragraph 13). That the hippocampus was among these areas was predicted based on the animal literature and therefore does not seem overly surprising.

We realized from the reviewer's comment that our method section obviously created the impression that a multitude of methods were employed. We would like to point out that we restricted our analyses to brain activity associated with a set of 4 contrasts defined a-priori on the individual level (i.e. remember>know for hippocampus-dependent memory, hit>miss for general encoding success, negative>neutral for emotional processing and the contrast for the emotional enhancement of memory). Group analyses were restricted to the two linear and non-linear regression analyses within SPM (i.e. straightforward, standard methods for analyzing fMRI data), testing for the two response functions suggested by from animal literature. These analyses were complemented only by confirmatory robust regression analyses to guard against potentially influential outliers. We would like to emphasize that no further advanced contrasts or any data mining were done.

In response to the reviewer's comment we now state more explicitly that there was task-evoked activity not only in the hippocampus and amygdala but also in larger networks (see Supplementary Figure 4 and Supplementary Tables 8 & 9) and describe the analyses more comprehensively.

Introduction

“Moreover, as the task activates a larger network of cortical and subcortical areas, it allowed us to use whole-brain functional magnetic resonance image (fMRI) analyses to also explore E2 effects in brain areas that express estrogen receptors but have not yet been addressed by animal studies³⁸⁻⁴⁰.”

Methods

“To investigate the effects of E2 on memory and affective processing, the following 4 contrasts, defined a priori, were calculated for each participant: 1. A remember/know contrast (remember > know) to specifically examine hippocampus-dependent memory; 2. General encoding success (subsequent hit > miss); 3. The effects of emotional processing (negative > neutral); 4. EEM (valence [negative > neutral] × general encoding success [hit > miss]). (...)”

Spalek, K. et al. Sex-Dependent Dissociation between Emotional Appraisal and Memory: A Large-Scale Behavioral and fMRI Study. *J. Neurosci.* 35, 920–935 (2015).

Spaniol, J. et al. Event-related fMRI studies of episodic encoding and retrieval: Meta-analyses using activation likelihood estimation. *Neuropsychologia* 47, 1765–1779 (2009).

SUMMARY:

The study is ambitious in terms of number of participants and doses. However, the lack of theory and the multitude of analyses that seem to encompass every possible result in the areas examined seem to fall short of making a contribution to the literature on E2 and brain function. At conclusion of the paper, it is not clear that there is any deeper or meaningful understanding of E2 and hippocampus, except, that “was no significant relationship between E2 and hippocampus-dependent memory performance”.

Response: We fully understand that the way the manuscript was originally written could give such an impression. In light of this comment, we have rewritten the manuscript substantially. We hope that in the revised version, we have better explained the theoretical background leading to our hypotheses, how our analyses tie in together, and the implications of our findings, to convince the reviewer that the current study does not only reflect a labor-intensive effort but also an important contribution to our understanding of E2 effects.

We believe that our data contribute to the understanding of the effects of E2 on memory-related processes in the human brain. We show that memory-dependent hippocampal activity in humans responds in a monotonically positive linear fashion to E2 dose until supraphysiological E2 levels are reached. At these levels, our results importantly show an inflection in the dose-response curve, resulting in an inverted U-shape. This result implies that increased E2 levels occurring during the menstrual cycle are beneficial to hippocampal activity, and only become disadvantageous at pregnancy levels. In a whole-brain analysis we see E2 affecting memory-related activity only in hippocampal areas.

We also found that E2 administration did not enhance hippocampus-dependent memory performance in a highly estrogen-sensitive sample (i.e. young, naturally cycling women). Our tightly controlled design and the wide range of induced E2 levels suggest that the lack of relationship found between E2 levels and memory performance very likely reflects a true null effect rather than confounds or noise introduced by the experimental setup. To illustrate how unlikely it is that there is any effect of E2 on recognition memory, we added a figure showing scatter plots for brain activity related to hippocampus-dependent memory (recollection) and hippocampus-independent memory (familiarity; please see below). We therefore suggest that a 24-hour period was long enough for an increase in E2 levels to increase synapse density and the associated glutamate release, as reflected by the fMRI signal, but too short for the new synapses to be functionally effective. This interpretation would also imply that the pre-ovulatory E2 peak in the human menstrual cycle might be too short – at least in some women, as there is large variability in cycles – to result in mnemonic changes, which could explain some of the inconsistencies across studies (Protopopescu et al., 2008; Resnick et al., 1998; Fehring et al., 2006).

Additionally, despite the wide range of E2 levels we induced in our tightly controlled experiment, we did not observe E2-related changes in amygdala activity, suggesting that menstrual-cycle related changes in affective processing related to amygdala activity should perhaps be attributed to progesterone, which also fluctuates across the menstrual cycle (Toffoletto et al., 2014; van Wingen et al., 2007 a,b; Akwa et al., 1999).

Finally, we learn that dose-dependent effects of E2 are not entirely confined to hippocampal regions. Activity associated with emotional processing and the emotional enhancement of memory effect showed E2 modulation within the precuneus, the brainstem, and the insula.

Effectively, our findings propose relevant areas to be further studied in animal studies, which are practically limited to pre-specified target brain regions.

Figure 3. Robust regression analyses between recognition memory performance and Day 1 to Day 2 increases in salivary 17-beta-estradiol (E2; N=123). Neither hippocampus-dependent memory (recollection; A.) nor hippocampus-independent memory (familiarity; B.) shows significant relationships to salivary Day 1 to Day 2 E2 increase. Colors represent experimental groups [all ρ_{corr} 's > .2].

Akwa, Y., Purdy, R. H., Koob, G. F. & Britton, K. T. The amygdala mediates the anxiolytic-like effect of the neurosteroid allopregnanolone in rat. *Behav. Brain Res.* 106, 119–125 (1999).

Fehring, R. J., Schneider, M. & Raviele, K. Variability in the Phases of the Menstrual Cycle. *Journal of Obstetric, Gynecologic, & Neonatal Nursing* 35, 376–384 (2006).

Protopopescu, X. et al. Hippocampal structural changes across the menstrual cycle. *Hippocampus* 18, 985–988 (2008).

Resnick, A. Neuropsychological performance across the menstrual cycle in women with and without premenstrual dysphoric disorder. *Psychiatry Research* 77, 147–158 (1998).

Toffoletto, S., Lanzenberger, R., Gingnell, M., Sundström-Poromaa, I. & Comasco, E. Emotional and cognitive functional imaging of estrogen and progesterone effects in the female human brain: a systematic review. *Psychoneuroendocrinology* 50, 28–52 (2014).

van Wingen, G. et al. How Progesterone Impairs Memory for Biologically Salient Stimuli in Healthy Young Women. *Journal of Neuroscience* 27, 11416–11423 (2007).

van Wingen, G. A. et al. Progesterone selectively increases amygdala reactivity in women. *Mol Psychiatry* 13, 325–333 (2007).

Response to Reviewer #2

Comment 1: Please provide a table with a breakdown of age, education and parity in the manuscript (or supplement). These data, excluding parity (see below) are analyzed but are not given in the manuscript itself. Is there a difference in parity between groups? Parity can alter menstrual cycle and estradiol levels, and while it likely may not have a role it would be important to verify that there are no significant differences between groups.

Response: We thank the reviewer for directing our attention to this point. There was no difference in parity [$\chi^2(4, N=125)=.94, p=.919$] between groups. We added this information to the methods section and a table with the descriptive and inferential statistics for the respective variables to the Supplementary Material (Supplementary Table 1).

Supplementary Table 1. Descriptive and inferential statistics of sample characteristics.

	experimental group										statistical analyses	
	0 mg		2 mg		4 mg		6 mg		12 mg		F	p
age (years)	25.57 ±	3.41	26.38 ±	3.85	26.50 ±	4.03	25.48 ±	4.15	25.83 ±	3.71	.34	.848
weight (kg)	61.47 ±	6.58	62.90 ±	8.62	64.85 ±	10.72	60.76 ±	8.82	66.17 ±	10.34	1.69	.157
bmi	21.83 ±	2.33	22.17 ±	2.14	23.48 ±	3.68	21.75 ±	2.93	22.71 ±	3.35	1.39	.242
											χ^2	p
education (n cases)											14.6	.558
~10 y	0		0		2		0		1			
~11 y	1		0		0		0		0			
~12 y	2		1		0		1		1			
~13-16.5 y	18		13		8		16		15			
> 16.5 y	9		7		10		8		12			
nulliparity	28		19		19		23		28		.94	.919

All analyses are based on N=125. BMI = body-mass-index. Women were considered as 'nulliparous' when they never have been pregnant for longer than 8 weeks.

Comment 2: The inter-individual variance within groups was high but also interesting. Even if the groups didn't differ by age – is there an age effect in the ability of E2 to increase E2? I am wondering if the variability comes from age-related changes in menstrual cycle fluctuations in E2/FSH?

Response: This is an interesting point to consider. We calculated a multiple robust regression analysis using mg/kg, age and the interaction term mg/kg*age as predictors for salivary Day 1 to Day 2 increase. However, neither age [$t(123)=-0.04, p=.964, \beta_{std}=-.002$] nor mg/kg* age [$t(123)=-0.68, p=.501, \beta_{std}=-.027$] reached significance. Interestingly, a high variability in resulting E2 levels has also been reported for transdermal patches in women and even subcutaneous capsules in rodents (Talboom et al., 2008; Garza-Meilandt et al., 2006). These observations highlight the importance of directly measuring hormone levels in future studies. We added the analyses to the Supplementary Material Section 1.1.

Garza-Meilandt, A., Cantu, R. E. & Claiborne, B. J. Estradiol's Effects on Learning and Neuronal Morphology Vary With Route of Administration. *Behavioral Neuroscience* 120, 905–916 (2006).

Talboom, J. S., Williams, B. J., Baxley, E. R., West, S. G. & Bimonte-Nelson, H. A. Higher levels of estradiol replacement correlate with better spatial memory in surgically menopausal young and middle-aged rats. *Neurobiology of Learning and Memory* 90, 155–163 (2008).

Comment 3: In the first sentence of Discussion and throughout the Introduction the authors refer to Linear versus quadratic...wouldn't this also depend on where on the curve you were? So for example if your dose or range of E2 isn't large enough you may expect to see a linear relationship but if your doses were larger you could move into the quadratic part of the curve. I just find this line of reasoning a little odd as it may just be due to the limited range of E2 choices per paper?

Response: We would like to thank the reviewer for directing our attention to the superficial and hence massively oversimplified way we presented our hypotheses and results (see also comment 1 of reviewer #1). In particular, we agree that comparing dose-response curves across studies is meaningless when it is not related to the induced range of estrogen levels.

In response to the reviewer's comment we substantially rewrote the introduction and discussion. Most importantly, we explicitly state in the revised introduction the dose-range that led to different response curves and specified based on that the hypothesis we tested in the current study.

We copied the most relevant parts of the modified introduction and discussion but would like to refer the reviewer to the revised manuscript in order to read these changes in context.

Introduction

"(...) the inconsistency in reported response functions might be resolved with the observation that neural plasticity monotonically improves with increasing E2 levels mostly at physiological E2 concentrations^{6,8–13}. Critically, only when supraphysiological E2 levels (i.e. levels above estrous cycle peaks) are reached does the dose-response relationship exhibit an inverted U-shape^{6,8,9,12,13}. However, a few studies have reported monotonic improvements also at clearly supraphysiological E2 levels, and some effects only occur at supraphysiological levels^{5–7} (...)

With respect to learning and memory, the diversity of regimens and tasks makes it more difficult to see a clear pattern: inverted U-shaped curves might already characterize high physiological E2 levels, and null effects and even monotonic decreases have been reported^{6,15–18}. The question whether the response to E2 turns into an inverted U even at physiological levels or only at supraphysiological levels is of relevance because only in the first case would E2 regularly reach concentrations disadvantageous to memory and learning during the normal menstrual cycle."

Discussion:

"Critically, the inflection point of the quadratic function in the hippocampus and the insula were clearly above the physiological levels of E2 reached during the menstrual cycle. It should be emphasized that the procedure of fitting a quadratic in addition to a linear function to the continuously distributed levels of E2 increase was strictly data-driven and did not require any a priori assumptions about the inflection point. Our

results therefore support the hypothesis that inverted U-shape dose-dependent responses for E2 and neural activity are only observed at supraphysiological levels. In other words, within the range of E2 levels experienced during the menstrual cycle, higher levels of E2 were increasingly beneficial also in the areas that exhibit an inverted U-shaped response function at supraphysiological levels. E2 effects on neural activity returned to baseline in these areas only at levels that occur during early to mid-pregnancy. In other brain areas, monotonically increasing but also decreasing response functions were observed across the full range of E2 levels.

(...)

Various mechanisms have been suggested to underlie the inverted U-shaped curves of rapid and genomic effects of E2. The inverted U-shaped response in rapid effects might reflect the optimal level of receptor activation¹⁴. Alternatively, different doses might lead to differences in coupling to distinct pathways via changes in receptor conformation, or the activated kinases might be inhibited at higher doses^{3,14}. Moreover, the activation of ER β in hippocampal interneurons at higher E2 levels can decrease neural activity⁹. Genomic effects might show an inverted U-shaped response due to differences between ER α and ER β . In particular, whereas ER α activation stimulates synaptogenesis, ER β activation and an increase in ER β levels result in a decrease in synaptogenesis^{50–53}. As the affinity of ER α for E2 is higher, low levels of E2 will mostly activate ER α , which is transcriptionally more active. In contrast, at higher levels, more of ER β are activated. The amount of synaptogenesis may additionally depend on the relative expression of ER α /ER β . ER α is overall less expressed in the hippocampus and its transcription is upregulated by E2, so that its expression increases with E2 levels. However, with higher E2 levels, activated ER β downregulates ER α -dependent transcription³. Interestingly, consistent with our finding of inverted U-shaped functions only at supraphysiological levels, it has been suggested that E2 levels within the physiological range promotes ER α expression in the hippocampus, whereas at supraphysiological levels, ER α expression decreases⁵⁴. “

Comment 4: E2 also biases strategy use as Korol and Galea’s groups have shown that higher E2 is associated with a greater likelihood of using a hippocampal based strategy (even if they may not be as good at it).

Response: The Reviewer raises a very pertinent point here which we considered when we designed the experiment. In particular, for this reason we selected a memory task that allows to disentangle hippocampus-dependent from hippocampus-independent contributions to memory. This dissociation is not only well-established in humans but also in the animal literature (Brown et al., 2010). Interestingly, although we did not observe behavioral differences for hippocampus-dependent memory, E2 was related to hippocampal activity associated to the remember/know contrast (i.e. hippocampus-dependent memory formation) but not general encoding success.

In response to the reviewer’s comment we now explain the rationale for the chosen task more explicitly in the introduction:

“Most animal studies have investigated E2 effects on hippocampal neurons and hippocampus-dependent learning and memory. It also has been reported to influence

the shift from hippocampus-independent to -dependent learning strategies^{34,35}. Therefore, we employed a memory paradigm that allowed us to disentangle hippocampus-independent from hippocampus-dependent recognition memory^{36,37}. 'Recollecting' an item, i.e. together with contextual information about the encoding episode, relies critically on the hippocampus, whereas the acontextual sense of 'familiarity' with an item relies on parahippocampal structures and is independent of the hippocampus³⁶.

And discussion:

"Interestingly, these relationships were specifically observed for brain activity related to hippocampus-dependent memory (i.e. the remember>know contrast) but not memory formation in general (general encoding success contrast)."

Comment 5: On page 3, lines 3-5 neurosteroids are referenced as 7-9 (but did you mean 9-11?) as 7 and 8 refer to distribution of ER subtypes. I think you mean 10 and 11..but clearly according to Kato's and Caruso work plasma levels follow brain levels of steroids and likely greater influence is gonadally-derived (or peripherally-derived estradiol rather than de novo synthesis which is not always seen in adult female rodents (see Barker, Caruso). Because you are not directly testing this I suggest removing this part or toning down your discussion. There is a lot of attention on neurosteroids when in reality there are a number of very distinct changes when ovaries are removed (in women and in rodents) on cognition.

Response: We thank the reviewer for the thorough reading of our manuscript and apologize for the mistake in the citations. We also agree that this part does not contribute to describe the rationale of our study and therefore removed it from the manuscript.

Comment 6: Was there a correlation between serum and saliva levels of E2 (even though one is free and one is bound, one would still expect some correlations?)

Response: The reviewer is right in his/her expectation. The Day 1 to Day 2 salivary E2 increase showed a high correlation with the Day 1 to Day 2 in serum E2 [$r(90)=.825, p<.001$], confirmed by linear robust regression analysis [$t(90)=28.61, p<.001, \beta_{std}=1.04$]. We added this information to the results section.

Comment 7: P values should be in figure captions as well as in text.

Response: We added p-values to the figure captions.

Comment 8: Reference 20 is missing an author

Response: We apologize for this oversight and corrected the reference.

Comment 9: Page 15, line 19 – reference 64 is given but this same relationship is seen with cell proliferation and E2 in the hippocampus (Barha, Tanapat)

Response: We agree with the reviewer. However, due to the substantial rewriting of the discussion, the particular sentence was removed from the manuscript.

Comment 10: P16, lines 19-24 – the same dose response is also seen in relationship to working memory in the radial arm maze or in the non-spatial delayed alternation T maze (Holmes, Wide, Galea, 2002; Wide et al., 2004) – it could be changes in task, but it also could be type of estradiol used as well or as you point out but may want to do so again – dose! Certainly when those studies have examined a variety of doses the data tend to show the curvilinear effect. It also matters on the timing of E2 to memory task (if measuring effects on consolidation or acquisition for example).

Response: We agree with the reviewer. However, due to the substantial rewriting of the discussion, the particular sentence was removed from the manuscript. Task specific effects of E2 are now discussed in the following sentence:

“Similarly, despite the many studies providing evidence for a positive relationship between hippocampus-dependent memory and E2 levels within a physiological range in rodents^{1,2}, some studies report this effect to be task-specific, while others did not find any effects on behavior^{15,16,18,62}.”

Comment 11: P16, line 24 - Please briefly expand on the idea that P or menopausal status may contribute to the variability of findings. You could also refer to study by Baxter and others in primates on the negative impact of P with E.

Response: We agree with the Reviewer that this point is important to emphasize, because most of the studies in humans either compare different menstrual cycle phases or investigate pharmacological interventions in menopausal women. For space reason we restricted the discussion of these confounds to the effects of progesterone and added the following sentences to the introduction and discussion:

Introduction

“For a variety of reasons, these studies can only be compared, to a limited degree, to the placebo-controlled, randomized E2 treatment regimens in young, healthy animals. Across the menstrual cycle, for instance, neuroactive progesterone also fluctuates, potentially and unpredictably interacting with E2²⁹.”

Discussion

“However, our current finding, that a well-controlled increase across a wide range of E2 levels does not result in changes in amygdala activity, suggests that these previous findings of changes in amygdala activity should perhaps be attributed to progesterone, which also fluctuates across the menstrual cycle. Indeed, amygdala reactivity to emotional stimuli is reliably modulated by progesterone, and its metabolite allopregnanolone has anxiolytic effects in rats via its action on the central nucleus of the amygdala^{68,69}.”

Comment 12: In your study your levels of P are consistent across groups. Relative to pregnancy or mid-luteal what are these levels of P like?

Response: On average, we had P levels of about 112.78 (SD=103.31) pg/ml in our study. This is considerably lower than P levels at mid-luteal peak (M=1142, SD=174 pg/ml; Luisi et

al., 1981) and during pregnancy (1st trimester: $M=720$, $SD=90$, 2nd trimester: $M=1140$, $SD=110$, 3rd trimester: $M=1300$, $SD=140$ pg/ml; Ghalayani et al., 2013). We added this information to the caption of Supplementary Figure 1.

Luisi, M. et al. Radioimmunoassay for progesterone in human saliva during the menstrual cycle. *Journal of Steroid Biochemistry* 14, 1069–1073 (1981).

Ghalayani, P., Tavangar, A., Nilchian, F. & Khalighinejad, N. The comparison of salivary level of estrogen and progesterone in 1st, 2nd and 3rd trimester in pregnant women with and without geographic tongue. *Dent Res J (Isfahan)* 10, 609–612 (2013).

Comment 13: Page 18, line 5 – subtype of ER?

Response: Both ER-subtypes have been detected in the brainstem and the parietal cortex (Kritzer, 2002; Vanderhorst et al., 2005). We added this information to the discussion.

Kritzer, M. F. Regional, Laminar, and Cellular Distribution of Immunoreactivity for ER α and ER β in the Cerebral Cortex of Hormonally Intact, Adult Male and Female Rats. *Cereb Cortex* 12, 116–128 (2002).

Vanderhorst, V. G. J. M., Gustafsson, J.-Å. & Ulfhake, B. Estrogen receptor- α and - β immunoreactive neurons in the brainstem and spinal cord of male and female mice: Relationships to monoaminergic, cholinergic, and spinal projection systems. *J. Comp. Neurol.* 488, 152–179 (2005).

Response to Reviewer #3

Comment 1: First sentence of the Abstract as well as the very first sentence of the Introduction are rather awkward. Authors should consider re-writing/simplifying. In general, the manuscript, although fairly well written, contains awkward language and should be proof-read. [example: tasks were administered? not accomplished... (page 5, line 16); “Beneath statistical tests...” (page 5, line 23)]

Response: We apologize for this shortcoming and rewrote the respective sentences. In general, a native speaker (G. Joue) helped us improving grammar and language of the revised manuscript.

Comment 2: Given the complexity of the study – it would be good to be very specific throughout the manuscript when it comes to measures used to assess various function, etc.

Response: We agree with the Reviewer, added more details to the schematic overview in Figure 1 and rewrote the respective parts from the introduction, methods and results section to avoid any ambiguities.

Figure 1. Study design. Volunteers participated in the study on three consecutive days. In the evening of Day 1, baseline hormone concentrations and mood were assessed, followed by the first dose of either 17-beta-estradiol (E2; 2, 4, 6 or 12 mg) or placebo, depending on their assigned group (double-blind). On the morning of Day 2, participants took the second dose on their own. In the afternoon of Day 2, when E2 levels were expected to peak, participants encoded emotional pictures inside the scanner (upper red box). In the evening of Day 3, participants performed a recognition test (lower red box) and rated the encoded pictures for arousal. See Supplementary Material Section

2.1 for more details on the timeline of the study.

Examples:

Introduction

“Therefore, we employed a memory paradigm that allowed us to disentangle hippocampus-independent from hippocampus-dependent recognition memory^{36,37}. ‘Recollecting’ an item, i.e. together with contextual information about the encoding episode, relies critically on the hippocampus, whereas the acontextual sense of ‘familiarity’ with an item relies on parahippocampal structures and is independent of the hippocampus³⁶”

Results

“Only the first component (dimensions: alertness, strength, energy, clear-headedness; see Supplementary Table 13), determined by a principal component analyses on data from the mood questionnaire, showed a non-significant trend after Bonferroni correction [$t(121)=2.31$, $p_{corr}=.092$, $\beta_{std}=.188$]. We will refer to this principal component as ‘subjective vigilance.’ To test whether changes in subjective vigilance are reflected in an objective measure of alertness, individual reaction times from the distractor task (i.e. arrow task) and the encoding task were used as a proxy for intrinsic alertness.”

“Independent linear and quadratic regression analyses on the whole brain were then performed to identify areas showing linear or inverted U-shaped relationships between E2 increase and recollection-related activity. These were performed in SPM using the remember>know whole brain contrast images as the dependent variable and the increases in E2 from Day 1 to Day 2 as the independent variable. “

Discussion

“We identified an inverted U-shape in the medial posterior hippocampus as well as a monotonically increasing dose-response function across the full range of E2 levels in the lateral posterior hippocampus for activity associated with hippocampus-dependent memory formation. Interestingly, these relationships were specifically observed for brain activity related to hippocampus-dependent memory (i.e. the remember>know contrast) but not memory formation in general (general encoding success contrast).”

Methods

“Linear and inverted U-shaped responses to E2 increases from Day 1 to Day 2 ($E2_{Day2-Day1}$) were tested using a hierarchical robust regression approach run in MATLAB 2014b (Mathworks, Inc, Natick, MA, USA; fitlm function, robust fitting option).”

“To investigate the effects of E2 on memory and affective processing, the following 4 contrasts, defined a priori, were calculated for each participant: 1. A remember/know contrast (remember > know) to specifically examine hippocampus-dependent

memory; 2. General encoding success (subsequent hit > miss); 3. The effects of emotional processing (negative > neutral); 4. EEM (valence [negative > neutral] × general encoding success [hit > miss]). “

Comment 3: Authors give some specific examples of u-shaped dose-response curve findings in the literature, but not linear. Introduction could benefit from both.

Response: We agree with the reviewer and rewrote the introduction (and discussion) much more specific with respect to linear and U-shaped effects. In particular, we cite now literature for both response functions:

“Interestingly, based on studies that have induced more than one level of E2 it was proposed that the dose-response relationship between E2 and neural plasticity as well as behavior might not follow the traditional sigmoidal but rather an inverted U-shaped function¹⁻³. That is, one would expect increasing benefits until medium levels of E2 but then a decrease and none or even negative effects at high levels. However, some studies have also observed the traditional sigmoidal or even linearly increasing dose-response relationships⁵⁻⁷.

Taking the heterogeneity of these studies into account, the inconsistency in reported response functions might be resolved with the observation that neural plasticity monotonically improves with increasing E2 levels mostly at physiological E2 concentrations^{6,8-13}. Critically, only when supraphysiological E2 levels (i.e. levels above estrous cycle peaks) are reached does the dose-response relationship exhibit an inverted U-shape^{6,8,9,12,13}. However, a few studies have reported monotonic improvements also at clearly supraphysiological E2 levels, and some effects only occur at supraphysiological levels⁵⁻⁷. Whereas the monotonic increase and saturation is compatible with the traditional sigmoid dose-response function of pharmacological studies, the inverted U-shaped curve needs to be explained by a more intricate pattern of underlying processes. It has been proposed that the different characteristics of the estrogen receptors and their interaction accounts for this more complex response curve^{3,14}.

With respect to learning and memory, the diversity of regimens and tasks makes it more difficult to see a clear pattern: inverted U-shaped curves might already characterize high physiological E2 levels, and null effects and even monotonic decreases have been reported^{9,15-18}. The question whether the response to E2 turns into an inverted U even at physiological levels or only at supraphysiological levels is of relevance because only in the first case would E2 regularly reach concentrations disadvantageous to memory and learning during the normal menstrual cycle.”

Comment 4: Authors mention potential effects of E2 on blood flow in carotid arteries – although it turns out that authors do not observe any effects, it should be discussed what the anticipated effects and related concerns were for those not familiar with the literature.

Response: We agree with the reviewer that the rationale of assessing effects of E2 on brain perfusion should be addressed more explicitly. In particular, the assessment of neural activity by fMRI relies on the blood-oxygen-level dependent (BOLD) effect, which is a temporal increase in oxyhemoglobin in active brain regions. This functional hyperemia is

caused by an increase in blood flow to supply the tissue with glucose for aerobic glycolysis which is accompanied by a parallel decrease in the oxygen extraction fraction resulting in the increase in regional oxyhemoglobin levels. Because E2 has vasomodulatory effects and can dilate cerebral arteries (Duckles and Krause, 2007) we were concerned that these unspecific, i.e. not task evoked effects might influence the BOLD-effect. In particular, a greater baseline perfusion could result in a smaller relative task related increase, and could potentially bias thus our measure of neural activity associated to the memory task. We added this rationale to the ASL methods section, which has been moved completely to the Supplementary Material section 2.4 for space reasons.

Duckles, S. P. & Krause, D. N. Cerebrovascular effects of oestrogen: multiplicity of action. *Clin. Exp. Pharmacol. Physiol.* 34, 801–808 (2007).

Comment 5: I may have missed - why were serum E2 levels assessed in only a subset of women? And did the distribution vary across groups?

Response: We apologize for this missing information. In general, salivary E2 are considered to be in an equilibrium with the free, unbound E2 in blood and in hence a good marker for the bio-active E2 (Fiers et al., 2017). Serum levels were assessed only to make our data also comparable to studies that assessed only total serum E2. For instance, E2 reference values (e.g. for pregnancy) are typically better available for serum than for saliva. However, not all women were willing to contribute blood, respectively it was not possible to draw blood from all participants after three attempts. The distribution of available serum levels did not significantly differ between groups [$\chi^2(4, N=125)=5.55, p=.235$]. We added the number of available serum levels and inferential statistics to Supplementary Table 12 and the reason for missing serum levels to the methods section.

Supplementary Table 12. Number of cases included in each analysis.

	n per experimental group					N	statistical analyses of missing values	
	0	2	4	6	12	Σ	χ^2	p
salivary hormone levels	30	21	20	25	29	125	-	-
serum E2	23	19	12	17	21	92	5.55	.235
behavioral analyses	30	21	20	25	27	123	6.73	.151
MRI								
functional imaging: negative vs. neutral, hits vs. misses, EEM contrast	30	20	20	25	26	121	7.42	.115
functional imaging: remember vs. know	29	20	20	24	25	118	5.31	.257
arterial spin labeling	28	19	20	25	25	117	6.07	.194

Fiers, T., Dielen, C., Somers, S., Kaufman, J.-M. & Gerris, J. Salivary estradiol as a surrogate marker for serum estradiol in assisted reproduction treatment. *Clin. Biochem.* 50, 145–149 (2017).

Comment 6: Measures of vigilance and alertness (page 6, lines 6+7) - what are those measures exactly? More specific language could be used throughout.

Response: We apologize for the confusion caused by unspecific language ‘Subjective vigilance’ refers to the first component of the PCA (originally labeled as ‘Factor 1’) derived from the mood questionnaire. ‘Objective alertness’ was meant to refer to reaction times from the distractor task (‘arrow’ task; previously labeled as ‘active baseline task’). We removed the term ‘objective alertness’ completely and rewrote the respective sentences. Moreover, we added analyses using accuracy in the arrow task as well as reaction times and accuracy in the encoding task (inside/outside) decision as further proxies for alertness in response to Reviewer #1.

“Only the first component (dimensions: alertness, strength, energy, clear-headedness; see Supplementary Table 13), determined by a principal component analyses on data from the mood questionnaire, showed a non-significant trend after Bonferroni correction [$t(121)=2.31$, $p_{corr}=.092$, $\beta_{std}=.188$]. We will refer to this principal component as ‘subjective vigilance.’ To test whether changes in subjective vigilance are reflected in an objective measure of alertness, individual reaction times from the distractor task (i.e. arrow task) and the encoding task were used as a proxy for intrinsic alertness. Neither reaction times (rt) nor accuracy in the arrow task [rt: $t(121)=-.68$, $p_{corr}>.999$, $\beta_{std}=-.011$; accuracy: $t(121)=-0.20$, $p_{corr}>.999$, $\beta_{std}=-.004$] or the encoding task (indoor/outdoor decision) [rt: $t(121)=-.57$, $p_{corr}>.999$, $\beta_{std}=-.047$; accuracy: $t(121)=-.37$, $p_{corr}>.999$, $\beta_{std}=-.023$] were significantly related to the Day 1 to Day 2 increase in salivary E2.”

We also acknowledge that there are more text passages that would benefit from more specific language to enhance the readability of our manuscript. We revised the manuscript accordingly.

Comment 7: What does the linear relationship between salivary E2 increase and neural pictures mean (page 9, Figure 3)?

Response: We agree that the significant relationship between E2 and arousal ratings for neutral but not negative pictures appears to be puzzling at first sight. First, we would like to point out how this result corroborates findings from a previous study, and then how it offers explanations for two of our hypotheses.

As we point out in the discussion, a similar pattern of arousal ratings for neutral and negative pictures was observed in a previous study on hormone replacement therapy (HRT; Pruis et al., 2009). However, whereas we found a more reliable E2 effect on neutral picture arousal ratings, they found more reliable effects with negative pictures. The differences in our results might be explained by differences in the baseline levels of arousal as well as the smaller sample size and lower E2 doses in the other study. Particularly remarkable are the baseline differences, the negative pictures in our study were rated substantially more arousing even in the placebo group than in Pruis et al. (2009), very likely leading to a ceiling effect.

In order to understand why higher E2 levels affect arousal for neutral pictures, we tested the following two hypotheses. First, E2 might increase the ‘emotional carry effect’, that is, arousal elicited by a negative picture could influence the rating of the next neutral picture. To test this possibility, individual arousal ratings for neutral stimuli following another neutral

stimulus ('neutral-after-neutral') and neutral stimuli directly following a negative stimulus ('neutral-after-negative') were each averaged for every participant. Consistent with an emotional carry-over effect, neutral-after-negative stimuli ($M=2.89$, $SD=1.19$) were generally perceived as more arousing than neutral-after-neutral stimuli [$M=2.79$, $SD=1.22$; $t(122)=3.761$, $p<.001$]. However, robust regression analyses showed significant associations between salivary increase in E2 from Day 1 to Day 2 and arousal ratings for both neutral-after-negative [$t(121)=2.75$, $p_{corr}=.014$, $\beta_{std}=.261$] as well as neutral-after-neutral [$t(121)=2.65$, $p_{corr}=.018$, $\beta_{std}=.251$] pictures. Moreover, the emotional carry-over effect (arousal neutral-after-negative – arousal neutral-after-neutral) was not significantly associated to E2 [$t(121)=.190$, $p_{corr}>.999$, $\beta_{std}=.017$]. Thus, although carry-over effects were present in our data, they did not explain the effects of E2 on arousal ratings for neutral pictures.

Second, it is possible that viewing negative pictures might induce a negative mood over time, which in turn leads to greater arousal also with neutral pictures ('mood induction effect'), an effect which might be increased by E2. Evidence that the mood induction effect could be enhanced by E2 is provided by Pruis and colleagues (2009; 2nd experiment). The authors found that women under HRT (compared to those not undergoing HRT) rated the neutral beginning of a story as less arousing than the also neutral third section, which was preceded by a negative story section. Our additional analyses corroborated the mood-induction hypothesis: arousal ratings for neutral pictures from the first third of the pictures seen were significantly lower than from the last third [$t(122)=-2.19$, $p=.030$]. Critically, the difference of arousal ratings for neutral pictures (last third – first third) was positively associated with salivary E2 Day 1 to Day 2 increase [$t(121)=3.09$, $p_{corr}=.005$, $\beta_{std}=.148$]. In other words, the higher the increase in salivary E2, the higher the increase in arousal ratings for neutral pictures over time. As such, it is quite plausible that the increase in E2 increased mood induction effects and thereby arousal ratings for neutral pictures.

In response to the reviewer's comment we added these additional analyses to the Supplementary Material Section 1.5 and revised the discussion as follows:

"The increase of arousal for the neutral stimuli might be caused by a more pronounced induction of negative mood in response to the negative pictures with higher E2 levels, as further results of Pruis and colleagues⁶⁷ and our own supplementary analyses suggest (Supplementary Material Section 1.5). This finding is consistent with reports that supraphysiological doses of E2 can increase depression-like behavior and anxiety in rats^{19,22}."

Pruis, T. A., Neiss, M. B., Leigland, L. A. & Janowsky, J. S. Estrogen modifies arousal but not memory for emotional events in older women. *Neurobiology of Aging* 30, 1296–1304 (2009).

Comment 8: I may have missed this, but 2 subjects were lost, so how many total per each group?

Response: We added the number of participants per group to Supplementary Table 12 (see Response 5). Moreover, in response to Reviewer 4, we added hormone values for the two participants for whom behavioral and imaging data are missing.

Comment 9: Authors refer to “task-related effects of E2” several times on page 15. What is a “task-related effects of E2”, should be defined much earlier in the paper and I apologize if I missed this. The manuscript in general uses a lot of jargon, which makes it harder for the reader to get a complete picture, especially given the multiple aspects of this complex study – the reader may not be an expert on all.

Response: We apologize for the confusion. The term ‘task-related effects of E2’ was meant to refer to the effects of E2 on brain activity elicited by the memory encoding task (i.e. the 4 a priori defined contrasts) compared to unspecific effects of E2 on brain perfusion. We aimed to avoid the use of jargon in the revised manuscript with the help of our native English proof reader.

In response to the reviewer’s comment we rephrased the particular sentences:

Abstract

“Memory-related activity in distinct posterior hippocampal clusters both linearly increased as well as followed an inverted U-shape with E2 dose.”

Discussion

“Of note, we did not observe an overlap between effects of E2 on any BOLD activity elicited by the emotional memory task and the subtle differences in brain perfusion during rest as measured with ASL. This implies that E2 effects on brain activity as measured with fMRI in the current study are not confounded by general task-unrelated effects of E2 on brain perfusion. Moreover, the effects of E2 on memory and affective processing-related BOLD activity were specific to a restricted set of brain regions, i.e. the hippocampus, the precuneus, brainstem and insula, although the task evoked activity in wide networks of areas (Supplementary Figure 4, Supplementary Tables 8 & 9). This suggests that E2 had no general effect on the neurovascular coupling underlying the BOLD effect but that the observed differences in the BOLD signal indeed reflect E2 effects on synaptic activity.”

Comment 10: Hippocampus-dependent and –independent memory processes should be defined more specifically and early in the manuscript.

Response: We agree with the reviewer and added a definition of hippocampus-dependent and –independent memory processes to the introduction:

“Therefore, we employed a memory paradigm that allowed us to disentangle hippocampus-independent from hippocampus-dependent recognition memory^{36,37}. ‘Recollecting’ an item, i.e. together with contextual information about the encoding episode, relies critically on the hippocampus, whereas the acontextual sense of ‘familiarity’ with an item relies on parahippocampal structures and is independent of the hippocampus³⁶.

Comment 11: I find the explanation of findings of both linear and inverted u-shape curve between E2-increase and memory-related activity somewhat unsatisfactory.

Response: We agree and discuss the cellular mechanisms that have been proposed to underlie the inverted U-shape as well as potential reasons for the existence of both types of

response curves more in detail:

“Various mechanisms have been suggested to underlie the inverted U-shaped curves of rapid and genomic effects of E2. The inverted U-shaped response in rapid effects might reflect the optimal level of receptor activation¹⁴. Alternatively, different doses might lead to differences in coupling to distinct pathways via changes in receptor conformation, or the activated kinases might be inhibited at higher doses^{3,14}. Moreover, the activation of ER β in hippocampal interneurons at higher E2 levels can decrease neural activity⁹. Genomic effects might show an inverted U-shaped response due to differences between ER α and ER β . In particular, whereas ER α activation stimulates synaptogenesis, ER β activation and an increase in ER β levels result in a decrease in synaptogenesis⁵⁰⁻⁵³. As the affinity of ER α for E2 is higher, low levels of E2 will mostly activate ER α , which is transcriptionally more active. In contrast, at higher levels, more of ER β are activated. The amount of synaptogenesis may additionally depend on the relative expression of ER α /ER β . ER α is overall less expressed in the hippocampus and its transcription is upregulated by E2, so that its expression increases with E2 levels. However, with higher E2 levels, activated ER β downregulates ER α -dependent transcription³. Interestingly, consistent with our finding of inverted U-shaped functions only at supraphysiological levels, it has been suggested that E2 levels within the physiological range promotes ER α expression in the hippocampus, whereas at supraphysiological levels, ER α expression decreases⁵⁴.

Strikingly, we not only found an inverted U-shaped relationship in the lateral posterior hippocampus but also a monotonically increasing response curve in a more medial posterior hippocampal area. An increase in hippocampal plasticity at supraphysiological levels of E2 has similarly been observed in animals^{9,12,13,55}. Although at first sight it seems surprising that we observed both types of response curves in distinct hippocampal clusters, previous studies have also described dissociable dose-response functions for different hippocampal subfields (CA1 vs. CA3) and depending on whether hippocampal subregions or whole hippocampus is considered (whole vs. rostral hippocampus)^{5,9}. It has been suggested, that such region-specific dose-response functions could be a result of different ER α /ER β ratios^{5,51}, which vary depending on hippocampal subregion⁵⁶⁻⁵⁹. Further explanations for the observed responses could lie in the differences in how E2 regulates ER α transcription in CA1 and CA3 (as suggested by the transcription patterns describe by Mendoza-Garcés and colleagues⁵⁷), or subfield-specific differences in nuclear E2 uptake⁶⁰. In fact, the inhibition of aromatase activity affects CA1 and CA3 differently⁶¹. However, the spatial resolution of our fMRI scans is too coarse to relate the different dose-response curves to specific anatomically defined hippocampal subfields and only permits a functional dissociation.

These mentioned differences in hippocampal subregions would also explain why the E2-BOLD signal relationship continues to increase at all at supraphysiological E2 levels in the more medial posterior hippocampus. A higher ER α /ER β ratio, as reported for instance in CA3 compared to CA1⁵⁹, could result in a shift of the inflection point of the dose-response curve at higher, supraphysiological E2 levels that were above the range induced in the present study. An alternative explanation for the different regional dose effects could be that the observed changes of the BOLD signal

with increasing E2 levels reflect distinct underlying neuronal actions of E2 in both hippocampal clusters, such as genomic regulation of synaptogenesis vs. the rapid effects on glutamatergic synaptic activity, each of which might manifest as different response functions.

Comment 12: At a later point in the manuscript, authors suggest the lack of spatial resolution elsewhere in the brain with sequences employed when addressing secondary aims – and the resolution required to delineate hippocampal subfields is rather challenging.

Response: We agree that the term ‘hippocampal subregional activity’ could be misleading as the spatial resolution of our fMRI-scans did not allow to relate activity patterns to specific hippocampal subfields. We now avoid the term ‘subregion’ when discussing our results and instead speak of ‘different/distinct hippocampal clusters’, e.g.:

“Although at first sight it seems surprising that we observed both types of response curves in distinct hippocampal clusters, previous studies have also described dissociable dose-response functions for different hippocampal subfields (CA1 vs. CA3) and depending on whether hippocampal subregions or whole hippocampus is considered (whole vs. rostral hippocampus) ^{5,9}.”

“However, the spatial resolution of our fMRI scans is too coarse to relate the different dose-response curves to specific anatomically defined hippocampal subfields and only permits a functional dissociation.”

“In conclusion, under tight experimental control in young, healthy women, we showed that the pharmacological increase of a wide range of physiological and supraphysiological E2 levels modulates memory-related activity differentially in distinct hippocampal clusters.”

Comment 13: I am not sure I agree with statement that non-human studies/tasks differ on motivation because often the animals are rewarded or punished. Same can be accomplished in humans by adding or subtracting points while performing the task, and many tasks do so. Humans are also generally motivated to do well, especially in front of others (i.e., when tested). Although there may be species-specific differences in how to accomplish those, I would not discount their presence in human studies.

Response: This is a very pertinent point raised by the reviewer. We removed this paragraph from the discussion.

Comment 14: One big difference between human and non-human studies is that in most non-human studies, animals are OVXed, The action of estrogen in such a system may be fundamentally different from that in an intact system. Moreover, women in this study were all young and relatively healthy.

Response: We thank the reviewer for bringing up this substantial difference between our approach and animal research. Although several studies provide evidence of the beneficial effects of naturally fluctuating E2 in the course of the estrous cycle with respect to memory and spine density (Frye et al., 2007; Woolley & McEwen, 1993), we only know of one study

that has investigated the effects of E2 administration on memory in intact female rats (de Macêdo Medeiros et al., 2014). In the study of de Macedo Medeiros and colleagues (2014), drug treatment raised E2 plasma levels to 20 fold of that reached during the estrous cycle, which resulted in protecting the rats from a scopolamine-induced impairment in a plus-maze discriminative avoidance task but interference with memory consolidation. This finding is consistent with inverted U-shaped dose-response functions reported for supraphysiological E2 levels in OVX animals.

In response to the reviewer's comment we added the limited comparability with animal studies to the limitations section:

“Second, although being one of the major strengths of the current study, investigating isolated E2 effects in intact women has the drawback that it is currently unknown how these findings translate to phases of the menstrual cycle when fluctuations in E2 and P4 co-occur and to findings with ovariectomized animals. Nonetheless, similar neural activity has been observed in intact and ovariectomized rats treated with physiological E2 doses⁹.”

de Macêdo Medeiros, A. et al. Estrogen levels modify scopolamine-induced amnesia in gonadally intact rats. *Prog. Neuropsychopharmacol. Biol. Psychiatry* 53, 99–108 (2014).

Frye, C. A., Duffy, C. K. & Walf, A. A. Estrogens and progestins enhance spatial learning of intact and ovariectomized rats in the object placement task. *Neurobiol. Learn. Mem.* 88, 208–216 (2007).

Woolley, C. S. & McEwen, B. S. Roles of estradiol and progesterone in regulation of hippocampal dendritic spine density during the estrous cycle in the rat. *J. Comp. Neurol* 336, 293–306 (1993).

Comment 15: There is generally a lack of acknowledgements of the study's limitations. Moreover, likely small Ns per group and numerous comparisons should be more of a concern.

Response: We agree that a limitations section should be added to the manuscript. We did not, however, split our sample into groups for the main analyses. Power calculations show that a minimum N of 99 is required to detect ($p < .05$) a small to medium effect size of Cohen's $f^2 = .10$ at a statistical power of .80 in a regression analyses with two predictors. Moreover, significant behavioral effects (i.e. the relationship between salivary E2 levels with arousal ratings for neutral pictures and with the first principal component of the mood questionnaire) were Bonferroni-corrected for multiple comparisons. To address multiple comparisons in our fMRI data we applied family-wise error correction within predefined anatomical masks for hypothesized areas and within the whole brain analysis for BOLD effects in unexpected brain areas.

We therefore believe that we had sufficient statistical power –in particular due to the use of regression analysis maintaining continuous variables as continuous rather than binning into groups as in ANOVAs- and only used conservative corrections for multiple comparisons.

In response to the reviewer we added the following limitations:

“Despite the well-controlled E2 treatment regimen and carefully chosen experimental design, the current study faces some limitations. As with most animal studies on which we based our hypotheses, we could not measure hippocampal but only serum and saliva E2 levels. On the other hand, hippocampal and ovarian E2 syntheses are synchronized by the gonadotropin-releasing hormone so that peripheral and central

levels of E2 are highly correlated^{84,85}. However, it is unclear whether these synchronizing feedback mechanisms also function with exogenously increased E2 levels. Therefore, our conclusion that the response function of E2 turns into an inverted U-shape only at supraphysiological levels can only be applied to peripheral E2 levels until this interaction is clarified. Second, although being one of the major strengths of the current study, investigating isolated E2 effects in intact women has the drawback that it is currently unknown how these findings translate to phases of the menstrual cycle when fluctuations in E2 and P4 co-occur and to findings with ovariectomized animals. Nonetheless, similar neural activity has been observed in intact and ovariectomized rats treated with physiological E2 doses⁹. Third, animal studies sometimes involve higher supraphysiological E2 doses which one would not give to human participants because of potential health risks (e.g. thrombosis). As such, we cannot make any statements on the relationship between salivary E2 levels and hippocampal activation for such high supraphysiological doses. Finally, the BOLD effect approximately reflects the net amount of local glutamatergic synaptic activity, which can be the sum of distinct, even opposing, cellular processes affected by E2 (e.g. E2 modulates inhibitory interneurons).”

Comment 16: The discussion is disappointing and should be placed within the context of non-human findings. It is unclear which questions exactly the results of the current study settle.

Response: We agree and substantially rewrote not only the discussion but also the introduction. We would like to refer the reviewer to the revised manuscript rather than copying the revised introduction and discussion in full length.

Response to Reviewer #4

Rationale

The authors sufficiently describe in their introduction the rationale of the study. The overall aim of the study is to examine the effect of estrogen on cognitive function. Although this is not a new research question, there are still unresolved question, which the authors aim to address by manipulating the level of estrogen in a controlled manner. However, the main underlying hypothesis is that estrogen is the main driving factor and, hence, that the follicular phase is the phase of the menstrual cycle is the mostly deviating cycle phase. While there are many studies supporting this view, there are also studies demonstrating that the luteal phase is deviating as well, i.e. the phase where estrogen and progesterone are high and that this interaction might influence cognitive functioning. The authors briefly mention this interaction in line 6 on page 4 but could have discussed that in more detail (see also comment on progesterone levels in Supp.Figure 1).

Response: We agree with the Reviewer that the interaction between progesterone and E2 is crucial to consider when interpreting results from menstrual cycle studies. For space reason we restricted the discussion of interaction of estrogen and progesterone to the following sentences to the introduction and discussion:

Introduction

“For a variety of reasons, these studies can only be compared, to a limited degree, to the placebo-controlled, randomized E2 treatment regimens in young, healthy animals. Across the menstrual cycle, for instance, neuroactive progesterone also fluctuates, potentially and unpredictably interacting with E2 ²⁹.”

Discussion

“However, our current finding, that a well-controlled increase across a wide range of E2 levels does not result in changes in amygdala activity, suggests that these previous findings of changes in amygdala activity should perhaps be attributed to progesterone, which also fluctuates across the menstrual cycle. Indeed, amygdala reactivity to emotional stimuli is reliably modulated by progesterone, and its metabolite allopregnanolone has anxiolytic effects in rats via its action on the central nucleus of the amygdala ^{68,69}.”

Comment 1: Have the authors considered using differences scores in E2 levels rather than total E2 concentrations?

Response: We apologize for our ambiguous description of the analyses. Indeed, all analyses were conducted on difference scores (i.e. E2 on Day 2- E2 on Day 1) salivary E2 levels – exactly as the reviewer suggests.

We modified the respective sentence as response to the reviewer’s comment:

Methods

“Linear and inverted U-shaped responses to E2 increases from Day 1 to Day 2 (E2_{Day2-Day1}) were tested using a hierarchical robust regression approach run in MATLAB 2014b (Mathworks, Inc, Natick, MA, USA; fitlm function, robust fitting option). This approach is robust to potential outliers and violations of the normality

assumption. Two regression models were implemented for every variable of interest: the first including a linear term only ($y = a + b * E2_{\text{Day2-Day1}}$), the second including the linear plus a quadratic term ($y = a + b_1 * E2_{\text{Day2-Day1}} + b_2 * E2_{\text{Day2-Day1}}^2$).

Results

“Further statistical analyses were conducted using Day 1 to Day 2 increase in salivary E2 as a predictor (E2 on Day 2 – E2 on Day 1).”

Comment 2: Was the study controlled for stereotype effects, i.e. were equally often male as female research leaders present?

Response: Only female research leaders conducted the testing. We added this information to the methods section.

Comment 3: Is it recorded whether participants have been pregnant before or have been using contraceptives in the last couple of month before they participated?

Response: We are sorry for missing this important point. We only included women who have not been pregnant or used oral contraceptives for at least 6 months. We added this information to the methods sections. We also recorded life-time parity (i.e. pregnancy for more than 8 weeks), which has now been added to Supplementary Table 1:

Supplementary Table 1. Descriptive and inferential statistics of sample characteristics.

	experimental group										statistical analyses	
	0 mg		2 mg		4 mg		6 mg		12 mg		F	p
age (years)	25.57 ± 3.41	26.38 ± 3.85	26.50 ± 4.03	25.48 ± 4.15	25.83 ± 3.71						.34	.848
weight (kg)	61.47 ± 6.58	62.90 ± 8.62	64.85 ± 10.72	60.76 ± 8.82	66.17 ± 10.34						1.69	.157
bmi	21.83 ± 2.33	22.17 ± 2.14	23.48 ± 3.68	21.75 ± 2.93	22.71 ± 3.35						1.39	.242
											X ²	p
education (n cases)											14.6	.558
~10 y	0	0	2	0	1							
~11 y	1	0	0	0	0							
~12 y	2	1	0	1	1							
~13-16.5 y	18	13	8	16	15							
> 16.5 y	9	7	10	8	12							
nulliparity	28	19	19	23	28						.94	.919

All analyses are based on N=125. BMI = body-mass-index. Women were considered as ‘nulliparous’ when they have never been pregnant for longer than 8 weeks.

Comment 4: Page 5, line 20: Can the authors comment on why serum levels were assessed only in a subsample? Just for quality check of the saliva samples?

Response: We apologize for this lack of information, also mentioned by Reviewer 3 and 5.

In general, salivary E2 are considered to be in an equilibrium with the free, unbound E2 in blood and is hence a good marker for the bio-active E2 (Fiers et al., 2017). Serum levels were assessed only to make our data comparable to studies that assessed only total serum E2. For instance, E2 reference values (e.g. for pregnancy) are typically better available for serum than for saliva. However, not all women were willing to contribute blood, respectively it was not possible to draw blood from all participants after three attempts. The distribution of available serum levels did not significantly differ between groups [$\chi^2(4, N = 125)=5.55, p=.235$]. We added the number of available serum levels and inferential statistics to Supplementary Table 12 and the reason for missing serum levels to the methods section.

Supplementary Table 12. Number of cases included in each analysis.

	n per experimental group					N	statistical analyses of missing values	
	0	2	4	6	12	Σ	χ^2	p
salivary hormone levels	30	21	20	25	29	125	-	-
serum E2	23	19	12	17	21	92	5.55	.235
behavioral analyses	30	21	20	25	27	123	6.73	.151
MRI								
functional imaging: negative vs. neutral, hits vs. misses, EEM contrast	30	20	20	25	26	121	7.42	.115
functional imaging: remember vs. know	29	20	20	24	25	118	5.31	.257
arterial spin labeling	28	19	20	25	25	117	6.07	.194

Fiers, T., Dielen, C., Somers, S., Kaufman, J.-M. & Gerris, J. Salivary estradiol as a surrogate marker for serum estradiol in assisted reproduction treatment. Clin. Biochem. 50, 145–149 (2017).

Comment 5: Page 6, Figure 1, Page 21, line 3/4 & page 23, line 3/4: Although displayed and mentioned in figure 1, the authors may want to emphasize more on the temporal structure of the three days with respect to sampling of hormonal concentration, administration of the drug and time of the day for the fMRI examination, as these are essential parameter for a study like this one.

Response: We agree with the Reviewer and added a more detailed description of the temporal structure to Figure 1 and the Supplementary Material Section 2.1. The saliva samples (3 per day, 20-30 minutes apart) were drawn on Day 1 between ~6 pm and ~7 pm (before initial drug administration), on Day 2 between ~4.30 pm and ~6 pm and on Day 3 between ~6 pm and ~7.30 pm. Blood was drawn at ~6 pm on Day 1 and Day 2. Drug intake on Day 2 was on average around 7 pm, on Day 1 around 10.30 am. Scanning took place on average around 6 pm on Day 2.

Figure 1. Study design. Volunteers participated in the study on three consecutive days. In the evening of Day 1, baseline hormone concentrations and mood were assessed, followed by the first dose of either 17-beta-estradiol (E2; 2, 4, 6 or 12 mg) or placebo, depending on their assigned group (double-blind). On the morning of Day 2, participants took the second dose on their own. In the afternoon of Day 2, when E2 levels were expected to peak, participants encoded emotional pictures inside the scanner (upper red box). In the evening of Day 3, participants performed a recognition test (lower red box) and rated the encoded pictures for arousal. See Supplementary Material Section 2.1 for more details on the timeline of the study.

Comment 6: Page 25, line 20: Have the authors included realignment parameters as well? Have they used cut-offs for movements?

Response: We apologize for this lack in clarity also mentioned by Reviewer #5. We used the ‘realign & unwarp’ algorithm as implemented in SPM12, which corrects the data for susceptibility-by-movement artifacts which are responsible for most of the movement related variance via estimation and application of a derivative field (i.e. estimation of how deformations change with movement, based on ‘the observed variance (after realignment) and known (estimated) movement’ (SPM12 Manual, p.33). We added this information to the methods section.

“Next, to correct for susceptibility-by-movement artifacts, all functional images were realigned and unwarped (as implemented in SPM12).”

Comment 7: Page 26: Was an additional cluster threshold applied?

Response: No, we did not apply an additional cluster threshold but used peak based statistics, i.e. report the statistical peaks when they survived the correction for multiple comparisons. However, cluster extents are now added to Supplementary Table 8,9,10 and 11.

Comment 8: Results/Discussion

- Supplementary Figure1: Why has panel A only two bars per group?

Response: We apologize for making this point not sufficiently clear. For practical reasons, blood was drawn only on the two first testing days. We added this information to the figure caption.

Comment 9: Further, only the placebo group showed a (significant?) drop in progesterone concentration from Day1 to Day2. Could the authors please confirm that the administration of estrogen didn't interact with progesterone decrease? They only comment (page 8, line 7) that they didn't find a linear / quadratic relationship, but that doesn't have to occur - it might be more the absence of the drop of progesterone concentration during the menstrual phase.

Response: We agree that visual inspection of the figure suggests not only in the placebo group but also in the 4 and 6 mg groups a drop in progesterone levels. We therefore computed an additional univariate ANOVA with the between-subject factor 'group' and Day 1/Day 2 change scores of progesterone as the dependent variable. Results speak against significant differences in progesterone changes between groups [$F(4,120)=.97, p=.426$]. We added this information to the caption of Supplementary Figure 1.

Reviewer's Conclusion

This is a well-designed and well-conducted study. However, the results are difficult to understand since most of the discovered effects are not directly reflected in behavioral measures and vice-versa. Consequently, this study contributes more to the heterogeneity of the current literature rather than giving a clear picture.

Further, the discussion is mostly related to the description of the results with only a few attempts to explain the results. In particular, given that the discovery of a u-shaped response appears to be important to the authors, a more extended explanation why they found a u-shaped response could have been expected. They only mention (page 3, line 13) that it is a matter of debate but this debate should have been discussed in light of the present results.

Response: We agree with the reviewer that the way the manuscript was originally written was insufficiently situated in the field of neuroendocrine research and gave the impression that the study contributed to the heterogeneity of findings rather than clearing up issues. Therefore, we substantially rewrote the introduction and discussion. We have grounded our hypotheses much more explicitly. We now discuss the discrepancy between neural and behavioral results more in depth. We also offer a much more thorough discussion on the cellular mechanisms that have been proposed to underlie the inverted U-shape, why different brain regions might show distinct response curves, and how responses vary depending on E2 levels. We offer a more integrated account of the different E2-response curves, whether at physiological or supraphysiological E2 levels, that has been reported in

animal studies and our own study.

We believe that our data contribute to the understanding of the effects of E2 on memory-related processes in the human brain. We show that memory-dependent hippocampal activity in humans responds in a monotonically positive linear fashion to E2 dose until supraphysiological E2 levels are reached. At these levels, our results importantly show an inflection in the dose-response curve, resulting in an inverted U-shape. This result implies that increased E2 levels occurring during the menstrual cycle are beneficial to hippocampal activity, and only become disadvantageous at pregnancy levels. In a whole-brain analysis we see E2 affecting memory-related activity only in hippocampal areas.

We also found that E2 administration did not enhance hippocampus-dependent memory performance in a highly estrogen-sensitive sample (i.e. young, naturally cycling women). Our tightly controlled design and the wide range of induced E2 levels suggest that the lack of relationship found between E2 levels and memory performance very likely reflects a true null effect rather than confounds or noise introduced by the experimental setup. To illustrate how unlikely it is that there is any effect of E2 on recognition memory, we added a figure showing scatter plots for brain activity related to hippocampus-dependent memory (recollection) and hippocampus-independent memory (familiarity; please see below). We therefore suggest that a 24-hour period was long enough for an increase in E2 levels to increase synapse density and the associated glutamate release, as reflected by the fMRI signal, but too short for the new synapses to be functionally effective. This interpretation would also imply that the pre-ovulatory E2 peak in the human menstrual cycle might be too short – at least in some women, as there is large variability in cycles – to result in mnemonic changes, which could explain some of the inconsistencies across studies (Protopopescu et al., 2008; Resnick et al., 1998; Fehring et al., 2006).

We believe that the revised manuscript gives a clearer picture and motivates hypotheses that can be tested in future studies. For space reasons, we were hesitant to copy the revised introduction and discussion here in full length but would like to refer the reviewer to the revised manuscript in order to judge whether our modifications were adequate.

Fehring, R. J., Schneider, M. & Raviele, K. Variability in the Phases of the Menstrual Cycle. *Journal of Obstetric, Gynecologic, & Neonatal Nursing* 35, 376–384 (2006).

Protopopescu, X. et al. Hippocampal structural changes across the menstrual cycle. *Hippocampus* 18, 985–988 (2008).

Resnick, A. Neuropsychological performance across the menstrual cycle in women with and without premenstrual dysphoric disorder. *Psychiatry Research* 77, 147–158 (1998).

Response to Reviewer #5

Comment 1: How was the emotional valence of the 180 lure items matched against the target 180 items? Were targets and lures counterbalanced across participants and group? What was the number/portions of emotional and neutral items in the target and lure sets of items?

Response: We agree that this information was missing. Targets and lures were counterbalanced across subjects. In particular, we created 90 negative and 90 neutral picture pairs where both items of each pair were matched for arousal based on the ratings from a previous study. For each participant the two pictures of each pair were pseudo-randomly assigned to be a target or a lure. Therefore, arousal of the target and lures did not vary as a function of group [$F(4,118) = .89, p = .475$] nor did it show a valence by group [$F(4,118) = .89, p = .475$], target/lure by group effect [$F(4,118) = 1.00, p = .411$] or a valence by target/lure by group interaction [$F(4,118) = .92, p = .456$].

We now provide this more detailed description how items were pseudorandomly assigned to be targets and lures for each participant in the methods section:

“Based on arousal ratings from a previous study, we created 90 negative and 90 neutral picture pairs matched for level of arousal. The pictures of each pair were pseudo-randomly assigned as a target or a lure for each volunteer.”

Comment 2: It is not possible to evaluate whether the effects of encoding can be assessed discretely from novelty based effects – i.e., in the current description, it is unclear whether each of the 180 target pictorial stimuli were presented once across the 5 blocks of trials at encoding, because neither the number of trials per encoding block and/or the number of times each stimulus presented are stated.

Response: We agree that this information should be added. All 180 target stimuli were presented once during encoding, 36 pictures were presented within each block. We added the information to the description of the task.

“All volunteers encoded 180 of the 360 pictures stimuli equally split over five runs inside the MR scanner (i.e. 36 pictures per block). All stimuli were shown only once during encoding.”

Comment 3: The authors state, “In this task, participants indicated every second the direction of a randomly left or right pointing arrow presented in the middle of the screen.”- this needs to be re-written, since it is not possible to decipher the task from this description and evaluate whether this was an appropriate baseline task. How were the presumed differences in the number and timing of manual responses between the baseline and encoding task modelled in the GLM analyses?

Response: We agree that the description of the task should have been better described and calling it a ‘baseline task’ was misleading. Although this task was originally designed as a baseline task (Starck and Squire, 2001), we used it for different purposes. Stark and Squire (2001) designed this task as an implicit baseline in fMRI studies as an alternative to the

traditional passive viewing of a fixation cross. Using this task has been shown to be more appropriate for contrasts describing general encoding success, as it elicited less hippocampal 'baseline' activity than passive viewing of a fixation cross. In our study, we did not contrast activity during encoding to the implicit baseline but between our conditions. That is, we contrasted activity during successful versus unsuccessful encoding and negative versus neutral picture processing. Our purpose for using this 'distraction task' (what we had originally called 'baseline task') was to prevent rehearsal during the inter-stimulus interval to avoid emotional carry-over effects from negative to neutral items and to preserve participant's attention throughout the task by keeping them active.

In response to the reviewer's comment we renamed the task in 'distraction task', describe it more clearly and explain why it was used in the revised methods section:

"(...) Then participants performed a distraction task⁹⁰ to prevent rehearsal and emotional carry-over effects by prolonging ISIs while still maintaining participant's attention throughout the task. In this task, an arrow pointing either to the left or the right was presented for 800 ms in the middle of the screen. Volunteers were asked to indicate the direction of the arrow as fast as possible by a button press. After an inter-stimulus interval of 200 ms, the next arrow was presented."

Stark, C. E. L. & Squire, L. R. When zero is not zero: The problem of ambiguous baseline conditions in fMRI. *Proceedings of the National Academy of Sciences of the United States of America* 98, 12760–12766 (2001).

Comment 4: The event structure of the recognition test cannot be inferred from the current description

Response 4: We apologize for the unclear description of the recognition test and added more details to the methods section as well as a schematic overview over the event structure of the emotional memory task to Figure 1.

Figure 1. Study design. Volunteers participated in the study on three consecutive days. In the evening of Day 1, baseline hormone concentrations and mood were assessed, followed by the first dose of either 17-beta-estradiol (E2; 2, 4, 6 or 12 mg) or placebo, depending on their assigned group (double-blind). On the morning of Day 2, participants took the second dose on their own. In the afternoon of Day 2, when E2 levels were expected to peak, participants encoded emotional pictures inside the scanner (upper red box). In the evening of Day 3, participants performed a recognition test (lower red box) and rated the encoded pictures for arousal. See Supplementary Material Section 2.1 for more details on the timeline of the study.

Comment 4.1: as written, it seems that a (second-order) confidence estimate was acquired, possibly conflating first-order and second-order estimates. If so, why?, “6 participants were required to indicate on a 6-point confidence scale ranging from 'very sure new' to 'very sure old' whether the image was presented during encoding and how sure they were about this decision.” Did the recognition test, in fact, ask participants' confidence separately from their type-1 ratings? It is unclear why the Elfman et al.. (2014) paper is cited on Line 17, Page 5 in support of the design of the recognition test?;

Response 4.1: The Reviewer raises a point that we considered at length before proceeding with our design. Participants first had to judge whether they had seen the picture before and how confident they were in their decision on a 6-point confidence scale ranging from 'very sure old' to 'very sure new'. This was followed by a remember/know judgment only for the items judged as 'old'.

We used confidence ratings to estimate recollection-based contribution to behavioral

memory performance (Koen et al., 2016). For fMRI analyses, we back-sorted each encoding trial with subsequent remember/know judgments to identify activity related specifically to hippocampus-dependent memory encoding. After much deliberation, we decided against an additional old/new judgment before the confidence ratings because we were concerned that three ratings following each item would have been confusing for the participants and negatively impact the validity of their responses. We did not consider the use of six-point confidence ratings as overly problematic, as it is commonly used in studies from different groups adopting the dual process model (e.g. Yonelinas et al., 2002; Vann et al., 2009). In a previous study, we observed menstrual-cycle related variations in recollection by using the same approach, i.e. a 6-point confidence scale (Bayer et al., 2014). Although we are not aware of a study directly addressing a conflation between Type-1 and Type-2 responses caused by 6-point confidence recognition memory judgments, we would be, of course, happy to discuss the impact of such a bias in the revised version of the manuscript.

We also apologize for the misplaced citation of the Elfman Paper. We have removed the citation.

Bayer, J., Schultz, H., Gamer, M. & Sommer, T. Menstrual-cycle dependent fluctuations in ovarian hormones affect emotional memory. *Neurobiology of Learning and Memory* 110, 55–63 (2014).

Koen, J. D., Barrett, F. S., Harlow, I. M. & Yonelinas, A. P. The ROC Toolbox: A toolbox for analyzing receiver-operating characteristics derived from confidence ratings. *Behav Res* 1–8 (2016). doi:10.3758/s13428-016-0796-z

Yonelinas, A. P. et al. Effects of extensive temporal lobe damage or mild hypoxia on recollection and familiarity. *Nature neuroscience* 5, 1236–1241 (2002).

Vann, S. D. et al. Impaired recollection but spared familiarity in patients with extended hippocampal system damage revealed by 3 convergent methods. *PNAS* 106, 5442–5447 (2009).

Comment 4.2: how long was each picture presented on each trial of the recognition test? What were the verbatim instructions given to participants for each event on the recognition test? What were the response options on the recognition test remember/know/new? State the instructions associated with the presumed remember/know/new response options?

Response: During recognition, each picture was presented throughout the whole decision period (i.e. during confidence as well as remember/know rating).

Before the recognition test, participants were given detailed instructions on the confidence scale and the remember/know rating (see below). The confidence scale consisted of 6 squares, which were labeled as ‘absolutely sure old’, ‘relatively sure old’ and so on (see Figure 1). Participants performed the remember/know judgements only for items judged as ‘old’ in the previous confidence rating. Consequently, there was no ‘new’ option during the remember/know rating. The remember/know rating consisted of two squares, labeled as ‘remember’ and ‘know’. The labels were presented below the picture until a response was given. We added this information to the results section. We added the instructions to the Supplementary Material Section 2.4.

The following instructions were given prior to the start of the recognition test:

‘Now we will present the pictures that were presented yesterday (‘old pictures’) randomly mixed with pictures you haven’t seen before (‘new pictures’). For each picture, you will be asked to judge whether the picture is ‘old’ or ‘new’. For this decision, you will have to choose between 6 options by selecting the corresponding

box: 'absolutely sure old', 'relatively sure old', 'rather old', 'rather new', 'relatively sure new', 'absolutely sure new'. These options will be presented under the picture for every decision. Please respond as correct and fast as possible. However, there is no time limit. Please try to use the whole scale.

*For each picture judged as 'old', you will be then asked whether you **'remember' details from the learning situation** or whether you just **'know' that the picture was presented yesterday**. Please choose **'remember'** if you remember any detail from the learning situation such as what you have thought or felt when viewing the picture, at what time the picture was presented (before or after another specific picture) and so on. Please choose **'know'** if you only remember that we presented the picture yesterday but not any further details from the learning situation.'*

Comment 5: State the method of correction applied to the behavioral statistics, “Statistical results were considered significant at a threshold of $p < .05$ corrected for multiple comparisons if adequate.” – presumably, the alpha criterion was modified according to the method of correction for multiple comparisons so that it was fixed or adjusted, such as the Holm-Bonferroni method.

Response: We apologize for this lack in clarity. The reviewer is right, the alpha-criterion was modified according to the Bonferroni method (i.e. alpha/number of comparisons). We added this information to the methods section and table legends.

Comment 6: As currently written, the rationale and method for obtaining the estimates of recollection and familiarity from ROC plots and the remember/know responses is difficult to understand without prior knowledge. This is undesirable for a widely-read journal such as Nature Communications. Also, add a new figure to depict the area under the ROC curve associated with type-1 and type-2 performance.

Response: We agree that more detailed rationale and methods should be provided for parameter estimation from the ROC curves and the remember/know responses and added a paragraph to the methods section. Moreover, we added values for the area under the curve to Supplementary Table 3 and ROC curves to the Supplementary Material (Supplementary Figure 2).

“These confidence ratings were used to estimate the contribution of recollection and familiarity to recognition memory. The Dual Process model of recognition memory was fitted to the receiver-operating characteristic (ROC) curves, which were derived from the hit and false alarm rates across confidence levels (Supplementary Figure 2) using maximum likelihood estimation⁴⁵. This was done separately for neutral and negative items and performed in Matlab R2014b (Mathworks, Inc, Natick, MA, USA). For exploratory reasons, confidence ratings were additionally used to estimate meta d' -prime and meta criterion⁹¹ (see Supplementary Table 3). The remember/know procedure (see Supplementary Material Section 2.3 for details) was used to back sort each encoding trial as to whether they were subsequently retrieved with contextual details (hippocampus-dependent ‘recollection’) or only with a sense of familiarity (hippocampus-independent ‘know’). Estimates for recollection and familiarity were also based on remember/know responses using the formulas provided⁹² for completeness.”

Supplementary Figure 1. Receiver-operating characteristic curves ($N=123$) based on confidence ratings in the recognition memory task for neutral (A.) and negative pictures (B.).

Comment 7: Are the d' values, estimates of recollection and familiarity, and criterion reported in Supplementary Table 1 significantly different as a function of experimental group? Report the relevant inferential statistics.

Response: Neither robust regression analyses nor univariate ANOVA's with the between-subject factor group show significant E2 effects on d prime, parameter estimates or response criterion. We added the descriptive and inferential statistics for robust regression analyses using these dependent variables to Supplementary Table 3 (formerly Supplementary Table 1; for the sake of brevity not inserted here). Results of the ANOVA are inserted below.

	ANOVA	
	F	p _{cor}
neutral pictures		
hit rate	.58	>.999
false alarm rate	.37	>.999
recollection	1.83	.254
familiarity	.35	>.999
dprime	.76	>.999
criterion	.40	>.999
area under the curve	.56	>.999
meta dprime	.16	>.999
meta criterion	.41	>.999
negative pictures		

hit rate	.66	>.999
false alarm rate	.50	>.999
recollection	.41	>.999
familiarity	.56	>.999
dprime	.36	>.999
criterion	.85	.990
area under the curve	.44	>.999
meta dprime	.51	>.999
meta criterion	.82	>.999

Comment 8: Was there a significant difference in mean confidence associated with all ‘old’ and all ‘new’ retrieval cues, as a function of group and stimulus type (neutral, negative); i.e., report whether the confidence ratings were diagnostic of recognition memory.

Response: We agree that this information should be reported. Paired *t*-tests showed that confidence ratings (coding: absolutely sure old/new = 3, relatively sure old/new = 2, unsure old/new = 1) were significantly higher for correct than for incorrect responses [targets: $t(122)=15.57$, $p<.001$; lures: $t(122)= 6.83$, $p<.001$] suggesting that confidence ratings were diagnostic of recognition memory. We added these analyses to the results section (2.1) . Moreover, we added descriptive and inferential statistics for mean confidence split by targets vs. lures and stimulus valence to Supplementary Table 5 (for the sake of brevity not inserted here), showing that confidence was not related to salivary E2 increase [all p 's >.2]. A table containing the results of the ANOVA's is inserted below:

		ANOVA	
		F	p_{corr}
neutral pictures			
	confidence targets, 'old' responses	.25	>.999
	confidence targets, 'new' responses	1.82	.258
	confidence lures, 'old' responses	.76	>.999
	confidence lures, 'new' responses	1.74	.292
negative pictures			
	confidence targets, 'old' responses	1.04	.784
	confidence targets, 'new' responses	1.01	.816
	confidence lures, 'old' responses	.89	.814
	confidence lures, 'new' responses	.73	>.999

Comment 9: It might also be worthwhile reporting analyses that incorporate participants' biases and first-order response such as meta-*d'* (Maniscalco & Lau 2012) or type-II ROC curve (Flemming et al. 2010).

Response: This is an excellent idea. We added estimates of meta-*d'* and meta-criterion (Maniscalco & Lau, 2012) and the respective inferential statistics of robust regression analyses to Supplementary Table 3 (for the sake of brevity not inserted here). A table containing the results of the ANOVA's is inserted below our response to comment 7.

Maniscalco, B. & Lau, H. A signal detection theoretic approach for estimating metacognitive sensitivity from confidence ratings. *Consciousness and Cognition* 21, 422–430 (2012).

Comment 10: Supplementary Table 2 – update to report beta and r-squared values associated with each of the regression analyses.

Response 10: We added standardized beta estimates to Supplementary Table 10 and the results section of the main manuscript. However, we would like to avoid reporting any absolute effect size measures (such as r or R^2) for the robust regression analyses, as these measures are not corrected for multiple comparisons and would overestimate effect sizes. Moreover, R^2 is not an appropriate measure for the fit of non-linear regression models nor optimal for robust regression (Willett & Singer, 1988; Spiess & Neumeyer, 2010). We therefore decided to report delta BIC instead of R^2 as a relative measure of model fit. We hope we have addressed the reviewer’s concerns in doing so.

Supplementary Table 10. Cluster extents and results of robust regression analyses using the increase in salivary Day 1 to Day 2 increase in 17-beta-estradiol (E2) and contrast estimates from the respective peak voxels.

	SPM analyses	Robust Regression			
relationships with salivary Day 1 to Day 2 E2 increases	k at $p_{unc} < .005$	t	p_{unc}	Δ BIC	β_{std}
linear					
remember > know contrast in the hippocampus (33, -30, -12)	57	3.57	<.001	-4.55	.315
negative > neutral contrast in the precuneus (6, -60, 15)	345	-4.18	<.001	-6.72	-.328
negative > neutral contrast in the brainstem (6, -30, -12)	118	-4.01	<.001	-5.79	-.311
PPI for negative > neutral contrast for precuneus-brainstem connectivity (6, -21, -9)	35	-3.01	.003	-2.85	-.256
quadratic					
remember > know contrast in the hippocampus (15, -33, -12)	29	4.36	<.001	8.50	.834
EEM contrast in the insula (42, -6, 9)	433	2.49	.013	16.93	.505

Δ BIC = Difference in Bayesian Information Criterion (BIC) between linear and quadratic models. EEM = emotional enhancement of memory (hit > miss x negative > neutral). All analyses are based on $N=121$, except of analyses using the remember>know contrast which are based on $N=118$.

Willett, J. B. & Singer, J. D. Another Cautionary Note about R^2 : Its Use in Weighted Least-Squares Regression Analysis. *The American Statistician* 42, 236–238 (1988).

Spiess, A.-N. & Neumeyer, N. An evaluation of R^2 as an inadequate measure for nonlinear models in pharmacological and biochemical research: a Monte Carlo approach. *BMC Pharmacol* 10, 6 (2010).

Comment 11: Similarly, there is information missing from the neuroimaging methods that makes it hard to adjudicate whether some of the choices in the analyses and reporting of

the results are appropriate:

Comment 11.1: Specifically state the voxel spatial resolution of the EPI and T1-weighted sequences in mm³ – it is currently not possible to evaluate the choices made in terms of spatial smoothing and critically the merit of consistent reference to the capacity to evaluate, “hippocampal subregional activity” and later qualification in the discussion section about limits on these types of analyses due to the acquired spatial resolution.

Response 11.1: We used a spatial resolution of 3 mm³ for the EPI- and of 1 mm³ for the T1 weighted sequences. We agree that the term ‘hippocampal subregional activity’ could be misleading as the spatial resolution of our fMRI-scans did not allow to relate activity patterns to specific hippocampal subfields. We now avoid the term ‘subregion’ when discussing our results and instead speak of ‘different/distinct hippocampal clusters’, e.g.:

“Although at first sight it seems surprising that we observed both types of response curves in distinct hippocampal clusters, previous studies have also described dissociable dose-response functions for different hippocampal subfields (CA1 vs. CA3) and depending on whether hippocampal subregions or whole hippocampus is considered (whole vs. rostral hippocampus) ^{5,9}.”

“However, the spatial resolution of our fMRI scans is too coarse to relate the different dose-response curves to specific anatomically defined hippocampal subfields and only permits a functional dissociation.”

“In conclusion, under tight experimental control in young, healthy women, we showed that the pharmacological increase of a wide range of physiological and supraphysiological E2 levels modulates memory-related activity differentially in distinct hippocampal clusters.”

Comment 11.2: State the approach used to assess and exclude the impact of movement on the fMRI data.

Response 11.2: We agree that this information was missing. We used the ‘realign & unwarp’ algorithm as implemented in SPM12, which corrects the data for susceptibility-by-movement artifacts which are responsible for most of the movement related variance via estimation and application of a derivative field (i.e. estimation of how deformations change with movement, based on “the observed variance (after realignment) and known (estimated) movement” (SPM12 Manual, p.33). The advantage of using the ‘realign and unwarp’ algorithm compared to including the movement parameters in the first level model is that the former one corrects the MRI images only for artificial, movement related signal changes whereas the latter approach removes also true, activity related signal variance. We added this information to the methods section.

“Next, to correct for susceptibility-by-movement artifacts, all functional images were realigned and unwarped (as implemented in SPM12).“

Comment 11.3: State the MNI coordinates of the precuneus seed region in the PPI analysis. What was the size of the sphere in this analysis?

Separately, what were the sizes of the spheres used to derive beta values extracted from

peaks inside clusters surviving multiple comparison correction and used for the hierarchical robust regressions?

Response: The MNI coordinates of the precuneus seed region were (6,-60,15). For the PPI, we did not use a sphere but the whole cluster showing a significant relationship with salivary E2 increase for brain activity associated to emotional processing thresholded at $p_{unc} < .001$. We added this information to the results section.

“(...) we created mask images from the clusters in the precuneus [peak: (6, -60, 15)] and the brainstem [peak: (6, -30, -12)] that showed a significant negative relationship with Day 1 to Day 2 salivary E2 increase thresholded at $p_{unc} < .001$.”

For the robust regressions, we extracted beta values from the respective peaks. As expected, given the voxel size of 3 mm³ and a smoothing kernel of FWHM 8 mm as well as consistent with a study showing that the extraction of time series from a peak voxel yields similar results to the application of a 5 mm sphere centered around the peak (Gonçalves & Hall, 2003), our robust regression analyses show comparable results when a 5 mm sphere was used:

	robust regression analyses				
	(beta values extracted were from 5 mm spheres centered around the peak voxels)				
relationships with salivary Day 1/Day 2 E2 increase	t	df	p	Δ BIC	stand. β
linear					
remember > know contrast in the hippocampus [x=33, y=-30, z=-12]	3.19	116	.002	-4.68	.28
negative > neutral contrast in the precuneus [x=6, y=-60, z=15]	-4.03	119	<.001	-6.16	-.31
negative > neutral contrast in the brainstem [x=6, y=-30, z=-12]	-3.92	119	<.001	-5.54	-.32
PPI for negative > neutral contrast for precuneus-brainstem connectivity [x=6, y=-21, z=-9]	-3.16	119	.002	-2.53	-.27
quadratic					
remember > know contrast in the hippocampus [x=-15, y=-33, z=-12]	3.78	116	<.001	8.97	.78
EEM contrast in the insula [x=42, y=-8, z=9]	2.63	119	.010	19.53	.53

Gonçalves, M. S. & Hall, D. A. Connectivity analysis with structural equation modelling: an example of the effects of voxel selection. *NeuroImage* 20, 1455–1467 (2003).

Comment 11.4: The authors should test their hypotheses against ROIs drawn from clusters that are not derived from the peak inside the clusters surviving multiple comparisons – more recent approaches contend the latter is a form of double-dipping; i.e, the authors should evaluate whether the results hold when ROIs are based on coordinates defined independently from the primary task based contrast.

Response: Based on the reviewer’s comment we got the impression that we described our

procedure ambiguously because we would certainly agree that the scenario summarized by the reviewer would be a form of double dipping.

In order to identify areas where brain activity shows a linear or quadratic relationship with E2 increases, we conducted linear and quadratic regression analyses using the E2 increase as independent variable within SPM. These analyses were family-wise corrected for multiple comparisons for the whole scan volume and within ROIs based on the Gaussian Random Field Theory. The a priori ROIs were anatomical defined masks of the amygdala and the hippocampus created with the Anatomy Toolbox (Amunts et al., 2005; Tzourio-Mazoyer et al., 2002). Betas were extracted then from the peaks identified by this procedure only for visualization purposes. Additional robust regression analyses were conducted using these betas were conducted exclusively for confirmatory reasons, i.e. to ensure that the results were not biased by outliers. In other words, the robust regression analyses were not conducted to generate new information but to confirm previous results.

In response to the reviewer's comment and to avoid any misunderstandings we modified the results and methods sections as follows:

Results

“Confirming the sensitivity of the memory paradigm to detect subtle differences hippocampal activity, the remember>know contrast showed robust effects in the left and right hippocampus [left: (-30, -30, -18); Z=7.39, $p_{corr}<.001$; right: (24, -9, -18); Z=5.58, $p_{corr}<.001$; Supplementary Figure 5] and a wide network of brain areas (Supplementary Table 8). Independent linear and quadratic regression analyses on the whole brain were then performed to identify areas showing linear or inverted U-shaped relationships between E2 increase and recollection-related activity. These were performed in SPM using the remember>know whole brain contrast images as the dependent variable and the increases in E2 from Day 1 to Day 2 as the independent variable.”

Methods

“Event-related BOLD responses were analyzed in a general linear model as implemented in SPM using a mass univariate approach. Subject-level models included 6 separate regressors, convolved with the canonical hemodynamic response function, for the factors valence (neutral, negative) and memory (hit remember, hit know, miss). To investigate the effects of E2 on memory and affective processing, the following 4 contrasts, defined a priori, were calculated for each participant: 1. A remember/know contrast (remember > know) to specifically examine hippocampus-dependent memory; 2. General encoding success (subsequent hit > miss); 3. The effects of emotional processing (negative > neutral); 4. EEM (valence [negative > neutral] × general encoding success [hit > miss]).

*Each individual's contrast images were used for second-level group analyses with subject as a random factor. To test whether the planned contrasts were associated with activity in the areas of primary interest, i.e. the hippocampus and amygdala, and other wide network of brain areas, the main effects for remember/know, general encoding success, emotional processing, and EEM contrasts were estimated. To identify brain regions showing linear or quadratic effects with E2 levels, additional linear only ($y = a + b * E2_{Day2-Day1}$) and quadratic regression models ($y = a + b_1 * E2_{Day2-$*

$_{Day1} + b_2^* E2_{Day2-Day1^2}$) were implemented for each contrast of interest (remember/know, general encoding success, emotional processing and EEM) on the group level. As we were interested in the inverted U-shaped dose-response function, only the negative contrast of the quadratic term was estimated. Results of all analyses, i.e. the main effects as well as the models testing for E2 relationships, were considered significant at $p < .05$, family-wise error corrected for multiple comparisons on the entire scan volume and within predefined anatomical regions of interests (ROIs), namely the hippocampus and the amygdala. The hippocampus and amygdala masks were created using the Anatomy toolbox ^{97,98}.

To verify that resulting relationships are not biased by outliers or violations of the normality assumption, beta values were extracted from peaks in clusters surviving multiple comparison correction and used for the robust regression analyses explained above. As long as an appropriate multiple comparison correction is applied, calculating more elaborate statistical models, in particular in confirmatory analyses, using extracted beta values is statistically sound ⁹⁹⁻¹⁰¹.

Amunts, K. et al. Cytoarchitectonic mapping of the human amygdala, hippocampal region and entorhinal cortex: intersubject variability and probability maps. *Anat. Embryol.* 210, 343–352 (2005).

Tzourio-Mazoyer, N. et al. Automated anatomical labeling of activations in SPM using a macroscopic anatomical parcellation of the MNI MRI single-subject brain. *Neuroimage* 15, 273–289 (2002).

Comment 11.5: The authors state, “ASL data of six participants were missing because of technical failure and excessive head movement” – state which experimental group these participants were assigned to; it is not possible to evaluate the impact of these missing data without this information.

Response: We apologize for the missing information. Of the six missing ASL-datasets, 2 belonged to the 0 mg, 2 to the 2mg and 2 to the 12 mg group. We added a table (see below) with the number of subjects included in each analyses to the Supplementary Material (Supplementary Table 12).

Supplementary Table 12. Number of cases included in each analysis.

	n per experimental group					N Σ	statistical analyses of missing values	
	0	2	4	6	12		X ²	p
salivary hormone levels	30	21	20	25	29	125	-	-
serum E2	23	19	12	17	21	92	5.55	.235
behavioral analyses	30	21	20	25	27	123	6.73	.151
MRI								
functional imaging: negative vs. neutral, hits vs. misses, EEM contrast	30	20	20	25	26	121	7.42	.115
functional imaging: remember vs. know	29	20	20	24	25	118	5.31	.257
arterial spin labeling	28	19	20	25	25	117	6.07	.194

Comment 12: Report was the final n / group for each set of analyses throughout the results section. In Figure 4, why are fMRI results based on $n=118$? – partial clues in the methods do not reveal the whole picture, “3 further participants had to be excluded because of empty cells (i.e. no correct ‘know’ responses for neutral pictures, no ‘remember’ responses)” and later, “Functional MRI data of two participants were missing due to technical failure and excessive head movement.”

In Figures 5 and 6, why are the fMRI results based on $n=121$?

Response: We apologize for the missing information. $N=121$ participants were included in the second level fMRI analyses based on the negative > neutral, hits>misses, EEM first-level contrast images. For the second level analysis of the remember> know contrast, three further participants had to be removed because they had no correct ‘know’-responses for neutral pictures ($n=1$ from 6, respectively the 12 mg group) or no correct ‘remember’-responses ($n=1$ from 0 mg group). For these participants, remember>know contrasts would be not possible to calculate. We added a table with the number of participants included in each analyses to the Supplementary Material (see above) and added information with respect to the group affiliations to the methods section.

Comment 13: Figures 4-6 – the figures are thresholded at $p < 0.005$ – is this threshold for a p -value corrected for multiple comparisons? Either way, the thresholding is at variance with the $p > 0.05$ FWE corrected for the statistical analyses. A t-map showing the results based on $p > 0.05$ FWE corrected should be included in updates to figures 4 and 5.

Response: We agree with the reviewer on the importance of using appropriate, unbiased statistical thresholds. We use only 2 different statistical thresholds in the main manuscript. All results in the main text and tables and, importantly, for any conclusions made in the discussion, are based on a corrected statistical threshold of $p < .05$ FWE, unless otherwise clearly noted as non-significant trends (e.g. brainstem results). We also use an uncorrected threshold of $p < .005$ purely for displaying purposes in the figures. Only the exploratory analyses in the Supplementary Material are reported using an uncorrected threshold of $p < .001$ to convey a more comprehensive picture of our findings.

Our motivation for the more lenient display threshold was to show a more comprehensive view of the activations. This is a conventional strategy employed by many other fMRI studies. In our opinion, this display threshold represents a good compromise between sensitivity and specificity when visualizing the results in this study. Lieberman & Cunningham (2009) have even suggested, based on statistical modeling, that an adequate balance between Type I and Type II errors can be achieved at an uncorrected voxel-wise threshold of $p < .005$ cluster and an extent threshold of $k > 10$. All our figures already meet this threshold. If the reviewer is still concerned, we would suggest reporting cluster sizes at $p_{\text{corr}} < .05$ in the relevant tables or add figures using the desired threshold.

Lieberman, M. D. & Cunningham, W. A. Type I and Type II error concerns in fMRI research: re-balancing the scale. *Soc. Cogn. Affect. Neurosci.* 4, 423–428 (2009).

Comment 14: Provide information on how post-error trials and reaction time outliers were handled in the analyses. What proportion of trials were affected by errors?

Response: To prevent that reaction time outliers bias our analyses, we used the individual median of reaction times. Moreover, as the robust regression analyses down weights the impact of outliers, we did not exclude outliers for any measure.

In the encoding task (i.e. inside/outside decision) participants had a hit rate of $M=.77$ ($SD=.076$). There was neither a significant relationship with salivary E2 increase in a robust regression analyses [$t(121)=-0.37$, $p=.709$, $\beta_{std}=-.023$] nor a significant main effect of group in an univariate ANOVA [$F(4,118)=.059$, $p=.993$]. For the distractor task (i.e. arrow task) participants had a hit rate of $M=.96$ ($SD=.08$). Again, there was no significant relationship with salivary E2 increase in a robust regression analyses [$t(121)=-0.20$, $p=.840$, $\beta_{std}=-.004$]. We added robust regression analyses of accuracy in the encoding and the arrow task to the manuscript.

With respect to post error trials we calculated the rate of subsequent hits in the recognition task for trials in which the last arrow before the picture was responded correctly and for trials in which the last arrow before the picture was responded falsely (i.e. two arrows were ignored in this analyses). Only 102 participants could be included in this analyses, because 21 participants had no wrong responses for the last arrows before pictures. A paired t -test did not show significantly different hit rates during the recognition test for the two trial types [$t(101)=1.088$, $p=.279$].

Comment 15: In the fMRI analyses, what was the duration of the response window examined for each back-sorted remember>know trials at encoding? Likewise for the other two main sets of event-related fMRI analyses.

Response 15: Participants had in the scanner a time window of 2 seconds for the encoding task (i.e. the inside/outside decision). The remember/know responses on the next day outside of the scanner were self-paced so that we did not restrict our analyses to a specific time window.

Comment 16: Were trial-by-trial manual response latencies at encoding included as regressors of interest in the fMRI GLM analyses?

Response 16: In the original analyses, we did not include response latencies as regressors. However, adding a regressor with the onsets of manual responses during the encoding task to the first level model does not alter the significance level of effects in the hippocampus [linear: (33, -30, -12), $Z=3.69$, $p_{corr}=.034$; quadratic: (-15, -33, -12), $Z=4.10$, $p=.008$], the insula [(42, -6, 9), $Z=4.65$, $p=.050$], the precuneus [(6, -63, 15), $Z=4.94$, $p_{corr}=.014$] or the brainstem [(6, -27, -15), $Z=4.53$, $p_{corr}=.075$].

Comment 17: Several important additional analyses are needed in order to improve how the main claims have been tested and thus provide more convincing evidence to strengthen the conclusions. In particular, report the following additional analyses:

Comment 17.1: What is the link between salivary and sera E2 levels? Include an analysis

to test this link; the results are important for understanding the implications of basing the main results and claims on the increase in Day 1 to Day 2 absolute salivary E2 levels, especially given that sera E2 levels were subsampled in 90/123 participants.

State why the results of salivary levels of E2 from 2 of the 125 are missing.

Response: The Day 1 to Day 2 increase in serum and salivary E2 levels show a high correlation [$r(90)=.825$, $p<.001$] confirmed by linear robust regression analysis [$t(90)=28.61$, $p<.001$, $\beta_{std}= 1.04$]. We added this result to the results section. In general, salivary E2 are considered to be in an equilibrium with the free, unbound E2 in blood and in hence a good marker for the bio-active E2 (Fiers et al., 2017). Serum levels were assessed only to make our data also comparable to studies that assessed only total serum E2.

We initially thought it would be better to remove the two participants with missing behavioral data from all of our analyses. However, we agree with the reviewer that the hormone levels are valuable data. We therefore now report hormone levels for the whole sample ($N=125$) and revised all relevant statistical analyses and figures.

Fiers, T., Dielen, C., Somers, S., Kaufman, J.-M. & Gerris, J. Salivary estradiol as a surrogate marker for serum estradiol in assisted reproduction treatment. *Clin. Biochem.* 50, 145–149 (2017).

Comment 17.2: Given the wide variances in absolute salivary E2 levels within each group, report inter-quartile ranges for the absolute salivary E2 levels associated with each of the 5 groups. Also report the corresponding effect size for the D1-D2 absolute salivary E2 levels, as a function of group.

Response: We now report inter-quartile ranges for absolute salivary E2 levels of all testing days and the difference between Day 1 to Day 2 changes as a function of group in Supplementary Table 2. Moreover, we added the effect size d for paired t -tests comparing absolute Day 1 and Day 2 salivary E2 levels separately for each group.

Supplementary Table 2. Interquartile ranges and Cohen's d for increases in salivary 17-beta-estradiol (E2) from Day 1 to Day 2 (N=125).

experimental group	Day1 to Day2 change in salivary E2	
	interquartile ranges	d
0 mg	2.23	-0.477
2 mg	6.31	1.484
4 mg	6.58	2.673
6 mg	17.25	2.344
12 mg	16.33	3.136

Comment 18: The results should not be discussed in terms of trend level significance; e.g., Lines 1-4, Section 2.1, “With respect to factors extracted from the mood questionnaire, only a positive linear relationship of factor 1 (dimensions: alertness, strength, energy, clear-headedness) survived Bonferroni correction with trend-level significance [$t(121)=2.31$,

pcor=.092].” On Page 9, “The analysis of rCBF revealed a trend-level significant positive relationship with salivary E2 increase in the right amygdala [$x=16, y=-4, z=-20; Z=2.85, p_{corr}=.067$; Supplementary Figure 2].” Similarly, on Pages 10-11, “with trend-level significance in the right brainstem [$x=6, y=-30, z=-12, Z=4.60, p_{corr}=.056$].” If the authors wish to pursue this approach, then, for equivalence sake and as an example, one of the key clusters in the right posterior hippocampus and the significant positive linear association with increases in E2 levels [$x=33, y=-30, z=-12; Z=3.69, p_{corr}=.034$; Figure 4AB] should be discussed as trending towards non-significance.

Response: The Reviewer raises a point here over which we belabored before deciding on presenting the results that we have. In the end, we decided to also report the three non-significant trends based on the following rationale. This was a particularly involved study, both in terms of the size of our sample and the E2 regimen (i.e. scanning only in the menstruation phase), which required much organization and financial resources. We believe it will be unlikely that too many similar studies will be conducted of this scale. We therefore reasoned that we should also report the trends that did not survive our conservative statistical thresholds in order to avoid type II errors, to indicate possible avenues of investigation, and as a comparison for future studies that might observe similar results. For instance, the non-significant trend in the brainstem might direct animal researcher’s attention to this brain area and can also provide an a-priori ROI for future human fMRI studies. The non-significant trend in the amygdala with respect to perfusion could motivate future studies to use more powerful methods to detect such differences (for instance a longer ASL sequence including more images). Finally, the non-significant trend for a positive linear relationship between salivary E2 increase and the first PCA component (previously labeled as ‘Factor 1’) derived from the questionnaire could be tested in studies using methods designed to specifically assess ‘subjective vigilance’.

In response to the reviewer’s comment, we decided to place even greater emphasis on the non-significance of the results which did not survive our statistical thresholds and replaced the term ‘trend-level significant’ by ‘non-significant trends/tendencies’ throughout the manuscript.

If the reviewer is still concerned with our results being misinterpreted, we would remove the non-significant findings from the manuscript.

“Only the first component (dimensions: alertness, strength, energy, clear-headedness; see Supplementary Table 13), determined by a principal component analyses on data from the mood questionnaire, showed a non-significant trend after Bonferroni correction [$t(121)=2.31, p_{corr}=.092, \beta_{std}=.188$].”

“Analysis of rCBF revealed a non-significant trend towards a positive relationship with salivary E2 in the right amygdala [peak at (16, -4, -20) in Montreal Neurological Institute (MNI) space; $Z=2.85, p_{corr}=.067$; Supplementary Figure 3].”

*“Although not predicted, there was a negative linear relationship between E2 levels and differences in brain activity for negative compared to neutral pictures in the right precuneus [(6, -60, 15), $Z=4.70, p_{corr}=.037$], which survived family-wise error (FWE) correction for multiple comparisons for the whole scan volume. A similar negative linear trend in the right brainstem was observed but was non-significant, that is, it did not quite survive FWE correction [(6, -30, -12), $Z=4.60, p_{corr}=.056$] (**Fehler! Verweisquelle konnte nicht gefunden werden.**)”*

Comment 19.1: State the hit and false alarm rate on the recognition test and plot the ROCs

Response: We added the hit and false alarm rate and corresponding inferential statistics to Supplementary Table 3 (not inserted here for brevity sake). Moreover, ROC's curves were added to the Supplementary material (Supplementary Figure 2).

Comment 19.2: What were the median number and range of hit and correct rejection trials, as a function of group; and

Response: We added the respective data Supplementary Table 4 (not inserted here for brevity sake).

Comment 19.3: Report the mean/median response latencies associated with the presumed remember/know/new responses on the recognition test and evaluate whether these differed as a function of group.

Response: We added a table containing descriptive and inferential statistics of robust regressions analyses for response latencies during the encoding task (i.e. inside/outside) decision as a function of subsequent memory (remember, know, new), valence and group to the Supplementary Material (Supplementary Table 6; not inserted here for brevity sake). A table containing the results of the ANOVA's is inserted below. Response latencies did not vary as a function of group (all $p_{corr} > .188$).

ANOVA		
	F	p_{cor}
neutral target pictures		
remember	.40	>.999
know	1.59	.365
new	2.04	.188
negative target pictures		
remember	1.17	.652
know	1.34	.521
new	1.68	.319

Comment 20: What were the mean arousal ratings for neutral and negative stimuli, as a function of group. Were the differences significant between the groups?

Response: We added descriptive and inferential statistics of robust regressions for arousal ratings as a function of group to Supplementary Table 7 (not inserted here for brevity sake). A table containing group-based statistics is inserted below our response to the next comment. Consistent with the robust regression analyses, arousal ratings varied as a function of group for neutral [$F(1,118)=3.77$, $p_{corr}=.012$] but not negative pictures [$F(1,118)=2.17$, $p_{corr}=.152$].

Comment 21: Further, what are the mean arousal ratings for neutral and negative stimuli associated with the presumed, remember, know and new responses, as a function of group? These data have relevance for the decision to examine E2 effects on amygdala activity by collapsing across what I presume is retrieval success; cf. Page 10, Lines 16-17.

Response: We apologize for the missing information and added descriptive and inferential statistics of robust regression analyses for arousal ratings associated to subsequent responses in the recognition test as a function of group to Supplementary Table 7 (not inserted here for brevity sake). A table containing the results of the ANOVA's is inserted below. As expected, arousal ratings for neutral pictures split by retrieval success show a similar pattern of an increase in arousal rating with E2 dose.

Supplementary Table 7. Descriptive and inferential statistics of arousal ratings as a function of responses in the recognition memory task.

ratings	experimental group										ANOVA	
	0 mg		2 mg		4 mg		6 mg		12 mg		F	pcor
neutral pictures												
total	2.49	± 1.11	2.28	± 1.05	3.04	± 1.21	3.20	± 1.29	3.28	± 1.05	3.77	.012
remember responses	2.69	± 1.15	2.94	± 1.35	3.05	± 1.18	3.49	± 1.38	3.55	± 1.07	2.44	.305
know responses	2.51	± 1.13	2.31	± 1.13	2.98	± 1.25	3.20	± 1.39	3.24	± 1.09	2.95	.139
new responses	2.42	± 1.14	2.06	± 1.08	2.96	± 1.24	3.03	± 1.20	3.20	± 1.15	3.97	.028
negative pictures												
total	5.92	± 1.11	5.70	± 1.16	5.76	± 1.44	6.48	± .91	6.26	± .87	2.17	.152
remember responses	6.49	± 1.11	6.48	± 1.10	6.32	± 1.50	6.98	± .81	6.73	± .81	1.34	>.999
know responses	5.76	± 1.22	5.71	± 1.29	5.65	± 1.53	6.26	± .99	6.08	± .86	1.22	>.999
new responses	5.19	± 1.35	5.00	± 1.36	5.32	± 1.57	5.97	± 1.19	5.84	± 1.05	2.58	.246

All analyses are based on N=123, except of analyses on confidence ratings for negative target items judged as 'remember' which is based on N=118 because of missing responses within this category (missing data: n=1 from 0 mg, n=1 from 2 mg, n=1 from 6 mg and n=2 from 12 mg group). All analyses were Bonferroni corrected for multiple comparisons (i.e. the two valence categories).

Comment 22: What are the precedents for the linear relationship between arousal ratings for neutral stimuli and D1-D2 and D1-D3 increases in E2? – this warrants consideration.

Response: We apologize for the ambiguous description in the original manuscript. Before the arousal rating on Day 3, participants were explicitly instructed to rate the subjective arousal they experienced when viewing the pictures for the first time (i.e. on Day 2). We added this information to the methods section:

“They were instructed to rate the arousal they experienced when viewing the pictures for the first time (i.e. during encoding on Day 2). For this purpose, pictures were shown in the same order and with the same duration as during encoding (2 s).”

We agree that the significant relationship between E2 and arousal ratings for neutral but not negative pictures appears to be puzzling at first sight. First, we would like to point out how this result corroborates findings from a previous study, and then how it offers explanations for two of our hypotheses.

As we point out in the discussion, a similar pattern of arousal ratings for neutral and negative pictures was observed in a previous study on hormone replacement therapy (HRT; Pruis et al., 2009). However, whereas we found a more reliable E2 effect on neutral picture arousal ratings, they found more reliable effects with negative pictures. The differences in our results might be explained by differences in the baseline levels of arousal as well as the smaller sample size and lower E2 doses in the other study. Particularly remarkable are the baseline differences, the negative pictures in our study were rated substantially more arousing even in the placebo group than in Pruis et al. (2009), very likely leading to a ceiling effect.

In order to understand why higher E2 levels affect arousal for neutral pictures, we tested the following two hypotheses. First, E2 might increase the ‘emotional carry effect’, that is, arousal elicited by a negative picture could influence the rating of the next neutral picture. To test this possibility, individual arousal ratings for neutral stimuli following another neutral stimulus (‘neutral-after-neutral’) and neutral stimuli directly following a negative stimulus (‘neutral-after-negative’) were each averaged for every participant. Consistent with an emotional carry-over effect, neutral-after-negative stimuli ($M=2.89$, $SD=1.19$) were generally perceived as more arousing than neutral-after-neutral stimuli [$M=2.79$, $SD=1.22$; $t(122)=3.761$, $p<.001$]. However, robust regression analyses showed significant associations between salivary increase in E2 from Day 1 to Day 2 and arousal ratings for both neutral-after-negative [$t(121)=2.75$, $p_{corr}=.014$, $\beta_{std}=.261$] as well as neutral-after-neutral [$t(121)=2.65$, $p_{corr}=.018$, $\beta_{std}=.251$] pictures. Moreover, the emotional carry-over effect (arousal neutral-after-negative – arousal neutral-after-neutral) was not significantly associated to E2 [$t(121)=.190$, $p_{corr}>.999$, $\beta_{std}=.017$]. Thus, although carry-over effects were present in our data, they did not explain the effects of E2 on arousal ratings for neutral pictures.

Second, it is possible that viewing negative pictures might induce a negative mood over time, which in turn leads to greater arousal also with neutral pictures (‘mood induction effect’), an effect which might be increased by E2. Evidence that the mood induction effect could be enhanced by E2 is provided by Pruis and colleagues (2009; 2nd experiment). The authors found that women under HRT (compared to those not undergoing HRT) rated the neutral beginning of a story as less arousing than the also neutral third section, which was preceded by a negative story section. Our additional analyses corroborated the mood-induction hypothesis: arousal ratings for neutral pictures from the first third of the pictures seen were significantly lower than from the last third [$t(122)=-2.19$, $p=.030$]. Critically, the

difference of arousal ratings for neutral pictures (last third – first third) was positively associated with salivary E2 Day 1 to Day 2 increase [$t(121)=3.09$, $p_{corr}=.005$, $\beta_{std}=.148$]. In other words, the higher the increase in salivary E2, the higher the increase in arousal ratings for neutral pictures over time. As such, it is quite plausible that the increase in E2 increased mood induction effects and thereby arousal ratings for neutral pictures.

In response to the reviewer's comment we added these additional analyses to the Supplementary Material Section 1.5 and revised the discussion as follows:

“The increase of arousal for the neutral stimuli might be caused by a more pronounced induction of negative mood in response to the negative pictures with higher E2 levels, as further results of Pruis and colleagues⁶⁷ and our own supplementary analyses suggest (Supplementary Material 1.5). This finding is consistent with reports that supraphysiological doses of E2 can increase depression-like behavior and anxiety in rats^{19,22}.”

Pruis, T. A., Neiss, M. B., Leigland, L. A. & Janowsky, J. S. Estrogen modifies arousal but not memory for emotional events in older women. *Neurobiology of Aging* 30, 1296–1304 (2009).

Comment 23: Is it being suggested here that manual RT should be considered as a proxy for alertness? If so, the following sentence needs to be unpacked and reasoning explained, “exploratory analyses using reaction times from the active baseline task of the fMRI paradigm (see below) showed that the subjective effects in vigilance are not reflected in objective differences in a measure of alertness [$t(116)=-1.46$, $p=.146$].

Response: We apologize for this lack in clarity. The reviewer is correct in his/her understanding that manual RT were used as a proxy for intrinsic alertness which often is assessed by using simple reaction time tasks such as the arrow pointing task (Sturm & Willmes, 2001). In response to Reviewer #1, we also used accuracy in the arrow task as well as accuracy and reaction time in the encoding task (indoor/outdoor decisions) as additional proxies for alertness. Moreover, we realized that we noted the degrees of freedom incorrectly (df were 121 instead of 116). We rewrote the respective sentence:

“Only the first component (dimensions: alertness, strength, energy, clear-headedness; see Supplementary Table 13), determined by a principal component analyses on data from the mood questionnaire, showed a non-significant trend after Bonferroni correction [$t(121)=2.31$, $p_{corr}=.092$, $\beta_{std}=.188$]. We will refer to this principal component as ‘subjective vigilance.’ To test whether changes in subjective vigilance are reflected in an objective measure of alertness, individual reaction times from the distractor task (i.e. arrow task) and the encoding task were used as a proxy for intrinsic alertness. Neither reaction times (rt) nor accuracy in the arrow task [rt: $t(121)=-.68$, $p_{corr}>.999$, $\beta_{std}=-.011$; accuracy: $t(121)=-0.20$, $p_{corr}>.999$, $\beta_{std}=-.004$] or the encoding task (indoor/outdoor decision) [rt: $t(121)=-.57$, $p_{corr}>.999$, $\beta_{std}=-.047$; accuracy: $t(121)=-.37$, $p_{corr}>.999$, $\beta_{std}=-.023$] were significantly related to the Day 1 to Day 2 increase in salivary E2.”

Sturm, W. & Willmes, K. On the Functional Neuroanatomy of Intrinsic and Phasic Alertness. *NeuroImage* 14, S76–S84 (2001).

Comment 24: What were the other brain regions associated with the remember>know whole-brain contrast? Indeed, in the discussion, the authors aver, “Moreover, the task-related effects of E2 were highly selective and restricted to well defined brain regions although the paradigm evoked activity in a wide networks of areas.” Report these data with the appropriate corresponding statistics.

Response: We added a table (see response to comment 25) including all cluster associated to the remember/know contrast surviving a threshold of $p < .05$, family-wise error corrected for the whole scan volume to the Supplementary Material (Supplementary Table 8).

Comment 25: The regression analysis does not provide sufficient evidence to adjudicate whether or not E2 selectively modulates hippocampal activity associated with the remember>know contrast. Hippocampal activity needs to be compared against activity for the same contrast from at least one other neural region that is hypothesized to be affected by E2, such as the prefrontal cortex (e.g., Ottowitz et al., 2008) and a control region associated with recollection-based recognition memory such as lateral parietal cortex. Currently, it is not possibility to assess the selectivity of the main effect associated with one of the key conclusions. Indeed, in the analysis of the link between E2 levels and amygdala activity, the authors already tested/reported results from at least two other regions (right precuneus, right brainstem).

Response: To support the claim that ‘E2 selectively modulates hippocampal activity associated with the remember>know contrast’, we created masks from prefrontal clusters as well as a parietal cluster showing a significant remember>know main effect (highlighted in blue in the table below). Brain activity associated to the remember>know contrast did neither show a significant linear nor quadratic relationship either in the right parietal [linear: (33, -81, 36), $Z=1.49$, $p_{corr}=.872$; quadratic: (24, -66, 48), $Z=.68$, $p_{corr}=.976$], left parietal [linear: (-36, -87, 15), $Z=2.40$, $p_{corr}=.220$; quadratic: (-45, -78, 18), $Z=2.39$, $p_{corr}=.209$], right inferior frontal [linear: (42, 6, 24), $Z=1.33$, $p_{corr}=.845$; quadratic: (54, 36, 3), $Z=.72$, $p_{corr}=.836$], or left inferior frontal cluster [linear: (-48, 33, 12), $Z=.30$, $p_{corr}=.630$; quadratic: (-45, 33, 12), $Z=-.22$, $p_{corr}=.678$] small volume corrected for the search volume (i.e. the respective masks).

Moreover, there was no significant linear or quadratic relationship to E2 within a 5mm sphere centered around the peak in the right superior frontal gyrus [linear: (15, 60, 24), $Z=1.20$, $p_{corr}=.405$; quadratic: (18, 57, 24), $Z=0.25$, $p_{corr}=.632$] and right middle frontal gyrus [(39, 18, 51), $Z=-.20$, $p_{corr}=.679$; quadratic: (36, 18, 51), $Z=-.26$, $p_{corr}=.672$] reported by Ottowitz et al. (2008). We added these analyses to the Supplementary Material Section 1.7.3.

	main effect: remember>know					
region	x	y	z	Z	p _{corr}	k at p _{corr} <.05
left hemisphere						
medial temporal cortex	-30	-30	-18	7.39	<.001	419
precuneus	-6	-54	12	6.33	<.001	49
superior parietal cortex	-42	-75	18	5.82	<.001	105
inferior frontal operculum	-39	6	24	5.79	<.001	27
orbito-frontal gyrus	-39	30	-18	5.58	.001	34
medial frontal gyrus	-3	42	-18	5.39	.002	31
inferior frontal gyrus	-48	30	12	5.37	.002	18
right hemisphere						
inferior frontal gyrus	48	36	9	7.55	<.001	176
inferior temporal gyrus	48	-57	-12	7.73	<.001	109
superior parietal cortex	30	-78	30	6.85	<.001	241
medial temporal cortex	30	-42	-9	6.78	<.001	204
orbito-frontal gyrus	27	30	-15	6.21	<.001	18
medial temporal gyrus	54	-6	-18	5.69	<.001	13
calcarine sulcus	18	-51	9	5.42	.001	17
posterior cingulate	9	-54	12	4.85	.023	2

Supplementary Table 8. Brain regions associated to the remember>know contrast family-wise error corrected for the whole scan volume (N=118).

Ottowitz, W. et al. Evaluation of prefrontal–hippocampal effective connectivity following 24 hours of estrogen infusion: An FDG-PET study. *Psychoneuroendocrinology* 33, 1419–1425 (2008).

Comment 26: Additionally, a PPI analysis is reported for the contrast between neutral and negative stimuli, yet a corresponding analysis is not reported to explore changes in functional connectivity associated with the remember>know contrast. Although not stated a priori, the rationale for the PPI analysis is reasonable, therefore, it is unclear why the authors did not conduct and report a PPI analysis for the remember>know contrast for a hippocampal seed region to examine functional connectivity with, for example, the PFC – another key target of E2.

Response: We agree and explored hippocampal functional connectivity associated with the remember>know contrast. We created a mask from the hippocampal cluster showing a linear relationship to salivary E2 increase [(33, -30, -12) thresholded at $p_{unc}<.001$]. This mask was then used as a seed region in a psycho-physiological interaction. However, neither connectivity with any other hippocampal cluster [greatest positive effect: (27, -27, -12), $Z=2.84$, $p_{corr}=.321$; greatest negative effect: (-21, -18, -27), $Z=3.04$, $p_{corr}=.207$] nor a region in the amygdala [greatest positive effect: (-24, 0, -15), $Z=1.28$, $p_{corr}=.902$; greatest negative effect: (-33, -6, -30), $Z=2.56$, $p_{corr}=.290$] or in the prefrontal cortex [greatest positive effect: (39, 39, -18), $Z=2.90$, $p_{corr}>.999$; greatest negative effect: (-39, 18, -6), $Z=2.95$, $p_{corr}>.999$] showed a significant relationship with changes in salivary E2.

For completeness sake, we also conducted a PPI using the cluster showing a significant quadratic relationship to salivary E2 increase [(-15, -33, -12) thresholded at $p_{unc}<.001$] as

the seed region. Neither connectivity with any other hippocampal cluster [greatest positive effect: (15, -33, -6), $Z=2.59$, $p_{corr}=.503$; greatest negative effect: (-36, -15, -21), $Z=2.14$, $p_{corr}=.801$] nor an amygdala cluster [greatest positive effect: (24, -6, -12), $Z=1.69$, $p_{corr}=.773$; greatest negative effect: (36, -6, -21), $Z=1.83$, $p_{corr}=.712$] or a prefrontal cluster [greatest positive effect: : (-39, 15, -18), $Z=2.82$, $p_{corr} >.999$; greatest negative effect: (21, 24, 36), $Z=2.57$, $p_{corr}=.806$]. We added these analyses to the Supplementary Material Section 1.7.4.

Comment 27: The neural regions associated with encoding success – hits>misses (Page 10, Lines, 14-15) – should (a) be reported in full and (b) the region(s) from this contrast tested in the reported regression should be stated along with the appropriate statistics.

Response: We agree and added a table containing all brain regions associated to general encoding success (i.e. subsequent hits>misses) family-wise error corrected for the whole scan volume to the Supplementary Material (Supplementary Table 9). Moreover, we added statistics for the regression between salivary E2 increase and brain activity associated to encoding success within regions of interest to the results section of the main manuscript.

“Brain activity linked to general encoding success (subsequent hits > misses; see Supplementary Table 9 for main effects) neither showed a linear nor quadratic relationship in the hippocampus [linear: lowest p_{corr} of .290 at (-18, -18, -18); quadratic: lowest p_{corr} of .184 at (27, -12, -18)] or the amygdala [linear: lowest p_{corr} of .798 at (27, -6, -6); quadratic: lowest p_{corr} of .451 at (-30, 0, -27)].”

Supplementary Table 9. Brain regions associated to general encoding success (i.e. subsequent hits>misses) family-wise error corrected for the whole scan volume (N=121).

region	main effect: hit>miss					
	x	y	z	Z	p_{corr}	k at $p_{corr} <.05$
left hemisphere						
medial temporal lobe	-27	-39	-15	8.70	<.001	1270
middle & inferior frontal gyrus	-36	33	-15	8.40	<.001	358
superior frontal gyrus	-6	57	27	7.29	<.001	92
precuneus	-18	-54	9	6.40	<.001	70
middle temporal gyrus	-54	-9	-18	6.38	<.001	56
superior temporal gyrus	-45	18	-21	6.21	<.001	8
corpus callosum	-6	6	24	4.78	.028	1
right hemisphere						
middle & inferior frontal gyrus	42	12	24	9.21	<.001	541
medial temporal lobe	21	-9	-21	8.82	<.001	1280
medial frontal gyrus	3	42	-18	7.93	<.001	112
middle temporal gyrus	57	-3	-18	7.27	<.001	63
precuneus	15	-51	12	6.94	<.001	94
superior temporal gyrus	45	18	-33	6.59	<.001	16
thalamus	3	-18	6	5.63	<.001	15
middle frontal gyrus/precentral gyrus	48	-3	51	5.43	.001	14
inferior occipital	27	-93	-6	4.74	.032	2

Comment 28: As mentioned, it is repeatedly stated that, “A large body of animal studies intensely characterized the beneficial effects of E2 on hippocampal neuroplasticity and hippocampus-dependent memory”, yet the analyses do not reveal a significant effect of E2 levels on hippocampal “neuroplasticity”. It thus remains unconvincing how these data necessarily “bridge the gap” in the way the authors suggest because:

Comment 28.1: there is no direct assessment of how hippocampal activity measured using event-related fMRI is a proxy for neuroplasticity observed in animal studies – the key contrast is based on a difference between remember>know rather than a learning-related neuroplastic change in activity over, for example, time at encoding. Indeed, a more standard approach based on encoding success (hits>misses) reveals no link with E2 levels; and,

Response: We agree that an explanation of how fMRI might be able to measure E2 induced changes in neural plasticity should be added. We also acknowledge that the rationale for using the remember>know rather than the general encoding success contrast was missing.

In response to the reviewer’s comment we added the following explanation to the discussion to address the first point:

“Via both rapid and genomic effects, E2 modulates synaptogenesis, glutamatergic neurotransmission and LTPs, among other processes^{3,4,14}. Given that the BOLD response measured in fMRI mainly reflects glutamatergic synaptic activity, with greater glutamate release at individual synapses and higher synapse density leading to greater BOLD signal^{47–49}, the E2-mediated changes in neural activity in our data likely reflect differences in synapse density and/or glutamatergic neurotransmission. Such changes seen 24 hours after E2 levels have been increased, as in the current design, can be driven by both rapid and genomic effects⁴.

In order to address the second point we added the following paragraph to the introduction:

“Most animal studies have investigated E2 effects on hippocampal neurons and hippocampus-dependent learning and memory. It also has been reported to influence the shift from hippocampus-independent to -dependent learning strategies^{34,35}. Therefore, we employed a memory paradigm that allowed us to disentangle hippocampus-independent from hippocampus-dependent recognition memory^{36,37}. ‘Recollecting’ an item, i.e. together with contextual information about the encoding episode, relies critically on the hippocampus, whereas the acontextual sense of ‘familiarity’ with an item relies on parahippocampal structures and is independent of the hippocampus³⁶.”

And this sentence to the discussion:

“Interestingly, these relationships were specifically observed for brain activity related to hippocampus-dependent memory (i.e. the remember>know contrast) but not memory formation in general (general encoding success contrast).”

Comment 28.2: there is no connection between hippocampal activity and hippocampal-dependent behavioural performance;

Response: We agree that we did not discuss this discrepancy sufficiently. In response to

the reviewer's comment we substantially rewrote the discussion of the missing memory effects as follows:

“Many animal studies have observed that E2 affects not only hippocampal neuronal plasticity but also hippocampus-dependent memory performance^{1,2}. In order to increase sensitivity to detecting E2 effects on memory, we employed a paradigm which disentangles hippocampus-dependent from hippocampus-independent memory processes. We were previously able to detect changes in hippocampus-dependent memory for negative stimuli throughout the menstrual cycle using a similar paradigm²⁷. Perhaps it seems surprising that we did not find a relationship between E2 levels and memory performance in the current study despite the effects of E2 on hippocampal activity. However, we have observed a similar dissociation of neural and behavioral effects of E2 before. Specifically, we had found that a genetic polymorphism related to higher E2 levels in young men was associated with larger posterior hippocampi but not with better hippocampus-dependent memory^{18,30}. Of note, the hormonal effects on memory performance that we detected in two other studies were very specific: we found effects on the hippocampus-dependent recollection of the temporal order of words but not word-color associations in postmenopausal women, and on the recollection of negative but not neutral or positive pictures in young women^{26,27}. Similarly, despite the many studies providing evidence for a positive relationship between hippocampus-dependent memory and E2 levels within a physiological range in rodents^{1,2}, some studies report this effect to be task-specific, while others did not find any effects on behavior^{15,16,18,62}.

Given our tightly controlled design, the wide range of E2 levels, our highly estrogen-sensitive sample (i.e. young, naturally cycling women), and use of an experimental paradigm which has been shown to be highly sensitive to changes in hippocampus-dependent memory, the lack of relationship found between E2 levels and memory performance very likely reflects a true null effect rather than confounds or noise induced by the design (Figure). Perhaps the reason why behavioral effects did not echo the neural effects of the E2 treatment regimen was because 24 hours was long enough to increase synapse density and the associated glutamate release, as reflected by the fMRI signal, but too short for the new synapses to be functionally effective. This interpretation would also imply that the pre-ovulatory E2 peak in the human menstrual cycle might be too short – at least in some women, as there is large variability in cycles - to result in mnemonic changes, which could explain some of the inconsistencies across studies^{32,63,64}. However, exogenous E2 can already enhance memory performance in rats after 4 hours⁵⁵, suggesting inter-species differences in the time course of the functional integration of newly formed synapses as it has been reported for newly generated neurons in rats and mice⁶⁵. An alternative account for the absence of an effect on memory performance could be that the hippocampal activity that is modulated by E2 does not reflect memory formation but hippocampus-guided visual exploration⁶⁶.”

Comment 28.3: what are the empirical precedents for postulating that the modulation in hippocampal activity is mediated by between-group differences in synaptic density that have occurred over a day? (Page 15); and,

Response: We agree with the reviewer that the reader who is not very familiar with the animal research on E2 effects should be informed about the time frame of the changes it induces on synaptic plasticity. In general, the effects of E2 on neuronal plasticity are surprisingly fast. Increases in spine density and LTPs have been observed after 30 minutes (Frankfurt and Luine, 2015; Hasegawa et al., 2015), and recognition memory enhancements 4.5 hours after estrogen treatment (Luine et al., 2003). In mice, estrous-cycle dependent changes of hippocampal volume occur within 24 hours (Qiu et al., 2013). Although the rapid effects of E2 are often attributed to the non-genomic action pathways of E2 – e.g. interactions with signaling cascades such as Erk MAPK (Hasegawa et al., 2015) and synaptic proteins such as PSD-95, spinophilin and synaptophysin (for a review see Frankfurt and Luine, 2015) – genomic effects have also been observed as early as after 2-4 hours (Marino et al., 2006; Sheppard et al., 2017). We added more details on the cellular mechanisms of E2 actions to the discussion.

In response to the reviewer's comment we describe the cellular actions of E2 in the revised discussion in more detail (see above) and explicitly state that:

“Such changes seen 24 hours after E2 levels have been increased, as in the current design, can be driven by both rapid and genomic effects (Mukai et al. 2010).”

Luine, V. N., Jacome, L. F. & MacLusky, N. J. Rapid enhancement of visual and place memory by estrogens in rats. *Endocrinology* 144, 2836–2844 (2003).

Frankfurt, M. & Luine, V. The evolving role of dendritic spines and memory: Interaction(s) with estradiol. *Horm Behav* 74, 28–36 (2015).

Hasegawa, Y. et al. Estradiol rapidly modulates synaptic plasticity of hippocampal neurons: Involvement of kinase networks. *Brain Res.* 1621, 147–161 (2015).

Marino, M., Galluzzo, P. & Ascenzi, P. Estrogen Signaling Multiple Pathways to Impact Gene Transcription. *Curr Genomics* 7, 497–508 (2006).

Mukai, H. et al. Modulation of synaptic plasticity by brain estrogen in the hippocampus. *Biochimica et Biophysica Acta (BBA) - General Subjects* 1800, 1030–1044 (2010).

Qiu, L. R. et al. Hippocampal volumes differ across the mouse estrous cycle, can change within 24 hours, and associate with cognitive strategies. *NeuroImage* 83, 593–598 (2013).

Sheppard, P. A. S., Koss, W. A., Frick, K. M. & Choleris, E. Rapid actions of estrogens and their receptors on memory acquisition and consolidation in females. *J Neuroendocrinol* n/a-n/a doi:10.1111/jne.12485

Comment 28.4: the current way in which the subfield-task-computational interactions are discussed as part of the conclusions almost implies that such analyses were conducted; this is misleading and should be revised to avoid the implication that the analyses were conducted at the level of hippocampal subfields, since it is entirely speculative in relation to the current study; namely, Lines Page 19-20.

Response: We agree that the wording was misleading. As stated in comment 11.1, we now avoid the term ‘subregion’ when discussing our results and instead speak of ‘different/distinct hippocampal clusters’. Moreover, we explicitly mention the impossibility of hippocampal subfield analyses due to the spatial resolution in the discussion.

“However, the spatial resolution of our fMRI scans is too coarse to relate the different dose-response curves to specific anatomically defined hippocampal subfields and only permits a functional dissociation.”

Comment 28.5: Another consideration overlooked when discussing the interpretation of the main result is that the link between E2, hippocampal activity and behaviour was only examined from the perspective of an offline recognition memory test – the hippocampus has been extensively implicated outside of recognition memory, so it is conceivable that the E2 mediated-modulation of encoding-related hippocampal activity has an impact on cognitive behaviour outside of “memory paradigms”. This possibility should at least be introduced and discussed.

Response: We thank the reviewer for directing our attention to this alternative interpretation. In response to the reviewer’s comment we added this alternative account to the discussion:

“An alternative account for the absence of an effect on memory performance could be that the hippocampal activity that is modulated by E2 does not reflect memory formation but hippocampus-guided visual exploration⁶⁶.”

Voss, J. L., Bridge, D. J., Cohen, N. J. & Walker, J. A. A Closer Look at the Hippocampus and Memory. *Trends Cogn. Sci. (Regul. Ed.)* 21, 577–588 (2017).

Comment 28.6: Furthermore, a more direct test of the significance of E2-mediated changes on hippocampus-dependent memory would have been to acquire fMRI data during the recognition test.

Response: We agree that exploring the effects of E2 on hippocampus-dependent memory retrieval would be an interesting research question as well. We decided for the present (first) study to focus on encoding, as much less is known about the effects of E2 on memory retrieval from animal research, while, in contrast, the underlying mechanisms by which E2 enhances neural plasticity that is relevant for memory encoding and storage have been well described (see above). While beneficial effects of E2 within the hippocampus are well evidenced for the learning as well as the consolidation phase, E2 does not enhance memory in rodents when it is administered after the consolidation window (for a review, see Sheppard et al., 2017 and Frick, 2013). Moreover, such a study would require administering E2 after memory encoding – otherwise fMRI data collected during memory recognition would reflect a mixture of E2 effects on memory acquisition, consolidation and potentially also on retrieval. We also decided to increase E2 levels prior to encoding and during the encoding phase because we were also interested in the effects of E2 on affective processing and on the modulation of memory encoding by arousal (Murty et al., 2010).

Frick, K. M. Epigenetics, estradiol, and hippocampal memory consolidation. *J Neuroendocrinol* 25, 1151–1162 (2013).

Murty, V. P., Ritchey, M., Adcock, R. A. & LaBar, K. S. fMRI studies of successful emotional memory encoding: A quantitative meta-analysis. *Neuropsychologia* 48, 3459–3469 (2010).

Sheppard, P. A. S., Koss, W. A., Frick, K. M. & Choleris, E. Rapid actions of estrogens and their receptors on memory acquisition and consolidation in females. *J Neuroendocrinol* n/a-n/a doi:10.1111/jne.12485.

Comment 29: Please re-write:

Comment 29.1: “24 h later a recognition test was accomplished outside of the scanner using confidence ratings followed by remember/know judgements for old responses (58).” - a recognition test was “administered” and explain event structure with more step-wise detail;

Response: We changed the wording of the respective sentence. A more detailed description of the event structure is now given in Figure 1 and the methods section.

Comment 29.2: Relatedly, in the abstract, the timeline is unclear, “Memory performance was assessed 24 hours later by a recognition test”...memory performed was assessed 24 hours after initial encoding?; and,

Response: We substantially revised the abstract and now state the timeline more clearly:

“Brain activity was measured when E2 levels peaked while volunteers encoded neutral and negative pictures, and their memory was tested 24 hours after encoding.”

Comment 29.3: Please clarify... “, designed to disentangle hippocampus-dependent from - independent memory processes.” - as written, it is unclear how a recognition test per se disentangles hippocampal-dependent and independent “retrieval” processes. This needs to be re-written to address this point.

Response: We revised the respective sentences in the abstract, the introduction and the discussion, e.g.:

Introduction

“Therefore, we employed a memory paradigm that allowed us to disentangle hippocampus-independent from hippocampus-dependent recognition memory^{36,37}. ‘Recollecting’ an item, i.e. together with contextual information about the encoding episode, relies critically on the hippocampus, whereas the acontextual sense of ‘familiarity’ with an item relies on parahippocampal structures and is independent of the hippocampus³⁶.”

“Twenty-four hours after encoding, a recognition test was administered outside of the scanner with confidence ratings followed by remember/know judgements for ‘old’ responses, in order to model recollection and familiarity (Figure 1).”

Methods

“These confidence ratings were used to estimate the contribution of recollection and familiarity to recognition memory. The Dual Process model of recognition memory was fitted to the receiver-operating characteristic (ROC) curves, which were derived from the hit and false alarm rates across confidence levels (Supplementary Figure 2) using maximum likelihood estimation⁴⁵. This was done separately for neutral and negative items and performed in Matlab R2014b (Mathworks, Inc, Natick, MA, USA). For exploratory reasons, confidence ratings were additionally used to estimate meta d-prime and meta criterion⁹¹ (see Supplementary Table 3). The remember/know procedure (see Supplementary Material Section 2.3 for details) was used to back sort each encoding trial as to whether they were subsequently retrieved with contextual details (hippocampus-dependent ‘recollection’) or only with a sense of familiarity (hippocampus-independent ‘know’).”

Comment 29.4: Should read, “were significantly,” in the following sentence: “Neither recollection, which is thought to be hippocampus-dependent, nor familiarity was significantly

associated to E2 in a linear or a quadratic manner [all $p < .05$; Supplementary Table 1].”

Response: We corrected the respective sentence.

Comment 29.5: Should read, “a psycho-physiological interaction (PPI; 62) analysis was conducted, in the following sentence: “To examine exploratory whether E2 administration also affected the connectivity between precuneus and brainstem during emotional processing, a psycho-physiological interaction (PPI; 62) analysis was calculated.”

Response: We corrected the respective sentence.

Comment 29.6: The following sentence needs to be re-written because it is unclear what was being examined in this subsidiary analysis, “However, exploratory analyses using reaction times from the active baseline task of the fMRI paradigm (see below) showed that the subjective effects in vigilance are not reflected in objective differences in a measure of alertness [$t(116) = -1.46$, $p = .146$].” See above.

Response: We revised the sentence as follows:

“To test whether changes in subjective vigilance are reflected in an objective measure of alertness, individual reaction times from the distractor task (i.e. arrow task) and the encoding task were used as a proxy for intrinsic alertness.”

Comment 29.7: Grammar, Page 11, Lines 5-6: “To examine exploratory whether E2 administration also affected the connectivity between precuneus and brainstem during emotional processing.”

Response: We corrected the sentence as follows:

“To explore whether E2 administration also affected the functional connectivity between precuneus and brainstem during emotional processing, a psycho-physiological interaction (PPI) ⁴⁶ analysis was conducted.”

Comment 29.8: Pages 20-21, what is the unit related to the mean value reported on lines 21-22;1-2, respectively?

Response: We apologize for these oversights and corrected the corresponding sentences:

“The first day was matched as closely as possible to menstruation onset ($M = 1$ day, $SD = 4$ days after menstruation onset). The first dose of E2 was administered double-blind by the experimenter in the evening of day 1 ($M = 15.45$ hours before second dose on Day 2, $SD = 1.37$ hours). Participants took the second dose on their own the next morning of Day 2 ($M = 5.71$ hours before testing on Day 2, $SD = 0.90$ hours).”

Comment 29.9: What was the amount of financial compensation given to the participants?

Response: The participants received a financial compensation of 120 € for the time spent in the institute. We added this information to the methods section.

Reviewers' comments:

Reviewer #2 (Remarks to the Author):

The manuscript has improved but there are still have some remaining mostly minor concerns.

1. The authors claim their doses are in the supraphysiological range but is this really correct? Levels of estradiol are very high during pregnancy so it would seem these levels are indicative of lower levels during the first trimester. I would reword this to indicate that these are high levels, and higher than during menstrual cycle but not during pregnancy. Also change on line 307
2. Line 87 – Although other studies may not have looked at exogenous E2, there are a number of studies examining the influence of menstrual cycle fluctuations in women – notably by the seminal studies of Elizabeth Hampson. Furthermore, there are a number of studies examining doses of E2 in postmenopausal women that could be at least referred to or one dose in younger women (Bartholomeusz CF).
3. in references 3,4,14, 50-53 – are these experiments done in males or females – given that this work is in women work cited should be in females. Especially for reference 4 – most of the data with brain estrogens has been obtained in males or in embryonic tissue. Some studies examining adult female levels show very little brain estrogens (Kato, Barker).
4. Lines 349-353 – While it is tempting to imagine that more spines is equivalent to better memory has this been definitively shown? Spines may not equal memory and if my memory serves LTP is not related to MORE spines but larger spines, a subtle but important distance and it's better to not equate number of spines to memory.
5. The discussion is very long and could benefit from some trimming (perhaps parts of the long discussion of spines).
6. lines 400-402 – the different findings are also related to different doses of E2 – EVEN WHEN IN THE physiological range. High E2 is associated with poorer memory while diestrous levels are associated with better spatial memory (very much akin to what E Hampson demonstrated in women across menstrual cycles). Other discrepancies lie within the idea that E2 is given AFTER or DURING the learning. Frick gives E2 during consolidation whereas other given it during the learning – so outcome differences may be due to these differences as well. Given that your data is based on during learning it is perhaps better to concentrate on those studies.
7. Some references are reviews and not primary papers. More papers are needed that have actually investigated dose effects in rodents. (Barha CK)

Minor

Line 53 – citation needed

Line 546 – something wrong here.

Reviewer #3 (Remarks to the Author):

This is an interesting study that attempts to answer several questions related to estrogen effects on memory. The first main question explores the dose-response curve focusing on the hippocampus. The second main question explores the same, but in relation to the amygdala and emotional processing. I do find the results to be interesting and worthy of dissemination. Authors have addressed my previous comments to my satisfaction and the manuscript has improved in terms of clarifications and methodological detail that is provided. In the process of addressing reviewers' comments, however, the manuscript has become very lengthy and much of this could be avoided, as most of the introduction is very repetitive. In trying to be thorough, the authors lose the reader as far as keeping the human vs. non-human literature clear, also amygdala vs. the hippocampus. The amygdala seems like an afterthought. Although the Discussion is fairly extensive, a lot of it doesn't focus directly on the results, and most of it focuses on dose-dependency of the curve – not the fact that there are robust and widespread network changes without much of a behavioral effect.

I would suggest that the author re-write, trim and better focus the manuscript prior to my being able to recommend publication.

Reviewer #4 (Remarks to the Author):

The authors have substantially revised their manuscript and clarified all my previous concerns. However, I still think that the results are complex and difficult to interpret and that this study contributes more to the heterogeneity of the current literature – but that might be an important aspect of this study in itself. But, nevertheless, the revision was quite satisfying and substantial.

Besides of that, I stumble about a few (minor) issues that need to be clarified:

1. Page 12, line 8: Please specify „relevnt activation peaks“ and provide analysis in supplementary material, if necessary.
2. Figure7: Although the authors used robust regression, the inverted U-shaped relationship in the insula (as displayed in Figure 7) might be driven by the single data point in the lower right corner. Have the authors run the analysis without that data point?
3. Page 36, line 17: It doesn't become quite clear to my why the authors created the ROIs of interest. It seems that they wanted to use them for a small volume correction, but they say at the same time that they use the FWE-correction of the entire scan volume. If that is already significant, a small volume correction does add something. Therefore, I'm wondering whether the reported $p(\text{corr})$ values refer to the scan volume or to the ROIs volume. Further, one shouldn't use more than one ROI for a small volume correction procedure, i.e. if one expect effects to appear in the amygdala AND hippocampus, one should create an ROI that cover both and not one for each area separately. Otherwise, one would need a Bonferroni correction (or similar) on the number of ROIs, as well.

Reviewer #5 (Remarks to the Author):

Bayer et al. have addressed the majority of concerns raised in my initial appraisal. The manuscript is much improved. Additional comments emerging from their responses to two of the original main comments and new comments in response to manuscript revisions are detailed below:

"We agree with the reviewer on the importance of using appropriate, unbiased statistical thresholds. We use only 2 different statistical thresholds in the main manuscript. All results in the main text and tables and, importantly, for any conclusions made in the discussion, are based on a corrected statistical threshold of $p < .05$ FWE, unless otherwise clearly noted as non-significant trends (e.g. brainstem results). We also use an uncorrected threshold of $p < .005$ purely for displaying purposes in the figures. Only the exploratory analyses in the Supplementary Material are reported using an uncorrected threshold of $p < .001$ to convey a more comprehensive picture of our findings.

Our motivation for the more lenient display threshold was to show a more comprehensive view of the activations. This is a conventional strategy employed by many other fMRI studies. In our opinion, this display threshold represents a good compromise between sensitivity and specificity when visualizing the results in this study. Lieberman & Cunningham (2009) have even suggested, based on statistical modeling, that an adequate balance between Type I and Type II errors can be achieved at an uncorrected voxel-wise threshold of $p < .005$ cluster and an extent threshold of $k > 10$. All our figures already meet this threshold. If the reviewer is still concerned, we would suggest reporting cluster sizes at $p_{corr} < .05$ in the relevant tables or add figures using the desired threshold."

(1) Appealing to prior conventions is not entirely appropriate here, particularly with the current debate about reproducibility of results and more recent discussions about the selection of appropriate thresholds. The Liberman & Cunningham (2009) paper coincides with an era where many of the main fMRI results in papers were reported as uncorrected, or, if corrected for multiple comparisons, were not based on a computation of, for example, FDR (Chumbley & Friston, 2009, Chumbley et al. 2010). In addition, the approach still lacks objectivity – all inferences about the neuroimaging data in the manuscript are drawn by applying a FWE-corrected $p < 0.05$ threshold, whereas the figures that refer and claim to depict same results do not correspond to the same thresholding. The figures should depict the appropriately thresholded results first and be elaborated with uncorrected $p < 0.005$ if desired. Therefore, please revise and add figures at the appropriately corrected threshold $p < 0.05$.

--

"The Reviewer raises a point here over which we belabored before deciding on presenting the results that we have. In the end, we decided to also report the three non-significant trends based on the following rationale. This was a particularly involved study, both in terms

of the size of our sample and the E2 regimen (i.e. scanning only in the menstruation phase), which required much organization and financial resources. We believe it will be unlikely that too many similar studies will be conducted of this scale. We therefore reasoned that we should also report the trends that did not survive our conservative statistical thresholds in order to avoid type II errors, to indicate possible avenues of investigation, and as a comparison for future studies that might observe similar results.

For instance, the non-significant trend in the brainstem might direct animal researcher's attention to this brain area and can also provide an a-priori ROI for future human fMRI studies. The non-significant trend in the amygdala with respect to perfusion could motivate future studies to use more powerful methods to detect such differences (for instance a longer ASL sequence including more images). Finally, the non-significant trend for a positive linear relationship between salivary E2 increase and the first PCA component (previously labeled as 'Factor 1') derived from the questionnaire could be tested in studies using methods designed to specifically assess 'subjective vigilance'. In response to the reviewer's comment, we decided to place even greater emphasis on the non-significance of the results which did not survive our statistical thresholds and replaced the term 'trend-level significant' by 'non-significant trends/tendencies' throughout the manuscript.

If the reviewer is still concerned with our results being misinterpreted, we would remove the non-significant findings from the manuscript."

(2) It is entirely reasonable to include reports of non-significant results, for the reasons outlined above. Non-significant results should be reported – that's not my main concern.

What remains at issue is the manner in which non-significant results are being described. The most neutral way to report a non-significant result is to state simply, that it was non-significant – the result is thereby reported for the record and the reader left to adjudicate on Type II considerations, within the statistical power of the design to detect intended and unintended effects.

Repeatedly stating results indicate a "non-significant trend/tendencies" is misleading and introduces a systematic bias in interpretation because (1) it is suggestive of Type II errors. Increased statistical power, for instance, may just as well push the result even further above the alpha criterion; and, (2) as mentioned previously, a significant p-value at the margins, such as $p=0.03$, needs by logical extension to be reciprocally referred to either as a "significant trend/tendencies or by extension "trending towards non-significance", especially when considered against the recent push towards a more conservative alpha criterion ($p=0.0005$) in the field (Benjamin et al., PsyArxiv, 10.17605/OSF.IO/MKY9J).

All that said, the potentially misleading nature of reporting trends can easily be avoided by stating simply that the results were non-significant when $p>0.05$ and significant when $p<0.05$. So, please just simply remove reference to the words trends/tendencies and leave the reporting of non-significant finding otherwise intact. In addition, further discussion for each of the outlined cases above can be introduced if the authors desire to speculate on why a particular non-significant result warrants further attention and might be a Type II

error.

Additional comments:

Abstract

“Brain activity was measured when E2 levels peaked while volunteers encoded neutral and negative pictures, and their memory was tested 24 hours after encoding. Memory-related activity in distinct posterior hippocampal clusters both linearly increased as well as followed an inverted U-shape with E2 dose.”

(a) The specific memory task and nature of the dependent measures, rather than “their memory”, should be stated in the abstract. Presumably, the authors are again referring to encoding related activity when going onto refer to “memory-related activity” – this needs to be restated to improve precision in the manner the main effect is being described in order to minimize the potential for confusion in a reader.

“E2 did not affect hippocampus-dependent memory performance, suggesting that the observed E2 effect on neural activity after 24 hours was not effective in improving memory formation.”

(b) One of the main conclusions is an inference based on null effect. There are several concerns here (1) the null effect refers to the relevance of E2 on encoding-related hippocampal activity for later recognition memory. It is not apparent from the current summary that this is the case; (2) it’s confusing how this summary refers to the “E2 effect on neural activity after 24 hrs” – only behaviour was measured at 24 hrs; reference can only be made to behaviour and not neural activity/experience-dependent plasticity; (3) there needs to be some indication in the abstract why encoding and recognition-based retrieval of neutral and emotional stimuli should be considered a de facto measure of hippocampal activity given the outcomes; (4) consideration in the abstract should also be added to reflect that the null result could reflect a suboptimal measure of “memory performance”, suboptimal interval post-initial encoding to assay the effects on memory performance, or effects of E2 on hippocampal activity that are associated with other operations such as perceptual and/or attentional processing.

Revision of the abstract is important because the implication, as currently written, is that E2 has no impact on all memory performance, but this was entirely conditional on (a) the use of a single assay, recognition memory, and, (b) assessing the effects of E2 at 24 hr post-scanning and using fMRI as a proxy for neural plasticity only at encoding. By the authors’ own estimation in the discussion, prior effects can be task-specific, so it is somewhat sweeping to infer with the current protocol that the use of a single recognition test stands as a de facto and sufficient measure with which to judge the effects of E2 on human memory-guided behaviour. Recognition based recognition does not always engage the hippocampus; inclusion of an additional test such as cued-recall would have provided a more compelling foundation on which to investigate the wider effects on memory-guided

behaviour. A version of these considerations need to be communicated in the abstract, especially because the results are at variance with evidence of hormonal effects on memory and affective processing. As it stands currently, the conclusion over reaches in this regard.

Introduction

"That is, one would expect increasing benefits until medium levels of E2 but then a decrease and none or even negative effects at high levels. even negative effects at high levels."..

"Whereas the monotonic increase and saturation is compatible with the traditional sigmoid dose-response function of pharmacological studies, the inverted U-shaped curve needs to be explained by a more intricate pattern of underlying processes.".. "With respect to learning and memory, the diversity of regimens and tasks makes it more difficult to see a clear pattern".. and later in relation to the links between behaviour measures used in studies of model organisms and humans, "However, in addition to differing in general design to that of animal studies.."

(a) A brief but more specific consideration regarding how neural plasticity and behaviour were quantified in prior studies involving model organisms could provide further mechanistic links between these variables, especially given that the current study does not elucidate a link between BOLD activity – as a proxy of neural plasticity – and a single assay of memory-guided behaviour. It is not unreasonable to conclude that some of the variability in prior studies may be epiphenomena of the research protocols rather than E2 effects on neuroplasticity and by extension on memory-guided behaviour, per se. Most usefully, this summary could draw specific links between behavioural protocols used in model organisms and rationale for the behavioural protocol in the current study, especially because the current study already uses a very different proxy/operational definition - from the studies of model organisms – of neural plasticity – and finds a null link with behaviour. Behavioural tasks between model organisms and humans are frequently equated to minimise the obvious sources of noise/confounds.

(b) Repeated reference is made to "neural plasticity" in model organisms and then changed in humans to a discussion of neural activity "The effects of E2 on neural activity and behavior"; e.g., "... dose-response relationship between E2 and neural plasticity as well as behavior.."; "Very few studies have investigated the effect of different E2 doses on neural plasticity...";

In a journal with a specialist and non-specialist readership, the use of the term neural plasticity needs to be applied with greater precision, explaining how neural plasticity is measured / conceptualized in each case; at best, the study design can indirectly examine experience-dependent change as a result of neural plasticity measured by BOLD activity. Please review the relevant text to improve the precision with which the terms is applied.

Results

"Determined by a principal component analyses on data from the mood questionnaire,

showed a non-significant trend after Bonferroni correction [$t(121)=2.31$, $p_{corr}=.092$, $\beta_{std}=.188$].”

(a) As per my earlier comment: this should read, was non-significant after Bonferroni correction – the p-value is non-significant at the Bonferroni corrected p value, it is not trending (to) non-significance. Also, I am not sure if it is entirely necessary to link the follow-up analysis of RT data to what is a non-significant component labelled subjective vigilance. The analysis of RT is helpful and relevant as a standalone subsection. I think the link diminishes the line of reasoning, but I leave it to the authors discretion if they wish to sever the link with the non-significant principal component.

“Neither recollection, which is thought to be hippocampus-dependent, nor familiarity was significantly associated with E2 in a linear or a quadratic manner [all p_{corr} 's $>.2$; ; Supplementary Table 3]. A similar pattern emerged when recollection and familiarity were computed based on the remember/know judgements [all p_{corr} 's $>.2$].

Neither total recognition accuracy in terms of d-prime, the emotional enhancement of memory (=d-prime for negative pictures – d-prime for neutral pictures), nor response criteria showed a linear or quadratic relationship to the increase in salivary E2 [all p_{corr} 's $>.3$].”

(b) It is helpful to state/summarise here in the main text whether recognition based measures were significantly different from chance in the first instance before then proceeding to report the results from the regression based analyses. d' is not synonymous with recognition accuracy – please correct. Also, did the authors examine the results from an analysis of retrieval success? This is often associated with hippocampal activity in univariate analyses with appropriate materials and may be worth revisiting the null result.

“Analysis of rCBF revealed a non-significant trend towards a positive relationship with salivary E2 in the right amygdala [peak at (16, -4, -20) in Montreal Neurological Institute (MNI) space; $Z=2.85$, $p_{corr}=.067$; Supplementary Figure 3].”

(c) Again, the relationship should be described simply as, non-significant. If the authors wish to discuss any considerations related to statistical power, then this can either be formalised in the results section and elaborated in a discussion, or the non-significant results should be accepted on face value against the alpha criterion to avoid a labile notion of the alpha criterion.

“To rule out the possibility that differences in rCBF might have biased the relationship observed between E2 increase and memory task-related blood-oxygen level-dependent (BOLD) effects (fMRI analyses detailed below), more sensitive volume of interest analyses were performed from signal extracted from a 5-mm sphere around relevant activation peaks.”

(d) (i) It is unclear, at least as written, how this analysis addresses this concern, or, for that

matter, what the origins of the bias may have been – it shouldn't be necessary to read the SI to understand how this was addressed. Please rewrite to include a brief rationale in the main text; and, (ii) state coordinates of these spheres.

Discussion

"The fact that we did not observe any effect of E2 on hippocampus-dependent memory performance in this controlled study suggests that although effects on neural activity were visible after 24 hours, this time span was too short to also manifest in behavioral changes."

(a) How was modulation in hippocampal activity (re-)assessed at 24 hrs? It is unclear what the authors mean to say here. Also, presumably the proxy of neural plasticity – neural activity – is argued here – quite post-hoc – for the first time to be too short to manifest/i.e., support the hypothesized changes in recognition memory. I think it is premature and confusing to introduce this interpretation at this juncture, especially because the authors refer to careful planning and design, and then add a post-hoc interpretation at the start of the discussion without prefacing the reasoning behind this interpretation. Please revise accordingly.

Methods

"Based on arousal ratings from a previous study²⁷, we created 90 negative and 90 neutral picture pairs matched for level of arousal."

(a) What were the sizes of these images and what was the visual angle subtended by these stimuli inside the scanner vs the offline behavioural testing environment?

(b) How was presentation of visual stimuli equated between the scanner and offline environment? Please state in the manuscript.

Minor

(a) "It also has been reported to influence the shift from hippocampus-independent to -dependent learning strategies" – it is not self-evident what this sentence is referring to without further exposition.

(b) "Experimental groups did not differ significantly with respect to weight, body-mass index, age, education or parity [all p 's $>.1$; Supplementary Table 1]" – what is the referent of parity in this sentence? Unclear.

(c) "Suggesting that confidence ratings were diagnostic of recognition memory.." – rephrase to improve clarity and indicate that this was, presumably, a two-tailed analysis, despite priors: i.e., In order to assess whether or not confidence ratings were diagnostic ...

(d) SI "Please respond as correct and fast as possible." – this instruction should probably read as per convention, "Please respond as accurately and quickly as possible," if this was

as intended in the original language of the verbatim instructions.

(d) Supplementary Tables 1 and 12 - How were "all analyses" 'simply' based on $N=125$, when there were unequal group sizes - for example, ranging from $n=29$ to $n=20$ or $n=12$ to $n=23$ - particularly with analyses involving ANOVA. It's not apparent how this was handled in each analysis. Please include an explanation and dovetail with considerations raised about group variance and sphericity (below).

(e) Supplementary Figure 1 - It is somewhat confusing to use a greyscale key as an analogue of the colour brightness used to indicate day within each group colour. Please revise and the colour coding for the groups should also be depicted on the graphs.

Also, it's fairly patent that the smallest group - 4 mg ($n=20$) - exhibits wide variability on day 2 for cortisol. In general, the SEM error bars also suggest that homogeneity of variance needs to be tested for progesterone and cortisol, and, by extension estradiol, and the results of this analysis reported. Likewise, please also report and test for sphericity, and adjust p-values if necessary.

(f) Supplementary Table 4 - not sure what is mean by "range" in this context? Please clarify.

(g) Supplementary Tables 5 + 7 - typo in figure legend: "except of analyses"

Preface to all Reviewers

We would like to thank the reviewers again for spending the time and effort in providing constructive comments to our work yet another time. We believe that the suggested edits again improved the quality of the manuscript.

Below in blue are our detailed responses to specific comments. Passages from the main manuscript and the Supplementary Material are in italics. Changes in the manuscript itself are marked in red.

Response to Reviewer #2

The manuscript has improved but there are still have some remaining mostly minor concerns.

Comment 1: The authors claim their doses are in the supraphysiological range but is this really correct? Levels of estradiol are very high during pregnancy so it would seem these levels are indicative of lower levels during the first trimester. I would reword this to indicate that these are high levels, and higher than during menstrual cycle but not during pregnancy. Also change on line 307

Response: We thank the reviewer for directing our attention to the potentially misleading wording. The reviewer is right that we did not intend to imply that our doses resulted in E2 levels higher than during pregnancy by using the term ‘supraphysiological’ E2 levels. This term was chosen based on studies from animal research, such as the study of Scharfman and colleagues (2007) who also used the term supraphysiological for E2 levels that exceeded peaks during the estrous cycle but were still within the range during pregnancy.

Replacing the ‘supraphysiological’ by ‘high E2 levels’ could also be ambiguous, because it does not differentiate between high E2 levels reached regularly within every menstrual cycle and E2 levels above menstrual cycle peaks that are reached only during pregnancy, i.e. relatively seldom in a woman’s life. The more specific term ‘supra-menstrual-cycle levels’ appeared to us too complicated a term to use. To enhance clarity, we therefore suggest explicitly defining the term ‘supraphysiological’ when it first occurs in each section of the manuscript:

Abstract:

*“We pharmacologically increased the E2 levels in 125 naturally cycling women while in their low-hormone menstruation phase, to physiological (equivalent to menstrual cycle peak) and **supraphysiological (equivalent to levels during early pregnancy) concentrations.**”*

Introduction:

*“Only **when supraphysiological E2 levels (i.e. levels above estrous cycle peaks)** are reached does the dose-response relationship exhibit an inverted U-shape^{12-14,17}”*

*“This E2 treatment regimen induced a wide range of E2 levels, from **physiological (within 2- and 4-mg groups; equivalent to cycle peak)** to **supraphysiological (within 6- and 12-mg groups; equivalent to early pregnancy)** on the second day.”*

Discussion:

*“In particular, we tested the hypothesis whether neural activity monotonically increases at **physiological levels (equivalent to menstrual cycle peak)** and*

becomes an inverted U-shape only at supraphysiological E2 levels (equivalent to early pregnancy)."

Scharfman, H. E., Hintz, T. M., Gomez, J., Stormes, K. A., Barouk, S., Malthankar-Phatak, G. H., ... MacLusky, N. J. (2007). Changes in hippocampal function of ovariectomized rats after sequential low doses of estradiol to simulate the preovulatory estrogen surge. *European Journal of Neuroscience*, 26(9), 2595–2612. <https://doi.org/10.1111/j.1460-9568.2007.05848.x>

Comment 2: Line 87 – Although other studies may not have looked at exogenous E2, there are a number of studies examining the influence of menstrual cycle fluctuations in women – notably by the seminal studies of Elizabeth Hampson. Furthermore, there are a number of studies examining doses of E2 in postmenopausal women that could be at least referred to or one dose in younger women (Bartholomeusz CF).

Response: We agree and added references to Hampson and Morley (2013), Bartholomeuzs and colleagues (2008), a review by Galea, Frick, Hampson et al (2017) and a meta-analysis by Hogervorst and colleagues (2010):

"In humans, hormonal effects on hippocampal and amygdala activity, as well as macroscopic volumes, have been detected using magnetic resonance imaging (MRI), and some studies evidence behavioral effects for memory performance and affective processing^{3,25–31}. However, most of these studies have focused on the consequences of natural fluctuations in hormone levels across the menstrual cycle, or induced by pharmacotherapies, e.g. hormone replacement therapy (HRT), in order to better understand these specific conditions^{25,27, 32}."

3. Galea, L. A. M., Frick, K. M., Hampson, E., Sohrabji, F. & Choleris, E. Why estrogens matter for behavior and brain health. *Neurosci. Biobehav. Rev.* 76, 363–379 (2017).

31. Bartholomeusz, C. et al. Estradiol treatment and its interaction with the cholinergic system: Effects on cognitive function in healthy young women. *Hormones and Behavior* 54, 684–693 (2008).

32. Hampson, E. & Morley, E. E. Estradiol concentrations and working memory performance in women of reproductive age. *Psychoneuroendocrinology* 38, 2897–2904 (2013).

33. Hogervorst, E. & Bandelow, S. Sex steroids to maintain cognitive function in women after the menopause: A meta-analysis of treatment trials. *Maturitas* 66, 56–71 (2010).

Comment 3: in references 3,4,14, 50-53 – are these experiments done in males or females – given that this work is in women work cited should be in females. Especially for reference 4 – most of the data with brain estrogens has been obtained in males or in embryonic tissue. Some studies examining adult female levels show very little brain estrogens (Kato, Barker).

Response: We would like to thank the reviewer for directing our attention to our carelessness. Indeed, some of E2 effects are sex dependent. We removed references to Tanaka & Sokabe (2013) from the manuscript, as they studied juvenile male rats.

With respect to brain estrogens, the review of Mukai and colleagues (ref 4) summarizes findings on the concentrations and effects of hippocampus-derived E2 in intact and ovx/castrated female and male rodents. Although the authors found the concentration of E2 to be lower in the female (0.5–2 nM) than the male hippocampus (8 nM), both levels exceeded the levels of circulating E2. Kato and Kawato (2013) investigated hippocampal

E2 levels and spine density across the estrous cycle of adult female rats. They confirmed lower E2 levels in the female compared to male hippocampus, but they also showed 10-60 fold higher E2 levels in the female hippocampus compared to E2 in circulation. The authors also confirmed the cyclic fluctuation of spine density already shown by Woolley and colleagues (1990). A study by Barker & Galea (2009) found high concentrations of E2 in the hippocampus of intact male and female rats but did not find any sex difference in hippocampal E2 levels for intact rats. Gonadectomy significantly reduced hippocampal E2 levels in both sexes, which is not surprising because hippocampal E2 production has been shown to be coupled with gonadal E2 production via GnRH (Prange-Kiel et al., 2009). Barker & Galea also showed that systemic E2 treatment raised hippocampal E2 levels in gonadectomized female and male rats.

The review by Bean et al. (2014; ref 3) discusses findings related to E2 receptor distributions and E2 dose-response curves, mostly stemming from studies conducted in female rodents (please also see our response to Comment 7).

References 50 to 53 (Fester et al., 2013; Foster et al., 2012; Szymczak et al., 2006 and Zhou et al., 2014) provide evidence for the opposing impact of E2 receptor subtype activation/expression on hippocampal synaptogenesis. These findings are mostly based on hippocampal slice cultures. Nevertheless, Szymczak and colleagues (2006) observed changes in ER-beta expression across the natural estrous cycle which varied inversely to synaptogenesis. Likewise, Zhou and colleagues showed increased spine synapse densities in female ER-beta-knockout mice.

Barker, J. M., & Galea, L. A. M. (2009). Sex and regional differences in estradiol content in the prefrontal cortex, amygdala and hippocampus of adult male and female rats. *General and Comparative Endocrinology*, 164(1), 77–84. <https://doi.org/10.1016/j.ygcen.2009.05.008>

Fester, L., Labitzke, J., Hinz, R., Behem, C., Horling, K., Bernhard, T., ... Rune, G. M. (2013). Estradiol responsiveness of synaptopodin in hippocampal neurons is mediated by estrogen receptor beta. *Journal of Steroid Biochemistry and Molecular Biology*, 138, 455–461.

<https://doi.org/10.1016/j.jsbmb.2013.09.004>

Foster, T. C. (2012). Role of Estrogen Receptor Alpha and Beta Expression and Signaling on Cognitive Function During Aging. *Hippocampus*, 22(4), 656–669. <https://doi.org/10.1002/hipo.20935>

Kato, A., & Kawato, S. (2013). Female hippocampal estrogens have a significant correlation with cyclic fluctuation of hippocampal spines. *Frontiers in Neural Circuits*, 7, 149.

<https://doi.org/10.3389/fncir.2013.00149>

Prange-Kiel, J., Fester, L., Zhou, L., Jarry, H., & Rune, G. M. (2009). Estrus cyclicity of spinogenesis: underlying mechanisms. *Journal of Neural Transmission*, 116(11), 1417–1425.

<https://doi.org/10.1007/s00702-009-0294-x>

Szymczak, S., Kalita, K., Jaworski, J., Mioduszevska, B., Savonenko, A., Markowska, A., ... Kaczmarek, L. (2006). Increased estrogen receptor beta expression correlates with decreased spine formation in the rat hippocampus. *Hippocampus*, 16(5), 453–463. <https://doi.org/10.1002/hipo.20172>

Woolley, C. S., Gould, E., Frankfurt, M., & McEwen, B. (1990). Naturally occurring fluctuation in dendritic spine density on adult hippocampal pyramidal neurons. *The Journal of Neuroscience*, 10(12), 4035–4039.

Zhou, L., Fester, L., Haghshenas, S., de Vrese, X., von Hacht, R., Gloger, S., ... Rune, G. M. (2014). Oestradiol-Induced Synapse Formation in the Female Hippocampus: Roles of Oestrogen Receptor Subtypes. *Journal of Neuroendocrinology*, 26(7), 439–447. <https://doi.org/10.1111/jne.12162>

Comment 4: Lines 349-353 – While it is tempting to imagine that more spines is equivalent to better memory has this been definitively shown? Spines may not equal memory and if my memory serves LTP is not related to MORE spines but larger spines, a

subtle but important distance and it's better to not equate number of spines to memory.

Response: We absolutely agree with the reviewer that a higher number (and/or size) of spines is not equivalent to better memory *per se* and that both number and size of spines are related to LTPs (Collin et al., 1997; Engert & Bonhoeffer, 1999; Segal et al; 2005). Likewise, one would expect both, more numerous and larger spines, to result in an increase in BOLD effect. In response to the reviewer's comment we also mention the size of spines as an alternative mechanism behind the effects of E2 on the BOLD effect:

*“Here, the 24 hours of heightened E2 levels in the current E2 regimen might have been long enough to **increase synapse density or the size of existing spines** and thereby increasing glutamate release, as reflected by the fMRI signal, but too short for the new synapses to become functionally effective.”*

Collin, C., Miyaguchi, K., & Segal, M. (1997). Dendritic spine density and LTP induction in cultured hippocampal slices. *Journal of neurophysiology*, 77(3), 1614–1623.

Engert, F., & Bonhoeffer, T. (1999). Dendritic spine changes associated with hippocampal long-term synaptic plasticity. *Nature*, 399(6731), 66–70.

Segal, M. (2005). Dendritic spines and long-term plasticity. *Nature Reviews Neuroscience*, 6(4), 277–284. <https://doi.org/10.1038/nrn1649>

Comment 5: The discussion is very long and could benefit from some trimming (perhaps parts of the long discussion of spines).

Response: We agree and shortened the discussion.

Comment 6: lines 400-402 – the different findings are also related to different doses of E2 – EVEN WHEN IN THE physiological range. High E2 is associated with poorer memory while diestrous levels are associated with better spatial memory (very much akin to what E Hampson demonstrated in women across menstrual cycles). Other discrepancies lie within the idea that E2 is given AFTER or DURING the learning. Frick gives E2 during consolidation whereas other given it during the learning – so outcome differences may be due to these differences as well. Given that your data is based on during learning it is perhaps better to concentrate on those studies.

Response: We agree that dose, paradigm, and treatment schedules have been shown to be important moderators of E2 effects. For this reason, we added citations to the review articles (Frick et al., 2015 and Luine, 2014), which both discuss the effects of E2 from studies varying in dose, paradigms and treatment schedule systematically.

In response to the Reviewers comment, we revised the sentence as follows:

*“However, despite the many studies providing evidence for a positive relationship between hippocampus-dependent memory and E2 levels **during learning** within a physiological range in female animals^{1,2}, even these effects appear to **depend highly on dose, treatment duration and, in particular, the specific task used**^{21-24,59}.”*

We also confirmed that all of the original cited papers (Aydin et al., 2008; Fader et al., 1999; Rissman et al., 2002; Holmes et al., 2002; Vierk et al., 2015) manipulated E2 levels during learning.

Comment 7: Some references are reviews and not primary papers. More papers are needed that have actually investigated dose effects in rodents. (Barha CK)

Response: We thank the reviewer for directing our attention to the article by Barha and colleagues (2009), and we have now cited it here:

“However, a few studies have reported monotonic improvements also at clearly supraphysiological E2 levels, and some effects only occur at supraphysiological levels^{12,18-20}.”

We had to replace citations to some original papers by review articles in an attempt to fulfil the journal's reference and word limits. We focused on citing as many original papers investigating dose-dependent effects of E2 on neural plasticity as possible, in order to motivate our main research hypotheses. With respect to E2 dose-dependent behavioral effects, the heterogeneity of treatment regimens and paradigms seems too high to derive specific conclusions with respect to our specific experimental design and would require much more elaboration than the reference and word limits would allow. We agree with the reviewer that systemizing these studies and their implication for human studies would be an important step for the whole field and should be addressed in the future.

14. Barha, C. K., Lieblich, S. E., & Galea, L. a. M. (2009). Different Forms of Oestrogen Rapidly Upregulate Cell Proliferation in the Dentate Gyrus of Adult Female Rats. *Journal of Neuroendocrinology*, 21(3), 155–166. <https://doi.org/10.1111/j.1365-2826.2008.01809.x>

Minor

Comment 8: Line 53 – citation needed

Response: We added a reference to the sentence:

“That is, one would expect increasing benefits until medium levels of E2 but then a decrease and none or even negative effects at high levels⁴.”

4 Bean, L. A., Ianov, L. & Foster, T. C. Estrogen Receptors, the Hippocampus, and Memory. *Neuroscientist* 20, 534–545 (2014).

Comment 9: Line 546 – something wrong here.

Response: We apologize for this oversight and corrected the figure reference.

Response to Reviewer #3

Comment 1: I would suggest that the author re-write, trim and better focus the manuscript prior to my being able to recommend publication.

Response: We certainly agree with the reviewer and substantially rewrote and trimmed the introduction and discussion.

Response to Reviewer #4

Comment 1: Page 12, line 8: Please specify „relevant activation peaks“ and provide analysis in supplementary material, if necessary.

Response: We apologize for the lack of clarity. The term ‘relevant activation peaks’ was intended to refer to significant effects in subsequent fMRI analyses. We rephrased the sentence:

*“To rule out the possibility that differences in rCBF might have biased the relationship observed between E2 increase and memory task-related blood-oxygen level-dependent (BOLD) effects (fMRI analyses detailed below), more sensitive volume of interest analyses were performed from signal extracted from a 5-mm sphere around all peaks **showing significant relationships with salivary E2 increase in the fMRI analyses (see below).**”*

Comment 2: Figure7: Although the authors used robust regression, the inverted U-shaped relationship in the insula (as displayed in Figure 7) might be driven by the single data point in the lower right corner. Have the authors run the analysis without that data point?

Response: When the respective data point is removed, robust regression analysis still evidences a significant quadratic relationship between EEM-related activity in the insula and salivary Day 1 to Day E2 increase [$t(118)=2.40$, $p=.018$].

Comment 3: Page 36, line 17: It doesn’t become quite clear to my why the authors created the ROIs of interest. It seems that they wanted to use them for a small volume correction, but they say at the same time that they use the FWE-correction of the entire scan volume. If that is already significant, a small volume correction does add something. Therefore, I’m wondering whether the reported p(corr) values refer to the scan volume or to the ROIs volume. Further, one shouldn’t use more than one ROI for a small volume correction procedure, i.e. if one expect effects to appear in the amygdala AND hippocampus, one should create an ROI that cover both and not one for each area separately. Otherwise, one would need a Bonferroni correction (or similar) on the number of ROIs, as well.

Response: We apologize for this lack in clarity. Small volume correction was only applied to regions of interest defined a priori (i.e. the hippocampus and the amygdala). P-values in all other brain regions were corrected for multiple comparisons with respect to the entire scan volume.

Based on the reviewer’s comment we rephrased the relevant sentence as follows:

*“Results of all analyses, i.e. the main effects as well as the models testing for E2 relationships, were considered significant at $p < .05$, family-wise error corrected for multiple comparisons within anatomical regions of interests (ROIs) **defined a priori, namely the hippocampus and the amygdala. In addition, results outside of these ROIs were considered significant at $p < .05$, family-wise error corrected for***

multiple comparisons on the entire scan volume. The hippocampus and amygdala masks were created using the Anatomy toolbox^{97,98}.

Moreover, we followed the reviewer's suggestion and created a ROI that covered the hippocampus as well as the amygdala. Using the combined ROI for FWE correction did not change the significance level of the linear and quadratic relationships between the BOLD effect and salivary E2 increase in the hippocampus [linear: $Z=3.69$, $p_{\text{corr}}=.043$; quadratic: $Z=4.22$, $p_{\text{corr}} = .006$] but diminished the statistical trend towards a linear positive relationship between rCBF and salivary E2 in the right amygdala [$Z=2.85$, $p_{\text{corr}}=.158$]. We revised all relevant statistics and figures.

Response to Reviewer #5

Comment 1: Appealing to prior conventions is not entirely appropriate here, particularly with the current debate about reproducibility of results and more recent discussions about the selection of appropriate thresholds. The Liberman & Cunningham (2009) paper coincides with an era where many of the main fMRI results in papers were reported as uncorrected, or, if corrected for multiple comparisons, were not based on a computation of, for example, FDR (Chumbley & Friston, 2009, Chumbley et al. 2010). In addition, the approach still lacks objectivity – all inferences about the neuroimaging data in the manuscript are drawn by applying a FWE-corrected $p < 0.05$ threshold, whereas the figures that refer and claim to depict same results do not correspond to the same thresholding. The figures should depict the appropriately thresholded results first and be elaborated with uncorrected $p < 0.005$ if desired. Therefore, please revise and add figures at the appropriately corrected threshold $p < 0.05$.

Response: We followed the reviewer's suggestion and created figures depicting a FWE-corrected threshold of $p < .05$ using a cold color scale and a threshold of $p < .005$ using a warm color scale.

Comment 2: It is entirely reasonable to include reports of non-significant results, for the reasons outlined above. Non-significant results should be reported – that's not my main concern.

What remains at issue is the manner in which non-significant results are being described. The most neutral way to report a non-significant result is to state simply, that it was non-significant – the result is thereby reported for the record and the reader left to adjudicate on Type II considerations, within the statistical power of the design to detect intended and unintended effects.

Repeatedly stating results indicate a “non-significant trend/tendencies” is misleading and introduces a systematic bias in interpretation because (1) it is suggestive of Type II errors. Increased statistical power, for instance, may just as well push the result even further above the alpha criterion; and, (2) as mentioned previously, a significant p-value at the margins, such as $p = 0.03$, needs by logical extension to be reciprocally referred to either as a “significant trend/tendencies or by extension “trending towards non-significance”, especially when considered against the recent push towards a more conservative alpha criterion ($p = 0.0005$) in the field (Benjamin et al., PsyArxiv, 10.17605/OSF.IO/MKY9J).

All that said, the potentially misleading nature of reporting trends can easily be avoided by stating simply that the results were non-significant when $p > 0.05$ and significant when $p < 0.05$. So, please just simply remove reference to the words trends/tendencies and leave the reporting of non-significant finding otherwise intact. In addition, further discussion for each of the outlined cases above can be introduced if the authors desire to speculate on why a particular non-significant result warrants further attention and might be a Type II error.

Response: We agree with the reviewer and have accordingly removed the term

trend/tendencies and state clearly that these results were not significant.

Comment 3: The specific memory task and nature of the dependent measures, rather than “their memory”, should be stated in the abstract. Presumably, the authors are again referring to encoding related activity when going onto refer to “memory-related activity” – this needs to be restated to improve precision in the manner the main effect is being described in order to minimize the potential for confusion in a reader.

Response: We agree that the wording was imprecise and have revised the abstract as follows:

*“Twenty-four hours after first E2 intake, we measured brain activity during encoding of neutral and negative pictures and then tested recognition memory 24 hours after encoding. Hippocampal activity during encoding was strongly associated with **subsequent recollection**, where E2 elicits not only a monotonically increasing but also an inverted U-shaped response depending on posterior hippocampal region.”*

*“E2 did not affect **recognition memory performance**, suggesting that E2’s effect on neural activity during encoding was **not effective in improving recognition memory after a consolidation interval of 24 hours in the current paradigm.**”*

Comment 4a: One of the main conclusions is an inference based on null effect. There are several concerns here (1) the null effect refers to the relevance of E2 on encoding-related hippocampal activity for later recognition memory. It is not apparent from the current summary that this is the case;

Response: We agree with the reviewer that the imprecise wording was potentially misleading. We now state more clearly in the abstract that we contrasted activity during successful encoding. We revised the abstract as stated in the response to Comment 3.

Comment 4b: (2) it’s confusing how this summary refers to the “E2 effect on neural activity after 24 hrs” – only behaviour was measured at 24 hrs; reference can only be made to behaviour and not neural activity/experience-dependent plasticity;

Response: We apologize for the confusion caused by the imprecise language. The E2 effect on neural activity after 24 hrs referred to the E2 effect 24 hrs after the first E2 intake. The reviewer is right that we are not able to draw any conclusions about neural E2 effects on Day 3 (i.e. 48 hrs after initial E2 intake, or 24 hrs after encoding). We revised the relevant sentences (see response to Comment 4a).

Comment 4c: (3) there needs to be some indication in the abstract why encoding and recognition-based retrieval of neutral and emotional stimuli should be considered a de facto measure of hippocampal activity given the outcomes;

Response: We have added a sentence about how successful encoding was related to robust main effects in the hippocampus and the amygdala in the current study.

“Hippocampal activity during encoding was strongly associated with subsequent

recollection, (...).”

Comment 4d: (4) consideration in the abstract should also be added to reflect that the null result could reflect a suboptimal measure of “memory performance”, suboptimal interval post-initial encoding to assay the effects on memory performance, or effects of E2 on hippocampal activity that are associated with other operations such as perceptual and/or attentional processing.

Response: We agree that these alternative explanations for the observed behavioral null effect are relevant and should be ideally mentioned in the abstract. Because we are already over the word limit for the abstract, we suggest explicitly acknowledging in the abstract that all of our conclusions apply exclusively to the employed memory paradigm and consolidation interval (see responses to Comments 4a and 4c). Additionally, we discuss these important alternative explanations in more detail in the discussion. We hope these changes address the concerns of the reviewer.

Discussion:

“An alternative account for the absence of an effect on memory performance could be that the hippocampal activity that is modulated by E2 does not reflect memory formation but hippocampus-guided visual exploration⁶⁹.”

“Finally, current results should not be overgeneralized as evidence against an effect of E2 on memory in humans, as the behavioral null effect could, for instance, be specific to the employed paradigm which differs from paradigms used in animal research.”

“It therefore seems plausible that a different task with the current E2 regimen could elicit observable behavioral changes. “

“The fact that we did not observe any effect of E2 on hippocampus-dependent memory performance in this rigorously controlled study suggests that although effects on neural activity during encoding were visible 24 hours after initial E2 intake, this time span was too short to also manifest in behavioral changes after a consolidation interval of 24 hours in the current paradigm.”

Comment 5: Revision of the abstract is important because the implication, as current written, is that E2 has no impact on all memory performance, but this was entirely conditional on (a) the use of a single assay, recognition memory, and, (b) assessing the effects of E2 at 24 hr post-scanning and using fMRI as a proxy for neural plasticity only at encoding. By the authors own estimation in the discussion, prior effects can be task-specific, so it is somewhat sweeping to infer with the current protocol that the use of a single recognition test stands as a de facto and sufficient measure with which judge the effects of E2 on human memory-guided behaviour. Recollection based recognition does not always engage the hippocampus; inclusion of an additional test such as cued-recall would could have provided a more compelling foundation on which to investigate the wider effects on memory-guided behaviour. A version of these considerations need to be communicated in the abstract, especially because the results are at variance with

evidence of hormonal effects on memory and affective processing. As it stands currently, the conclusion over reaches in this regard.

Response: We fully agree with the reviewer that the abstract should clearly state that we can only draw conclusions about E2's effects on memory with respect to the employed memory paradigm and E2 treatment regimen. (However, it might be of interest that we did not see a significant influence of E2 deprivation on verbal free recall in postmenopausal women (Bayer et al., 2015)). We therefore revised the abstract (see excerpts in our responses to comments 3-4). We also agree that recollection is not always hippocampus-dependent. We therefore explicitly refer to robust recollection-evoked hippocampal activity in the current paradigm in the abstract now (see response to Comment 4c).

We hope to have succeeded in addressing the reviewer's concerns while keeping within the word limits.

Bayer, J., Rune, G., Schultz, H., Tobia, M. J., Mebes, I., Katzler, O., & Sommer, T. (2015). The effect of estrogen synthesis inhibition on hippocampal memory. *Psychoneuroendocrinology*, 56, 213–225. <https://doi.org/10.1016/j.psyneuen.2015.03.011>

Comment 6: A brief but more specific consideration regarding how neural plasticity and behaviour were quantified in prior studies involving model organisms could provide further mechanistic links between these variables, especially given that the current study does not elucidate a link between BOLD activity – as a proxy of neural plasticity – and a single assay of memory-guided behaviour. It is not unreasonable to conclude that some of the variability in prior studies may be epiphenomena of the research protocols rather than E2 effects on neuroplasticity and by extension on memory-guided behaviour, per se. Most usefully, this summary could draw specific links between behavioural protocols used in model organisms and rationale for the behavioural protocol in the current study, especially because the current study already uses a very different proxy/operational definition - from the studies of model organisms – of neural plasticity – and finds a null link with behaviour.

Behavioural tasks between model organisms and humans are frequently equated to minimise the obvious sources of noise/confounds.

Response: We fully agree that the operationalization of neural plasticity in the current study is different from animal studies. We respond to this comment in detail below.

With respect to the chosen paradigm, the reviewer raises an intriguing point here which we considered deeply before proceeding with our selected design choice. We decided on an emotional memory task in the end rather than attempting to more directly translate a task from animal studies for a variety of reasons.

First, and most importantly, the task we chose allows us to investigate activity related to hippocampus-dependent memory formation and amygdala-dependent emotional processing, as well as emotional memory formation which is based on the interaction of both. The effects of estrogen on amygdala and amygdala-related affective processing has gained increasing interest in recent years (Toffoletto et al., 2014). Moreover, the task we chose relies on a large network of cortical and subcortical areas that express estrogen receptors but that have not yet been addressed by animal studies (Spaniol et al., 2009; Spalek et al 2015, Kritzer et al., 2002). By acquiring whole brain fMRI volumes, this task allowed us to also therefore explore estrogen effects in other brain regions, which is a

great advantage for such an extensive, large-scale project.

Second, we already used a similar emotional memory task before and observed an influence of menstrual-cycle-dependent hormone fluctuations, not only on hippocampal activity, but also on behavior (Bayer et al., 2014).

Third, in animal studies, a wide variety of spatial but also non-spatial memory tasks have been employed (Luine, 2015). Although we certainly agree that it would be desirable to more thoroughly review these approaches in order to arrive at an informed decision on what task would best translate to humans, this is much less straightforward due to the large variability in the combinations of paradigms and E2 regimens employed. Moreover, most animal paradigms, such as the widely used water maze task, are very difficult to translate into an MR-scanner compatible paradigm without substantial modifications that would probably substantially compromise comparability.

However, we certainly agree that it is important to clearly state that the null effect is limited to the current paradigm which differs in many aspects to the paradigms used in animal research. We also agree, that the variability in existing literature can be likely explained by experimental variables. We now give samples for paradigms used in animal research in the introduction and revised the manuscript as follows:

Discussion:

*“However, despite the many studies providing evidence for a positive relationship between hippocampus-dependent memory and E2 levels **during learning** within a physiological range in female animals^{1,2}, even these effects appear to **depend highly on dose, treatment duration and, in particular, the specific task used**”^{21–24,59} .*

“Finally, current results should not be overgeneralized as evidence against an effect of E2 on memory in humans, as the behavioral null effect could, for instance, be specific to the employed paradigm which differs from paradigms used in animal research.”

Bayer, J., Schultz, H., Gamer, M., & Sommer, T. (2014). Menstrual-cycle dependent fluctuations in ovarian hormones affect emotional memory. *Neurobiology of Learning and Memory*, 110, 55–63. <https://doi.org/10.1016/j.nlm.2014.01.017>

Kritzer, M. F. (2002). Regional, Laminar, and Cellular Distribution of Immunoreactivity for ER α and ER β in the Cerebral Cortex of Hormonally Intact, Adult Male and Female Rats. *Cerebral Cortex*, 12(2), 116–128. <https://doi.org/10.1093/cercor/12.2.116>

Luine, V. (2015). Recognition memory tasks in neuroendocrine research. *Behavioural Brain Research*, 285, 158–164. <https://doi.org/10.1016/j.bbr.2014.04.032>

Spalek, K., Fastenrath, M., Ackermann, S., Auschra, B., Coynel, D., Frey, J., ... Milnik, A. (2015). Sex-Dependent Dissociation between Emotional Appraisal and Memory: A Large-Scale Behavioral and fMRI Study. *Journal of Neuroscience*, 35(3), 920–935. <https://doi.org/10.1523/JNEUROSCI.2384-14.2015>

Spaniol, J., Davidson, P. S. R., Kim, A. S. N., Han, H., Moscovitch, M., & Grady, C. L. (2009). Event-related fMRI studies of episodic encoding and retrieval: Meta-analyses using activation likelihood estimation. *Neuropsychologia*, 47(8–9), 1765–1779. <https://doi.org/10.1016/j.neuropsychologia.2009.02.028>

Toffoletto, S., Lanzemberger, R., Gingnell, M., Sundström-Poromaa, I., & Comasco, E. (2014). Emotional and cognitive functional imaging of estrogen and progesterone effects in the female human brain: a systematic review. *Psychoneuroendocrinology*, 50, 28–52.

Comment 7: Repeated reference is made to “neural plasticity” in model organisms and then changed in humans to a discussion of neural activity “The effects of E2 on neural activity and behavior”; e.g., “.. dose-response relationship between E2 and neural plasticity as well as behavior..”; “Very few studies have investigated the effect of different E2 doses on neural plasticity...”;

In a journal with a specialist and non-specialist readership, the use of the term neural plasticity needs to be applied with greater precision, explaining how neural plasticity is measured / conceptualized in each case; at best, the study design can indirectly examine experience-dependent change as a result of neural plasticity measured by BOLD activity. Please review the relevant text to improve the precision with which the terms is applied.

Response: We agree that the term neural plasticity should be applied with greater care, especially when referring to such different assessment methods of cellular markers of neural plasticity as used in animal models and experience-dependent changes as assessed by fMRI. We therefore added specific examples how neural plasticity is assessed and clarify whether neural plasticity is assessed on the cellular level or whether proxies such as fMRI were used:

Introduction

*“On the cellular level, E2 affects hippocampal and amygdalar neural plasticity through synaptogenesis and glutamatergic neurotransmission leading to, for example, increases in **neuronal spine density or long-term potentiation (LTP) magnitude** ^{1,4,5,10,11} .”*

*“E2 and **cellular measures of** eural plasticity as well as behavior might not always follow the traditional sigmoidal but often an inverted U-shaped function⁶.”*

*“The inconsistency in reported responses might be resolved at least for the hippocampus with the observation that **cellular measures of** neural plasticity monotonically increase with E2 levels mostly at physiological E2 concentrations¹²⁻¹⁷ .”*

*“In humans, hormonal effects on hippocampal and amygdala activity, as well as macroscopic volumes, have been **detected using magnetic resonance imaging (MRI)**, and some studies evidence behavioral effects for memory performance and affective processing^{3,25-32} .”*

*“This wide range enabled us to test our hypothesis that the dose-response function between E2 and neural activity **assessed by fMRI (as a proxy for changes in neural plasticity)** increases monotonically within physiological ranges and turns into an inverted U-shape only at supraphysiological levels. ”*

Discussion

*“**We** characterized dose-response functions between E2 and episodic memory, affective processing **and** related neural activity **assessed by fMRI** in humans.”*

*“This is however consistent with the animals studies mentioned in the Introduction, which have observed monotonic increases in **cellular measures** of hippocampal*

plasticity also at supraphysiological levels of E2^{12,18-20}. “

*“Many animal studies have observed that E2 affects not only for **cellular measures** of hippocampal neuronal plasticity but also hippocampus-dependent memory performance^{1,2}.”*

Comment 8: As per my earlier comment: this should read, was non-significant after Bonferroni correction – the p-value is non-significant at the Bonferroni corrected p value, it is not trending (to) non-significance. Also, I am not sure if it is entirely necessary to link the follow-up analysis of RT data to what is a non-significant component labelled subjective vigilance. The analysis of RT is helpful and relevant as a standalone subsection. I think the link diminishes the line of reasoning, but I leave it to the authors discretion if they wish to sever the link with the non-significant principal component.

Response: We revised the sentence as suggested. We understand the reviewer's remark but have left in the link between 'subjective vigilance' and RT analyses.

*“According to principal component analyses on data from the mood questionnaire, the main factor which explained most of self-reported mood variance with E2 correlated with alertness, strength, energy, clear-headedness (see Supplementary Table 13), **but this did not survive Bonferroni correction** [$t(121)=2.31$, $p_{corr}=.092$, $\beta_{std}=.188$].”*

Comment 9 a: It is helpful to state/summarise here in the main text whether recognition based measures were significantly different from chance in the first instance before then proceeding to report the results from the regression based analyses.

Response: One-sample t-tests against chance level (0.5) confirmed that hit rates [neutral: $t(122)=2.94$, $p=.004$; negative: $t(122)=10.17$, $p<.001$] as well as correct rejection rates [neutral: $t(122)=48.36$, $p<.001$; negative: $t(122)=30.92$, $p<.001$] were significantly above chance levels. We added this information to the results section and the corresponding statistics description in the Supplementary Material Section 1.3.

Comment 9 b: d' is not synonymous with recognition accuracy – please correct.

Response: We agree and have removed this term from the relevant paragraphs.

Comment 9 c: Also, did the authors examine the results from an analysis of retrieval success? This is often associated with hippocampal activity in univariate analyses with appropriate materials and may be worth revisiting the null result.

Response: We are unsure whether we understand what the reviewer means by 'retrieval success'. If the reviewer is referring to hit rates, please refer to Supplementary Table 3 where we report robust regression analyses between hit rates and salivary Day 1 to Day 2 E2 increase [all p_{corr} 's $>.336$].

Comment 10: Again, the relationship should be described simply as, non-significant. If

the authors wish to discuss any considerations related to statistical power, then this can either be formalised in the results section and elaborated in a discussion, or the non-significant results should be accepted on face value against the alpha criterion to avoid a labile notion of the alpha criterion.

Response: We have revised the relevant sentence.

Comment 11: (i) It is unclear, at least as written, how this analysis addresses this concern, or, for that matter, what the origins of the bias may have been – it shouldn't be necessary to read the SI to understand how this was addressed. Please rewrite to include a brief rationale in the main text; and, (ii) state coordinates of these spheres.

Response: We have included a brief rationale for ASL to the introduction and clarified the reference to the 'relevant activation peaks' in the results section:

Introduction

*“Regional cerebral blood flow (rCBF) was also measured via arterial spin labeling (ASL) at rest to assess task-independent effects of E2, which **could increase the baseline perfusion, thereby leading to smaller relative task-related BOLD effects**⁴².”*

Results

*“To rule out the possibility that differences in rCBF might have biased the relationship observed between E2 increase and memory task-related blood-oxygen level-dependent (BOLD) effects (fMRI analyses detailed below), more sensitive volume of interest analyses were performed from signal extracted from a 5-mm sphere around all peaks **showing significant relationships with salivary E2 increase in the fMRI analyses** (see below).”*

Comment 12: How was modulation in hippocampal activity (re-)assessed at 24 hrs? It is unclear what the authors mean to say here. Also, presumably the proxy of neural plasticity – neural activity – is argued here – quite post-hoc – for the first time to be too short to manifest/i.e., support the hypothesized changes in recognition memory. I think it is premature and confusing to introduce this interpretation at this juncture, especially because the authors refer to careful planning and design, and then add a post-hoc interpretation at the start of the discussion without prefacing the reasoning behind this interpretation. Please revise accordingly.

Response: We thank the reviewer for directing our attention to this ambiguous wording “after 24 h” (i.e. 24 hrs after initial E2 intake and 24 hrs after encoding). We revised the corresponding sentences. Examples:

Discussion

*“Here, the **24 hours of heightened E2** levels in the current E2 regimen might have been long enough to increase synapse density or the size of existing spines,*

thereby increasing glutamate release, as reflected by the fMRI signal, but too short for the new synapses to become functionally effective.”

“The fact that we did not observe any effect of E2 on hippocampus-dependent memory performance in this rigorously controlled study suggests that although effects on neural activity during encoding were visible **24 hours after first E2 intake**, this time span was too short to also manifest in behavioral changes after a **consolidation period of 24 hours** in the current paradigm.”

Comment 13: What were the sizes of these images and what was the visual angle subtended by these stimuli inside the scanner vs the offline behavioural testing environment?

Response: The images had a size of 480 x 720 pixels. The visual angle was 11° x 16° and kept equal between stimulus presentation inside (encoding) and outside (retrieval test, arousal rating) the scanner. We added this information to the methods section.

Comment 14: How was presentation of visual stimuli equated between the scanner and offline environment? Please state in the manuscript.

Response: Participants saw the pictures at the same visual angle inside and outside the scanner. The only difference was that the participants were lying supine in the MR scanner but sat upright outside the scanner.

Comment 15: “It also has been reported to influence the shift from hippocampus-independent to -dependent learning strategies” – it is not self-evident what this sentence is referring to without further exposition.

Response: We agree that this claim (added in response to Reviewer #2 during the first revision) is difficult to understand without prior knowledge. We have therefore added examples for hippocampus-independent and -dependent learning strategies:

*“Most evidence exists for E2 effects on hippocampal neurons and hippocampus-dependent learning and memory in rodents, including E2-dependent shifts from hippocampus-independent (e.g. **egocentric**) to -dependent (e.g. **allocentric**) learning strategies^{34,35}.”*

Comment 16: “Experimental groups did not differ significantly with respect to weight, body-mass index, age, education or parity [all p’s >.1; Supplementary Table 1]” – what is the referent of parity in this sentence? Unclear.

Response: We apologize for the unclear wording. The term was intended to refer to lifetime pregnancy (i.e. whether a women has ever been pregnant). We replaced the term ‘parity’ by ‘lifetime pregnancy’.

Comment 17: “Suggesting that confidence ratings were diagnostic of recognition memory..” – rephrase to improve clarity and indicate that this was, presumably, a two-

tailed analysis, despite priors: i.e., In order to assess whether or not confidence ratings were diagnostic

Response: We have rewritten the sentences as suggested by the reviewer.

Comment 18: SI “Please respond as correct and fast as possible.” – this instruction should probably read as per convention, “Please respond as accurately and quickly as possible,” if this was as intended in the original language of the verbatim instructions.

Response: Thank you for pointing that out. We have revised the sentence.

Comment 19: Supplementary Tables 1 and 12 - How were “all analyses” ‘simply’ based on N=125, when there were unequal group sizes – for example, ranging from n=29 to n=20 or n=12 to n=23 – particularly with analyses involving ANOVA. It’s not apparent how this was handled in each analysis. Please include an explanation and dovetail with considerations raised about group variance and sphericity (below).

Response: We apologize for the confusion. Except of group characteristics (age, weight, BMI, education, lifetime pregnancy), missing values and hormones levels, all variables were analyzed using regression analyses based on the whole sample (SI Table 3, 5,6,7 9, 10). The reason for the confusion is most likely, that we added sub-group analyses (i.e. ANOVA’s) to the last rebuttal which were not included in the manuscript or the Supplementary Material in the submission. We added tests of variance homogeneity for age, weight, BMI (SI Table 1) and hormone levels to the Supplementary Material Sections 1.1 and 1.2 and replaced the standard ANOVA’s with Welch’s Robust Test of Equality of Means when necessary. Education (SI Table 1) and missing values (SI Table 12) were analyzed using non-parametric Chi-Square Tests which are robust to unequal group sizes.

Comment 20: Supplementary Figure 1 – It is somewhat confusing to use a greyscale key as an analogue of the colour brightness used to indicate day within each group colour. Please revise and the colour coding for the groups should also be depicted on the graphs.

Response: We have revised the color key of Figure 2 (Main Manuscript) and Supplementary Figure 1, as suggested.

Comment 21: Also, it's fairly patent that the smallest group – 4 mg (n=20) – exhibits wide variability on day 2 for cortisol. In general, the SEM error bars also suggest that homogeneity of variance needs to be tested for progesterone and cortisol, and, by extension estradiol, and the results of this analysis reported. Likewise, please also report and test for sphericity, and adjust p-values if necessary.

Response: We thank the reviewer for directing our attention to the missing tests for variance homogeneity. We have added these analyses to the Supplementary Material and used Welch's Robust Tests for Equality of Means when necessary (see below). Because we only calculated univariate ANOVA's on the difference scores (e.g. E2 on Day 2 – E2 on Day 1), there were no within-subject conditions for which sphericity could be tested.

Supplementary Material

“Levene's test for homogeneity of variances indicated inhomogeneous variances across groups for Day 1 to Day 2 changes in salivary and serum E2 [saliva: $F(4,120)=10.66$, $p<.001$; serum: $F(4,120)=12.68$, $p<.001$]. There was no evidence for inhomogeneous variances for Day 1 to Day 2 changes for progesterone or cortisol [progesterone: $F(4,120)=1.66$, $p=.165$, cortisol: $F(4,120)=1.15$, $p=.339$].”

Caption of Figure 2:

“Confirming the effectiveness of the pharmacological manipulation, the increases in salivary and serum E2 levels from baseline (Day 1) to the expected peak (Day 2) differed as intended between experimental groups [saliva: $F(4, 47.98)=102.84$, $p <.001$; serum: $F(4, 31.24)= 87.54$, $p<.001$; see Supplementary Material Section 1.2 for tests of variance homogeneity].”

Comment 22: Supplementary Table 4 – not sure what is mean by “range” in this context? Please clarify.

Response: The term ‘range’ referred to the difference between the highest and the lowest value within each group. We added this explanation to the caption of Supplementary Table 4.

Comment 23: Supplementary Tables 5 + 7 – typo in figure legend: “except of analyses”

Response: We thank the Reviewer and corrected the typos.

Reviewers' comments:

Reviewer #2 (Remarks to the Author):

I am fine with the corrections as done..although personally I would not use the term supraphysiological (as this is meant to depict never seen in normal physiology) - but I am fine given there is now a definition for readers.

Reviewer #4 (Remarks to the Author):

The authors have sufficiently revised their manuscript and addressed all my concerns. I have no further comments.

Reviewer #5 (Remarks to the Author):

Overall, the manuscript has been improved by the latest revisions and reporting of additional analyses. However, the manuscript still warrants attention regarding the interpretation of the results and a consideration of additional potential caveats.

"Twenty-four hours after first E2 intake, we measured brain activity during encoding of neutral and negative pictures and then tested recognition memory 24 hours after encoding. Hippocampal activity during encoding was strongly associated with subsequent recollection, where E2 elicits not only a monotonically increasing but also an inverted U-shaped response depending on posterior hippocampal region."

"E2 did not affect recognition memory performance, suggesting that E2's effect on neural activity during encoding was not effective in improving recognition memory after a consolidation interval of 24 hours in the current paradigm."

(1) Reference to subsequent recollection – especially without any mention in the abstract about how remember and know responses have been mapped onto the theoretical quantities (based on a dual-process interpretation of recognition memory) of recollection and familiarity – is confusing. It is unclear how a reader would reconcile "hippocampal activity during encoding was strongly associated subsequent recollection" with the later statement in the abstract where the authors aver, "E2's effect on neural activity during encoding was not effective in improving recognition memory." Presumably, subsequent recollection simply refers to hippocampal activity that emerged from univariate contrast between remember>know responses; i.e., the subjective 'recollective' experience associated with retrieval on a recognition memory task that was not shown to be relevant for improving/changing recognition memory. This distinction remains important because it underscores the general absence of a link with the E2 levels, hippocampal activity for a contrast that was tied with changes in recognition memory performance, and later recognition memory, as examined using d' ; meta- d' ; successful encoding, subsequent hits>misses; and the EEM measure. Please revise the wording to resolve this outstanding

issue.

(2) (a) SI, Section 1.2 Hormone levels: What are the implications of the inhomogeneous variances across groups for Day 1 to Day 2 changes in salivary and serum E2 [saliva: $F(4,120)=10.66$, $p<.001$; serum: $F(4,120)=12.68$, $p<.001$]. The inhomogeneity is a source of noise, physiological or otherwise, associated with the manipulation of E2 that may have relevance for the interpretation observed fMRI results, particularly given that the 2 and 4 mg groups have 10 fewer participants than the placebo group in the fMRI analyses, and possibly for the behavioural analyses (please see later comments below). The authors should comment on the implications of this new finding for their results.

(b) Supplementary Figure 1. Please plot the values from the saliva E2 levels and not just the serum E2, as per the title: Hormone levels in serum and saliva. Both should be plotted to align with the revisions to Supplementary Table 2. Also, state whether mean or median values are plotted both in the figure legend and y-axis for each of the graphs.

These considerations are important because the new analysis with Levene's test for homogeneity of variances indicates inhomogeneous variances between groups for Day 1 to Day 2 changes in salivary and serum E2 [saliva: $F(4,120)=10.66$, $p<.001$; serum: $F(4,120)=12.68$, $p<.001$] and because Supplementary Figure 3 plots statistical t-map of relationships between salivary E2 and rCBF and regressions in Supplementary Table 10 are also based on salivary E2.

Also, in the methods the authors state, "On all testing days, participants rated their mood on a standardized questionnaire and gave saliva samples". So, please also plot Day 3 saliva E2 values – i.e., the day of the recognition memory test. This has been omitted on the current figure and obviously of interest in terms of E2 drop-off/change and the potential implications for behaviour on Day 3.

(3) In relation to Supplementary Table 12 Please update table to provide information on the number of cases for all of the behavioural analyses.

"Interestingly, we identified different patterns in which neural activity associated with hippocampus-dependent memory formation responded to E2 dose: we found an inverted U-shape in the medial posterior hippocampus but a monotonically increasing dose-response function across the full range of E2 levels in the lateral posterior hippocampus."

(4) The modulations in neural activity associated with E2 dose have not been shown to be associated with hippocampus-dependent memory 'formation' in this study, per se. At the very least, to support such a conclusion the results would need to have revealed a significant association between an E2-mediated modulation in hippocampal activity and successful encoding or differences recognition memory performance – i.e., the modulation in hippocampal activity would need to be relevant for how well participants recognized subsequent old pictures.

Also, there needs to be a bit more caution in giving precedence in the discussion to

subregional parcellation of the hippocampus and function given that there has been no prior reference the foregoing sections – the later comments in the discussion notwithstanding. Obviously, there is an extensive extant literature on functional heterogeneity and subregional hippocampal organization of the hippocampus, but the reader remains uninformed how this relates to the manipulation of E2 levels, until the extended discussion in the subsequent paragraph of the discussion. This aspect of the discussion section thus comes across as rather post-hoc, when a more measured consideration of the extant literature relevant to this issue in the introduction would be appropriate if this result and its prominence are not to be seen as unexpected and without an a priori basis – however, if the result was not predicted, then this should be flagged up in the discussion.

“...but increases in E2 are beneficial within the levels experienced regularly during the menstrual cycle.”

(6) The nature of the ‘benefits’ should be stated.

“However, the spatial resolution of our fMRI scans is too coarse to relate the different dose-response curves to specific anatomically defined hippocampal subfields and only permits a functional dissociation.”

(7) Is the functional dissociation that between the form of the E2 response functions associated with medial posterior hippocampus and lateral posterior hippocampus? If so, it is unclear why this is being referring to as a functional dissociation because it has not been shown to have consequence for function/i.e., the behavioural measures – essentially, there is a coarse resolution based (probabilistic) anatomical dissociation associated with the two E2 dose-response functions.

“What might seem surprising is that whereas E2 did not affect the encoding success of hippocampus-dependent memory in the task used here (Figure 3), it modulated the activity related to successful encoding of exactly these memories. ”

“E2 positively affected memory-related processing but that any beneficial effects might return to baseline only at supraphysiological levels.”

(8) In the first sentence, it is unclear what the authors intended to convey here and in the second sentence, it is unclear what is the rationale for this prediction and the sentence also needs to be revised to improve clarity.

Minor

Figure 1. Study design.

(1) Please change the reference to “picture retrieval” on the figure to visual recognition memory or similar – the nature of the memory test should be stated and aligned with the main text and legend. Also, check the main text to ensure that reference is made to the

recognition memory test and not to a memory retrieval test.

“Of interest is also that E2 did not modulate amygdala activity during affective processing and memory formation, implicating that previous reports of such effects in menstrual cycle studies should be attributed to progesterone instead.”

(2) Should read, “, implying that..”

“neural activity during encoding were visible 24 hours after first E2 intake, this time span was too short to also manifest in behavioral changes after a consolidation period of 24 hours in the current paradigm.”

(3) Please revise to improve clarity.

(4) More generally, please update the text to indicate which atlas has been used to assign the anatomical labels in the fMRI results

REVIEWERS' COMMENTS:

Reviewer #6 (Remarks to the Author):

This study explored the effects of estrogen on memory and brain activity in humans. Participants were given varying doses of estrogen and were tested on episodic memory formation. The study used a subsequent memory paradigm in which participants were scanned with fMRI while encoding a series of neutral and emotional pictures and later tested on both recognition and familiarity.

The paper addresses an interesting question and offers several strengths. The pharmacological design is robust and well thought out and the sample is relatively large. However, the findings are perplexing. In particular, estrogen doses - even the larger ones - do not appear to have an effect on memory, while at the same time the authors report several differences in the fMRI data in areas of interest. The lack of an effect on behavior combined with the reported effects on the brain are difficult to interpret. It raises the question of whether this could be related to the lack of sensitive in the behavioral measurements of memory and whether a more sensitive test would have been more appropriate.

I also had some specific questions about the fMRI design and whether the multiple analyses of BOLD activity in the hippocampus were independent from each other or partly overlapping. This was not quite clear in the description of the many analyses. Similarly, I did not quite follow the description of the PPI analysis and here, again, was left perplexed by what these results mean given the lack of effects on memory.

Altogether, I think this is a well designed study that should be published, but I am not sure what the best outlet would be. I think the results will likely be most valuable to people within this specific field of research.

Response to Reviewer #6

We would like to thank the reviewer for spending the time and effort in providing constructive comments to our work.

Below in blue are our detailed responses to specific comments. Passages from the main manuscript and the Supplementary Material are in italics. Changes in the manuscript itself are marked in red.

The paper addresses an interesting question and offers several strengths. The pharmacological design is robust and well thought out and the sample is relatively large.

Comment 1: However, the findings are perplexing. In particular, estrogen doses - even the larger ones - do not appear to have an effect on memory, while at the same time the authors report several differences in the fMRI data in areas of interest. The lack of an effect on behavior combined with the reported effects on the brain are difficult to interpret. It raises the question of whether this could be related to the lack of sensitive in the behavioral measurements of memory and whether a more sensitive test would have been more appropriate.

Response: We certainly agree with the reviewer that the lack of an effect of E2 on memory performance is at first glance surprising, in particular, given the effects on hippocampal activity. However, we would like to emphasize that our null finding with respect to memory performance is not without precedence. In particular, reports on estrogen effects on learning and memory are much more inconsistent than its effects on neural plasticity also in the animal literature (e.g. Aydin et al., 2008; Fader et al., 1999; Rissman et al., 2002; Vierk et al., 2015; Holmes et al., 2002).

We also would like to point out that it is widely accepted that group differences in neural activity, including those that were pharmacologically induced, do not necessarily result in behaviorally observable differences (e.g. Koester et al., 2013; Marquand et al., 2011; Rasetti et al., 2010). As a potential interpretation of this discrepancy, we suggested in the last version of the manuscript that the 24 hours of heightened E2 levels might have been long enough to increase synapse density or the size of existing spines, thereby increasing glutamate release, as reflected in the fMRI signal, but too short for the new synapses to become functionally effective. An alternative explanation could be that the hippocampal activity that is modulated by E2 does not reflect memory formation but hippocampus-guided visual exploration (Voss et al., 2017).

Having said that, it is certainly possible, as pointed out by the reviewer, that the missing behavioral effect is a lack of sensitivity of the employed task – although we specifically chose this task for its sensitivity (i.e. confidence ratings as well as a remember/know procedure). In response to the reviewer's comment, we have de-emphasized all memory-related claims throughout the manuscript, removed the paragraph with the above-mentioned discussion of the null result, and summarized the discussion of the missing behavioral effect with the following sentence:

*"It therefore seems plausible that a different, **more sensitive** task with the current E2 regimen could elicit observable behavioral changes."*

Aydin, M. *et al.* Effects of letrozole on hippocampal and cortical catecholaminergic neurotransmitter levels, neural cell adhesion molecule expression and spatial learning and memory in female rats. *Neuroscience* **151**, 186–194 (2008).

Fader, A. J., Johnson, P. E. M. & Dohanich, G. P. Estrogen Improves Working But Not Reference Memory and Prevents Amnesic Effects of Scopolamine on a Radial-Arm Maze. *Pharmacology Biochemistry and Behavior* **62**, 711–717 (1999).

Holmes, M. M., Wide, J. K. & Galea, L. A. M. Low levels of estradiol facilitate, whereas high levels of estradiol impair, working memory performance on the radial arm maze. *Behav. Neurosci.* **116**, 928–934 (2002).

Koester, P. *et al.* Decision-making in Polydrug Amphetamine-type Stimulant Users: an fMRI Study. *Neuropsychopharmacology* **38**, 1377 (2013).

Marquand, A. F. *et al.* Pattern Classification of Working Memory Networks Reveals Differential Effects of Methylphenidate, Atomoxetine, and Placebo in Healthy Volunteers. *Neuropsychopharmacology* **36**, 1237 (2011).

Rasetti, R. *et al.* Modulatory Effects of Modafinil on Neural Circuits Regulating Emotion and Cognition. *Neuropsychopharmacology* **35**, 2101 (2010).

Rissman, E. F., Heck, A. L., Leonard, J. E., Shupnik, M. A. & Gustafsson, J.-Å. Disruption of estrogen receptor β gene impairs spatial learning in female mice. *PNAS* **99**, 3996–4001 (2002).

Vierk, R. *et al.* Structure-function-behavior relationship in estrogen-induced synaptic plasticity. *Horm Behav* (2015). doi:[10.1016/j.yhbeh.2015.05.008](https://doi.org/10.1016/j.yhbeh.2015.05.008)

Voss, J. L., Bridge, D. J., Cohen, N. J. & Walker, J. A. A Closer Look at the Hippocampus and Memory. *Trends Cogn. Sci. (Regul. Ed.)* **21**, 577–588 (2017).

Comment 2: I also had some specific questions about the fMRI design and whether the multiple analyses of BOLD activity in the hippocampus were independent from each other or partly overlapping. This was not quite clear in the description of the many analyses.

Response: We agree with the reviewer that the descriptions of the fMRI analyses using the 'remember > know' contrast to test the relationship between E2 increase and hippocampal activity were ambiguous. We have reworded the relevant paragraph as follows:

Confirming the sensitivity of the memory paradigm to detect subtle differences in hippocampal activity, the main effect associated with the remember>know contrast (i.e. irrespective of E2 increase) showed robust effects in the left and right hippocampus [left: (-30, -30, -18); $Z=7.39$, FWE-corrected $p<.001$; right: (24, -9, -18); $Z=5.58$, FWE-corrected $p<.001$; t -tests; $N=118$; Supplementary Figure 4] and a wide network of brain areas (Supplementary Table 9). A linear regression analysis on the whole brain using salivary E2 increase as a predictor and the remember>know contrast estimates as the dependent variable, was then performed to identify areas showing linear relationships between increases in E2 from Day 1 to Day 2 and hippocampal activity. Activity in a cluster in the right posterior hippocampus was positively linearly associated with E2 [(33, -30, -12); $Z=3.69$, FWE-corrected $p=.043$; t -test; $N=118$; Figure 5 A,B]. A quadratic regression analysis using the squared salivary E2 increase as a predictor and again the remember>know contrast estimates as the dependent variable, revealed an inverted U-shaped relationship between hippocampal activity and E2 increases in a more medial cluster within the left posterior hippocampus [(-15, -33, -12); $Z=4.22$, FWE-corrected $p=.006$; t -test; $N=118$; Figure 5 C,D]. Additional analyses showed that these relationships between salivary E2 increase and brain activity were specific to hippocampal regions (see

Supplementary Note 5).

Comment 3: Similarly, I did not quite follow the description of the PPI analysis and here, again, was left perplexed by what these results mean given the lack of effects on memory.

Response: The PPI using the two hippocampal seed regions was added in response to a previous reviewer's comment. We modified the text as follows:

“Finally, exploratory psycho-physiological interaction (PPI)²⁹ analyses did not show any effect of salivary E2 increase on the functional coupling of either of the two posterior hippocampal clusters [peaks at [(33, -30, -12) and (-15, -33, -12), respectively] with other brain regions for remember compared to know trials (see Supplementary Note 6).”

“To explore the effects of salivary E2 increase on the functional coupling between precuneus and brainstem for the processing differences between negative and neutral pictures, a PPI analysis was conducted. For this purpose, we created mask images from the clusters in the precuneus [peak: (6, -60, 15)] and the brainstem [peak: (6, -30, -12)] which showed a negative relationship with salivary E2 increase thresholded at $p_{unc} < .001$. The connectivity analysis (seed: precuneus cluster; region of interest: brainstem cluster) revealed a negative linear relationship between E2 increases and precuneus-brainstem connectivity [(6, -21, -9), $Z=2.94$, FWE-corrected $p=.038$; t-test; $N=118$]. In other words, the greater the E2 increase the smaller the brain activity difference between encoding negative and neutral pictures as well as the less coupled the activity between the precuneus and the brainstem.”